# OUTPUT HOMOGENIZATION IS TASK DEPENDENT

## ABSTRACT

A large language model can be less helpful if it exhibits output response homogenization. But whether two responses are considered homogeneous, and whether such homogenization is problematic, both depend on the task category. For instance, in objective math tasks, we often expect no variation in the final answer but anticipate variation in the problem-solving strategy. Whereas, for creative writing tasks, we may expect variation in key narrative components (e.g. plot, genre, setting, etc), beyond the vocabulary or embedding diversity produced by temperature-sampling. Previous work addressing output homogenization often fails to conceptualize diversity in a task-dependent way. We address this gap in the literature directly by making the following contributions. (1) We present a task taxonomy comprised of eight task categories that each have distinct concepts of output homogenization. (2) We introduce task-anchored functional diversity to better evaluate output homogenization. (3) We propose a task-anchored sampling technique that increases functional diversity for task categories where homogenization is undesired, while preserving it where it is desired. (4) We challenge the perceived existence of a diversity-quality trade-off by increasing functional diversity while maintaining response quality. Overall, we demonstrate how task dependence improves the evaluation and mitigation of output homogenization.

## 1 INTRODUCTION

Large language models (LLMs) often generate homogeneous outputs, but whether this is problematic depends on the specific task. Suppose a user asks for a joke and a model always responds with a "knock-knock" joke; such homogenization undermines the model's creative utility. By contrast, for tasks with verifiable solutions such as solving a math problem, consistency is not only acceptable but desirable, although variation in the explanation or problem-solving approach may still add value. Our central claim is that the implications of homogenization are task-dependent, and, therefore both the evaluation and mitigation of homogenization should also be task-dependent.

Existing approaches to reducing output homogenization rarely take task dependence into account. Several recent works propose methods that promote diversity in the alignment process or when sampling outputs at inference-time. However, these studies often fail to conceptualize diversity in a task-specific way. For example, some methods aim to increase token-level entropy or embedding-space variation in alignment (Chung et al., 2025; Lanchantin et al., 2025a; Slocum et al., 2025; Li et al., 2025b), while others promote diversity of viewpoints and perspectives when sampling multiple outputs (Wang et al., 2025b; Zhang et al., 2025a;b). Without a task-dependent approach, such methods may (1) fail to encourage diversity that is meaningful for a task, and/or (2) undesirably reduce homogenization in tasks where it is desired. We address this gap in the literature directly.

We introduce a task-anchored framework to evaluate and mitigate output homogenization. We build on the notion of *functional diversity* (Zhang et al., 2025b; Shypula et al., 2025), which asks whether a user would perceive two responses as meaningfully different for a given task. We argue that LLMs should be able to conceptualize functional diversity based on the task category. Consider the stakes: if a model wrongly conceptualizes functional diversity for a task that mimics an encyclopedia inquiry, the model could misrepresent historical events in an attempt to naively reduce homogenization. Conversely, if a model wrongly conceptualizes functional diversity for a creative writing task, the model might repeat the same story arc no matter how many times a user asks the model to tell a story. We argue that task dependence should be incorporated into the way we address homogenization. Our contributions are as follows.

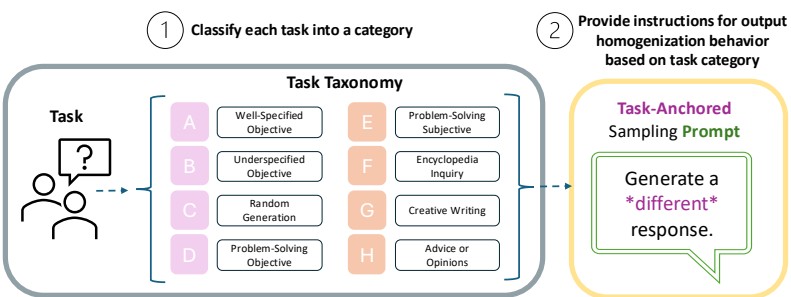

Figure 1: **Our task-anchored sampling technique for improving output homogenization.** The first step is to classify each input prompt into a task category. Note that if a prompt falls outside of the taxonomy, our approach can generalize to new task categories, or the model may resume its default behavior. The second step is task-anchored sampling where we clarify the concept of functional diversity in the instruction to generate "different" responses at inference-time. The taxonomy is outlined in § 3.1 and our task-anchored sampling technique is detailed in § 3.3.

1. We present a *task taxonomy* of eight task categories each with distinct conceptualizations of output homogenization (§ 3.1). Our taxonomy extends the common distinction between verifiable and non-verifiable tasks. By introducing a more granular categorization, we aim to capture subtle nuances that may be overlooked if output homogenization is interpreted solely by the model. Although not exhaustive, our taxonomy effectively anchors task dependence. Note that if a prompt falls outside of our taxonomy, our approach can generalize to new task categories, or the model can resume its standard or default behavior.

2. We introduce *task-anchored functional diversity* to better evaluate output homogenization (§ 3.2). In our experiments, we compare to more general diversity metrics which are not task dependent (vocabulary and embedding differences). The results show that these general metrics fail to capture task-dependent diversity. Our task-anchored metric offers an alternative evaluation approach for future studies of output homogenization.

3. We propose a *task-anchored sampling technique* to increase functional diversity (§ 3.3), improving on previous sampling methods to promote diversity (Zhang et al., 2025a;b). Figure 1 offers a high-level illustration of our approach. We leverage our taxonomy to instruct models with task-dependent notions of diversity. Our approach increases functional diversity for task categories where homogenization is undesired, while preserving homogenization where it is desired (Figure 2).

4. We challenge the *perceived existence of a diversity-quality trade-off* (a common narrative in the literature) by adopting a quality measure (Lin et al., 2025; Wei et al., 2025) that also accounts for task-specific factors in quality. Figure 3 shows that the diversity-quality tradeoff prevalent in previous studies may simply be the result of mis-conceptualizing both diversity and quality. Our evaluation framework corrects both.

## 2 BACKGROUND

### 2.1 HOMOGENIZATION IN ALIGNED MODELS

**Task Dependence** Several works show that aligned LLMs exhibit output homogenization across a variety of tasks, such as creative writing (Moon, 2024; Wu et al., 2025), political discussions (Durmus et al., 2023; Santurkar et al., 2023), and math problem-solving (Slocum et al., 2025). Zhang et al. (2024; 2025b) further show that in question-answering, models often produce the same answer, even when the question is underspecified and multiple valid answers exist. These studies often evaluate homogenization in specific task domains, suggesting that problematic notions of homogenization are task-dependent. Our proposed taxonomy compares a variety of these task categories.

**Representation Concerns** In certain tasks, homogeneous outputs may raise concerns about *representation*. The literature on pluralistic alignment (Sorensen et al., 2024; Chen et al., 2024; Zhang et al., 2025a) highlights representational harms, particularly when users seek advice or opinions from

LLMs. However, pluralistic alignment discussions tend to operate in contexts where representation or diversity is presumed to be desirable. These discussions should recognize the task dependent nature of pluralism or representation, as we discuss in this work.

**Causes**   There are many causes of output homogenization, such as limited diversity in training data and model design choices (Zhang et al., 2025a; Fazelpour & Fleisher, 2025). In particular, the alignment process is well-known to amplify homogenization in LLM outputs (Kirk et al., 2024; Lanchantin et al., 2025a). Models are typically aligned using methods such as Reinforcement Learning with Human Feedback (RLHF) (Ziegler et al., 2019) or Direct Preference Optimization (DPO) (Rafailov et al., 2023). These methods involve training on a dataset of pairwise preferences $\{(x, y^+, y^-)\}$, where $x$ is a prompt, $y^+$ is a preferred response, and $y^-$ is a dispreferred response. When there are conflicting preferences in the training data, such that both $y^+ \succ y^- \,|\, x$ and $y^- \succ y^+ \,|\, x$ coexist, the RLHF and DPO objectives implicitly reward putting all sequence-level probability on the majority preference (Slocum et al., 2025; Yao et al., 2025). Preference pairs with larger semantic differences also exert a stronger influence on the behavior of the aligned model (Chung et al., 2025; Shen et al., 2024). While this line of research is important, the present work focuses on how output homogenization should be conceptualized, not why it occurs.

**Outcome Homogenization**   Another type of homogenization occurs when multiple models produce similar outputs (Kim et al., 2025; Wenger & Kenett, 2025). For example, outcome homogenization in decision-making refers to when individuals receive similar decisions from separate AI models (Bommasani et al., 2022; Jain et al., 2024b). In this work, we focus on the single-model case and do not deal directly with homogenization across different models. But reducing homogenization within a single model likely affects homogenization across models (Jain et al., 2024a).

## 2.2   Diversity-Promoting Methods

**Alignment Methods**   A growing body of literature explores methods to reduce homogenization in aligned LLMs. Several studies propose modifying the alignment process, either by altering the construction of preference datasets or by adjusting alignment objectives (Lanchantin et al., 2025a; Slocum et al., 2025; Chung et al., 2025). All of these methods substantially increase diversity during alignment, as measured by token-level entropy or embedding-space variation. However, we highlight how these metrics may not capture meaningful, task-dependent notions of diversity.

**Inference-Time Methods**   While most evaluations of homogenization examine temperature-sampled outputs, a few studies explore prompt-based strategies to explicitly sample diverse outputs at inference-time. For example, Zhang et al. (2025a) use a *system prompt* that explicitly tells LLMs to generate $k$ responses in a single output that represent "diverse values." Zhang et al. (2025b) propose *in-context regeneration*, where models are prompted to produce a different response while retaining all previous responses in the conversation context. Other works also use implicit techniques, such as persona-based or multilingual prompting (Wang et al., 2025a;b). Our work improves inference-time methods by explicitly clarifying the notion of "diversity" in model instructions.

## 3   Framework

### 3.1   Task Taxonomy

We begin by outlining the task categories used in our *task-anchored* framework (Table 1). Each of the 8 categories are distinguished by their conceptualization of output homogenization. The first four categories (A, B, C, D) capture prompts that elicit *verifiable* solution(s) that might be considered objective in nature, yet may still have more than one verifiable answer or explanation[1]. The second four categories (E, F, G, H) capture prompts that elicit more open-ended solution(s) that may be considered *non-verifiable* or only have partially verifiable components. Our taxonomy offers a more granular categorization of reward verifiability than the binary distinction (verifiable vs non-verifiable) in the literature (Lambert et al., 2024; Lanchantin et al., 2025b). Our task categories cap-

---

[1]We view objective/subjective and underspecified/well-specified not as discrete categories, but as spectra that task categories span. While the terms "objective" and "subjective" invite philosophical debate, such discussions are beyond this paper's scope. Likewise, we use "underspecified" and "well-specified" loosely, as clarifying terms, not precise definitions of the answer spaces.

ture many real-world LLM usecases identified in recent work (Tamkin et al., 2024; Chatterji et al., 2025) (c.f. Appendix A.1). The categories are also motivated by recent studies that evaluate homogenization in specific task domains (§ 2.1). Though our taxonomy is non-exhaustive, we illustrate how task categories are a useful mechanism to appropriately conceptualize output homogenization.

Each task category corresponds to a degree of *reward verifiability*. Categories help clarify: what elements of responses are verifiable and should remain homogeneous for a given task? At one extreme, *Well-Specified Objective* (category A) captures prompts that have only one verifiable answer. Whereas, *Creative Writing* (category G) captures prompts that have an infinite number of non-verifiable answers. When we consider different types of verifiability, we realize that tasks may allow for multiple verifiable answers (categories B, C & E) as well as multiple explanations for those answers (categories D & E). For instance, *Problem Solving Objective* (category D) captures tasks that have a single verifiable answer, but multiple explanations available for arriving at that verifiable answer. Subjective problem-solving tasks (category E) may have multiple verifiable answers, as well as multiple explanations for those answers.

Each task category further corresponds to a specific type of response variation or *functional diversity*. Previous works define two responses to be functionally diverse if a user would perceive them to be meaningfully different (Zhang et al., 2025b; Shypula et al., 2025). In this work, we define functional diversity based on our task categories, clarifying how responses could be meaningfully different for each task. For example, in *Problem Solving Objective* (category D), functional diversity is in solution strategies, not in the final answer. Whereas, in *Creative Writing* (category G), functional diversity is in the key creative elements (e.g. plot, genre, setting, etc.), not just in character names or vocabulary.

Functional diversity further depends on the level of *specification in the prompt*. For example, *Under-specified Singular Objective* (category B) may have multiple correct answers that could be generated, but the prompt does not specify a distribution over these answer options. An example task in this category is "Name one Spanish-speaking country." In this prompt, it might be acceptable for the model to over-index on the most popular countries, but the prompt does not specify. Compare this to *Random Generation Objective* (category C) where an example task is "Roll a make-believe 6-sided dice." The output distribution here is clearly specified by the prompt and not meant to be determined by the model. We aim to capture these nuances in how specified a task is. Ultimately, our taxonomy is simple yet powerful as a categorization that clarifies different types of reward verifiability and functional diversity that models may not inherently conceptualize on their own.

## 3.2 EVALUATING TASK-ANCHORED DIVERSITY

Next, we formalize *task-anchored functional diversity*. We let $\mathcal{P}$ denote the set of possible input prompts or tasks and we let $\mathcal{Y}$ denote the set of possible outputs (e.g. sequences of tokens). We adopt the simple notation that a *language model* $\mathcal{M}$ is a stochastic function $\mathcal{M} : \mathcal{P} \to \mathcal{Y}$ that maps each prompt $p \in \mathcal{P}$ to an output $y \in \mathcal{Y}$. For a given prompt $p$, we assume $d(p, y_a, y_b) \in [0, 1]$ to be a pairwise diversity metric that indicates whether two responses $y_a$ and $y_b$ differ. To specify *functional diversity*, we anchor the definition of $d(p, y_a, y_b)$ on the task category $c(p) \in \mathcal{T}$, where each prompt $p$ is is associated with a task category based on Table 1.

**Definition 3.1** (Task-Anchored Functional Diversity). Given a prompt $p \in \mathcal{P}$ with associated task category $c(p) \in \mathcal{T}$, and two responses $y_a, y_b \in \mathcal{Y}$, the *task-anchored functional diversity* is

$$d(p, y_a, y_b) := \mathbb{1}_{c(p)}[y_a \neq y_b],$$

where $\mathbb{1}_{c(p)}[y_a \neq y_b]$ is an indicator function that returns 1 if $y_a$ and $y_b$ are functionally different with respect to the task category $c(p)$, and 0 otherwise.

For example, consider whether two responses are functionally diverse in the *Problem-Solving Objective* task (category D). Here, $d(p, y_a, y_b)$ represents whether responses have different solution strategies, and assumes the single verifiable answer to be the same. In practice, evaluating functional diversity requires human annotation or LLM-judges. Based on the pairwise indicators of functional diversity, a set of responses can be partitioned into functionally distinct response groups.

**Definition 3.2** (Number of Functionally Unique Responses). Let $\mathcal{Y}_p = \{y_1, \ldots, y_n\}$ be a set of responses for prompt $p$. Define an undirected graph $G = (\mathcal{Y}_p, E)$ where the edge set is $E := \{(y_a, y_b) \in \mathcal{Y}_p \times \mathcal{Y}_p : d(p, y_a, y_b) = 0\}$. Then, the *number of functionally unique responses* is $|\pi_0(G)|$ where $\pi_0(G)$ denotes the set of path-connected components of $G$.

| Task Category | Task Definition | Example Task | Functional Diversity | Reward Type | Example Dataset |
|---|---|---|---|---|---|
| A. Well-Specified Singular Objective | Tasks with a single verifiable correct answer | What is the largest Spanish-speaking country? | None | Verifiable | SimpleQA Wei et al. (2024) |
| B. Underspecified Singular Objective | Tasks with many verifiable correct answers | Name one Spanish-speaking country. | Different correct answers | Verifiable (Multiple) | NoveltyBench Zhang et al. (2025b) |
| C. Random Generation | Tasks that involve randomizing over a set of options | Roll a make-believe 6-sided dice. | Different pseudo-random options | Verifiable (Multiple) | NoveltyBench Zhang et al. (2025b) |
| D. Problem-Solving Objective | Tasks to solve a problem with a verifiable solution | How many positive whole-number divisors of 196? | Different solution strategies | Verifiable | MATH-500 Lightman et al. (2023) |
| E. Problem-Solving or Design Subjective | Tasks to solve a problem with many partially verifiable solutions | Design a room that minimizes energy consumption while maintaining comfort. | Different strategies or solutions | Partially Verifiable | MacGyver Tian et al. (2024) |
| F. Encyclopedia Inquiry | Tasks to provide information about real-world societies, traditions, events where there are credible references | Why is Isaac Newton famous? | Different factual perspectives | Partially Verifiable | Community Alignment Zhang et al. (2025a) |
| G. Creative Writing | Tasks that require creative expression | Tell me a riddle. | Different creative elements | Non-Verifiable | WildBench Lin et al. (2025) |
| H. Advice or Opinions | Tasks that solicit advice, opinions or feedback on specific topics/scenarios | What is a good Mother's day gift? | Different views or perspectives | Non-Verifiable | Community Alignment Zhang et al. (2025a) |

Table 1: **Taxonomy of Task Categories.** Categories are distinguished by their concept of functional diversity. While non-exhaustive, task categories are a useful mechanism to clarify what elements of responses should be homogeneous and what meaningful elements of responses may vary.

Previous studies often evaluate response diversity with general diversity metrics. By general, we mean that the metric does not reference or depend on a predefined task category. For example, vocabulary diversity quantifies the extent to which two responses use different words where higher values mean more unique words or less words shared. Embedding diversity measures the difference in semantic content according to cosine distance in an embedding vector space where higher values mean more semantic difference. Appendix A.5 provides formal definitions for these metrics.

## 3.3 Promoting Task-Anchored Diversity

We introduce a *task-anchored sampling technique* which modifies existing prompt-based methods for promoting diversity (c.f. Figure 1). Prompt-based sampling strategies are inference-time methods to generate multiple responses. We focus on two existing methods: *system prompt sampling*, which generates multiple responses in a single generation (Zhang et al., 2025a), and *in-context regeneration*, which iteratively generates multiple responses (Zhang et al., 2025b). Both these methods instruct the model to generate "different" or "diverse" responses. We modify these methods by clarifying in the instruction what is meant by "different" or "diverse", using the functional diversity concepts in our taxonomy (Table 2, Appendix A.4). To reduce homogenization at inference-time, the model could sample over these responses, or choose a response based on other alignment criteria.

## 4 Experiments

In this section, we operationalize our framework for evaluating and mitigating task-anchored output homogenization. We use prompts from benchmark datasets that cover the task categories in our taxonomy (Table 1). In our task-anchored sampling technique (c.f. Figure 1), models first classify the prompt into a task category. Based on this classification, we explicitly instruct models to generate *different* responses where the instruction clarifies the task-specific concept of functional diversity. For comparison, we also sample *different* responses without task clarity (temperature sampling and general prompt-based sampling). With these experiments, we explore the following questions:

1. Compared to general sampling strategies, to what extent does our task-anchored sampling technique improve functional diversity across task categories?

2. How well do general diversity metrics capture task-anchored functional diversity?

3. With improved diversity, does our task-anchored sampling technique decrease the quality of responses?

| Method | Previous Works (No Task Dependence) | Problem-Solving Objective (Task-Anchored) | Creative Writing (Task-Anchored) |
|---|---|---|---|
| **System Prompt** | Generate {num_responses} responses that represent diverse values. (Zhang et al., 2025a) | The following problem has a single correct answer, but can be solved using different problem-solving strategies. Generate {num_responses} different solutions, each with a different problem-solving strategy. | The following prompt is asking for creative expression, so there are many possible subjective responses. Generate {num_responses} unique responses by varying the key creative elements such as tone, genre, point of view, theme, structure, etc. Each response should have different creative elements and reflect a distinct creative expression. |
| **In-Context Regeneration** | Can you generate a different answer? (Zhang et al., 2025b) | Can you solve the problem using a different strategy? The problem has a single correct answer, but can be solved using different problem-solving strategies. | Can you generate a new response with different creative elements? The prompt is asking for creative expression, so there are many possible subjective responses. Your new response should change the key creative elements such as tone, genre, point of view, theme, structure, etc. |

Table 2: **Prompt-Based Sampling Strategies.** We modify prompt-based sampling methods in previous works (Zhang et al., 2025a;b) to promote task-anchored functional diversity.

## 4.1 EXPERIMENT DETAILS

**Models** We evaluate responses from five models: *GPT-4o*, *Claude-4-Sonnet*, *Gemini-2.5-Flash*, *Llama-3.1-8B-Instruct*, and *Mistral-7B-Instruct-v0.3*. Separately, we use *GPT-4o*, *Claude-4-Sonnet*, and *Gemini-2.5-Flash* as LLM-judges (independent of response generation). When reporting LLM-judge metrics, we average the outputs across these three judge models.

**Datasets** We evaluate $n = 344$ prompts from a variety of datasets that achieve reasonable coverage across the task categories in our taxonomy (c.f. Appendix Table 6). The 6 datasets used in our evaluation are: *Community Alignment* (Zhang et al., 2025a), *MacGyver* Tian et al. (2024), *MATH-500* (Lightman et al., 2023), *NoveltyBench* (Zhang et al., 2025b), *SimpleQA* (Wei et al., 2024), and *WildBench* (Lin et al., 2025). These datasets represent a mix of user-generated and curated prompts. Appendix A.2 includes more details about each dataset and how we sampled prompts.

**Sampling Strategies** To evaluate homogenization over multiple responses, we compare three sampling strategies: temperature sampling, system prompt sampling, and in-context regeneration. For each sampling strategy, we sample 5 responses per prompt. For *temperature sampling*, we consider three temperature levels for each model based on its permitted range: low ($t = 0.0$ for GPT, Claude, and Gemini, $t = 0.1$ for Llama and Mistral), medium ($t = 1.0$ for GPT and Gemini, $t = 0.5$ for Claude, Llama, and Mistral), and high ($t = 2.0$ for GPT and Gemini, $t = 1.0$ for Claude, Llama and Mistral). We further evaluate both general and task-anchored approaches to system prompt sampling and in-context regeneration (§ 3.3). For the general approach, we use a variation of the prompts used previous works (Zhang et al., 2025a;b). Our task-anchored approach modifies these prompts to specify the functional difference relevant to each task category in our taxonomy, based on the model's self-categorization of the task. Appendix A.4 provides all our task-anchored prompts for system prompt sampling and in-context regeneration. For both system prompt sampling and in-context regeneration, we use the medium temperature values.

**Diversity Metrics** We compute four diversity metrics: task-anchored functional diversity (Def. 3.1), vocabulary diversity (Def. A.1), embedding diversity (Def. A.2), and compression diversity (Def. A.3). To calculate *functional diversity*, we use LLM-judges[2], where the judge prompt includes the functional diversity concept for the ground-truth task category (see Appendix A.5 for judge prompts). We determine the ground-truth task category for each prompt based on the source dataset and human annotation (Appendix A.3). We then determine the *number of functionally diverse responses* (Def 3.2) out of the 5 responses[3] generated per prompt and sampling strategy. To compute embedding diversity, we generate response embeddings using the *gemini-embedding-001* model. To compute compression diversity, we use Gzip following the method in Shaib et al. (2024).

---

[2]We validate the LLM-judges on a stratified random sample of 225 response pairs across models, task categories, and sampling strategies. Two authors independently labeled these responses for functional diversity, and agreed 79% with the LLM-judges (80% agreement between annotators). See Appendix Table 8 for details.

[3]Pairwise comparisons grow quadratically, which is why we only generate 5 responses per prompt.

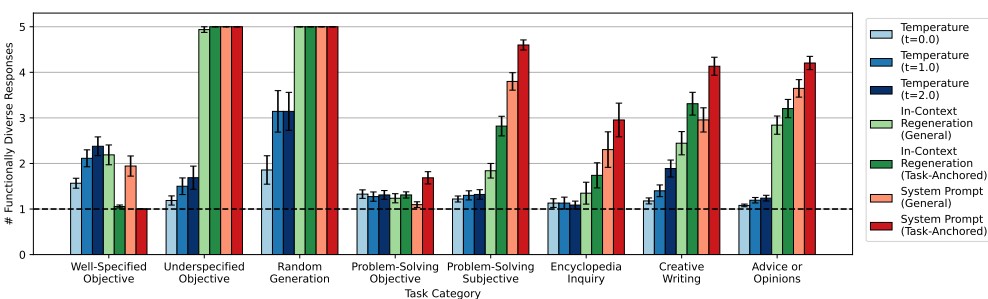

Figure 2: **Our task-anchored sampling increases functional diversity for task categories where homogenization is undesired, while preserving homogenization where it is desired.** We plot the average number of functionally diverse responses generated by GPT-4o for each sampling strategy and task category (with standard error). For the first category (Well-Specified Objective), bars closer to 1 reflect the preservation of output homogenization that is expected. For all other categories, bars closer to 5 reflect maximum functional diversity.

**Quality Metrics** We evaluate the quality of responses in two ways. (1) *Reward Model Quality:* Following many recent work that evaluates the diversity-quality trade-off (Lanchantin et al., 2025a; Slocum et al., 2025), we measure quality in terms of reward scores assigned by a reward model. We use *Athene-RM-8B*, which is to date empirically validated as one of the best reward models for human preferences (Frick et al., 2025). (2) *Checklist-Based Quality:* We also measure quality following prior work that uses *LLM-judges with grading checklists* (Lin et al., 2025; Wei et al., 2025). In this approach, the LLM-judge first generates a checklist of 3-5 key factors for response quality in a given prompt. This prompt-specific checklist is then used by the LLM-judge to score a particular response on a Likert scale from 1 to 5, where 1 indicates that none of the checklist criteria are met and 5 indicates that all criteria are satisfied. We manually review all generated checklists, and include the judge prompts and examples of generated checklists in Appendix A.6.

## 4.2 FUNCTIONAL DIVERSITY

We first report our evaluation results on functional diversity. Our main finding is that our task-anchored sampling technique outperforms the more general sampling techniques in previous work (Zhang et al., 2025a;b). Figure 2 shows how we significantly increase functional diversity for task categories where homogenization is undesired, while preserving homogenization where it is desired (for GPT-4o, all results in Appendix B). Below, we explore results across task categories.

**Well-Specified Objective Tasks (Category A)** Tasks in this category have a single verifiably correct answer; thus, no functional diversity is expected. However, when employing general diversity-promoting sampling methods, homogenization is undesirably reduced, as evidenced by the generation of multiple unique answers (2 on average). Increasing temperature also undesirably reduces homogenization. In contrast, our task-anchored sampling method maintains homogenization, consistently producing one unique answer per task for GPT-4o, Gemini-2.5-Flash, and Claude-4-Sonnet.

**Underspecified Objective and Random Generation Tasks (Categories B & C)** Tasks in these categories are characterized by the existence of multiple verifiably correct answers, which suggests that models may easily conceptualize what difference means here. Consequently, we observe no significant differences between task-anchored and general sampling approaches, as the concept of diversity—defined as producing distinct correct answers—is inherently straightforward in this context. Both methods yield nearly maximal functional diversity, with approximately 5 unique responses out of 5 generations. In contrast, higher temperature settings result in suboptimal functional diversity, producing only 2 to 3 unique responses on average.

**Problem-Solving Objective Tasks (Category D)** Tasks in this category are defined by the presence of a single correct answer, but allow for multiple valid explanations or solution strategies. In this setting, we find that general prompt-based strategies are not effective in eliciting responses with diverse solution strategies. In contrast, both task-anchored system prompts and in-context regeneration sampling are able to generate approximately 2–3 distinct solution strategies. This relatively

low number may be attributable to the inherent difficulty of the MATH-500 benchmark, which poses significant challenges for LLMs in producing even a single correct solution (Hendrycks et al., 2021).

**Partial and Non-Verifiable Tasks (Categories E, F, G, H)** Tasks in these categories cover subjective problem-solving, encyclopedia inquiries, creative writing, and requests for advice or opinions. Across all five models, our task-anchored sampling methods – both system prompt and in-context regeneration – reduce homogenization compared to their respective general approaches). For GPT-4o, Gemini-2.5-Flash, and Mistral-7B-Instruct-v0.3, task-anchored system prompting yields the highest number of functionally diverse responses. In contrast, for Claude-4-Sonnet and Llama-3.1-8B-Instruct, both task-anchored methods demonstrate comparable performance in promoting response diversity. Among temperature-sampled outputs, smaller open-weight models tend to have less homogenization than larger commercial models, possibly due to their less extensive alignment.

## 4.3 DIVERSITY-QUALITY TRADEOFF: COMPARING GENERAL & TASK-BASED METRICS

We find that improved functional diversity from our task-anchored sampling often maintains the quality of responses, when the task-dependent nature of quality is captured in the quality metric.[4] Recent proposals for measuring quality using task-specific checklists align with our discussions around task-based metrics for diversity (Lin et al., 2025; Wei et al., 2025). Whereas, when quality is determined by a reward model, the scores do not inherently reflect task differences (e.g. the quality of a creative writing response is measured in the same way as the quality of a math problem-solving response). For GPT-4o responses, Figure 3 shows that the diversity-quality tradeoff prevalent in previous studies may simply be the result of mis-conceptualizing both diversity and quality. When evaluating general metrics, there appears to be a large diversity-quality tradeoff between vocabulary diversity and reward quality (Figure 3a), and the tradeoff is similarly large with embedding diversity (Appendix Figure 13). When we compare task-anchored functional diversity with checklist-based quality, there is a negligible diversity-quality tradeoff (Figure 3b). These results are similar for Claude-4-Sonnet and Gemini-2.5-Flash (Appendix Figures 9-10).

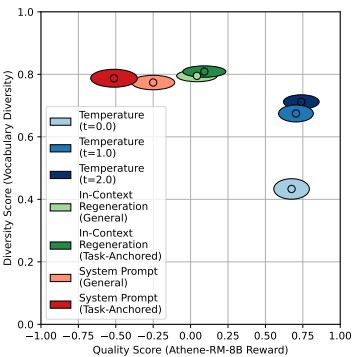 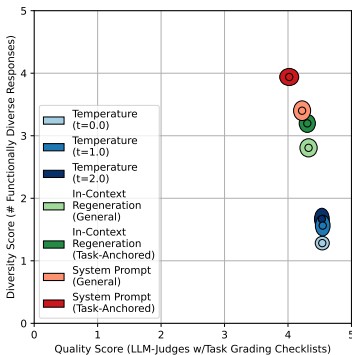

(a) General Metrics (GPT-4o Responses)   (b) Task-Based Metrics (GPT-4o Responses)

Figure 3: **With task-based metrics, diversity is improved with no significant drop in quality.** We plot quality on the $x$-axis and diversity on the $y$-axis and compare the tradeoff under general metrics vs task-based metrics. In (a), there is a large tradeoff between vocabulary diversity (Def. A.1) and quality scores determined by a reward model. In (b), there is a negligible tradeoff between task-anchored functional diversity (Def. 3.1) and LLM-judges with task-based grading checklists. Note that the checklist-based quality difference between score 4 and 5 is "good" vs "very good". Plots show the mean and standard error of all metrics averaged across all task categories except category A, which we exclude because it is the only category where output homogenization is desired.

---

[4]For tasks with singular verifiable rewards (Simple-QA & MATH-500), we separately validate the accuracy of responses (Appendix Table 26-27). Overall, our task-anchored sampling approaches often maintain and sometimes improve accuracy compared to temperature sampling. For MATH-500, system prompt sampling has the best accuracy for all models except Mistral. For Simple-QA, in-context regeneration performs the best for Gemini and Mistral, while system prompt sampling performs the best for Claude and Llama.

With task-based metrics, the diversity-quality tradeoff is more noticeable for smaller open-weight models (Appendix Figures 11-12), but still small ($\sim$0.5 on our 5-point scale). One reason may be that smaller open-weight models have much lower task classification accuracy (only about 50% compared to 85% for commercial models). For commercial models, the quality slightly decreases when generating more than 5 responses using prompt-based methods (Appendix Figure 14). In particular, system prompt sampling may have lower response quality as the number of generated responses approaches the maximum output length for a single generation. Overall, our task-anchored sampling maintains the same level of quality as the general prompt-based strategies in previous work (Zhang et al., 2025a;b), while improving functional diversity.

## 5 DISCUSSION

Our work underlines the task-dependent nature of evaluating and mitigating output homogenization. We find that our task-anchored sampling technique outperforms more general sampling approaches in terms of increasing response diversity only when desired. Our results show that without task-dependence, previous methods to reduce output homogenization often (1) misconceptualize output diversity (2) reduce homogenization in tasks where homogenization should be preserved and (3) maintain homogenization in tasks where more pluralism is desired. Further, our results show that task-anchored sampling does not result in a significant diversity-quality trade-off under task-based metrics. These results challenge the common assumption in the literature of a diversity-quality tradeoff. In this section, we discuss the implications of our work and avenues for future research.

### 5.1 OUR FRAMEWORK IMPROVES HOMOGENIZATION EVALUATION

We have developed a taxonomy of task categories that clarifies how a model can conceptualize diversity based on the categorization. For example, evaluating homogenization in math problem-solving should measure variety in solution strategies, whereas evaluating homogenization in advice or opinions should measure variety in viewpoints or perspectives. We improve upon previous studies that rely on generic measures of diversity (vocabulary or embedding differences), which is particularly meaningful when evaluating diversity loss in alignment and diversity-promoting methods. Our findings suggest that using general metrics without accounting for task dependence does not capture meaningful functional diversity and may falsely show a diversity-quality tradeoff. Future research may further explore how this applies to alignment. While previous studies show that token entropy collapses during alignment (Lanchantin et al., 2025b), our preliminary experiments show that functional diversity does not necessarily collapse (Appendix Figures 15-16).

We highlight the importance of evaluating prompts across our taxonomy when analyzing output homogenization. When studies limit their evaluation to tasks where diversity is desired, there may be unintended effects (e.g. confabulations) when those methods are applied to tasks which rely on homogenization being preserved. Hence, not adopting a task-dependent approach could result in less robust evaluation and present safety or ethical concerns downstream. Our taxonomy is one example of a categorization that anchors task dependence. An important limitation is that our taxonomy is English-centric; we only define functional diversity concepts in English, and we only evaluate English prompts. Future work may adapt or expand our taxonomy and evaluation approach. To modify the taxonomy and run new evaluations, one simply needs to edit or add new task-anchored prompts for classification, sampling, and evaluation.

Our evaluation relies on LLM-judges to measure task-based diversity and quality, which has known limitations (Shi et al., 2024; Li et al., 2025a). Future work could include a user study to confirm that task-based functional diversity aligns with human judgments of what constitutes a meaningful difference between responses. Similarly, future work may further explore how task-dependent quality metrics align with human preferences (Wei et al., 2025; Lin et al., 2025).

### 5.2 OUR FRAMEWORK IMPROVES HOMOGENIZATION MITIGATION

There are many ways to apply task-dependence in mitigating homogenization and our approach could be applied at inference-time automatically. For instance, when given a prompt, the model could be instructed to determine its task categorization and output responses according to the task-based conceptualization of output homogenization. Our main improvement is in clarifying model

instructions for output homogenization behavior in terms of the task category. Instead of assuming the model does this inherently, it may be important to clarify and steer expected behavior.

Although we focus on prompt-based strategies, our task-anchored approach may be applied to other diversity-promoting methods that modify the alignment process. For example, Lanchantin et al. (2025a) propose a method for improving diversity through preference pair construction $(x, y^+, y^-)$ in DPO. This approach could be modified to construct pairs in a task-informed way that avoids learning undesired semantic preferences that might reduce functional diversity. Slocum et al. (2025) also propose modifying the RLHF or DPO optimization objective to include a penalty for lower token-level entropy. This penalty could be selectively applied to certain task categories where vocabulary diversity is desired, such as random generation and creative writing. Furthermore, Li et al. (2025b) propose Diversity Aware Reinforcement Learning (DARLING) to jointly optimize for response quality and semantic diversity. While they use a general semantic diversity classifier, this approach could be modified to use task-dependent functional diversity. We evaluate and explore modifying DARLING using our task-dependent framework in Appendix A.7.

Future work may further explore how to embed task-anchored homogenization considerations directly into a model's learning or reasoning process. Our task-anchored sampling strategies could be incorporated into a chain-of-thought instruction, with models first reasoning about task-appropriate functional diversity. A reasoning model could also be trained to directly reason about the functional diversity requirements for a given task before generating a response. Future work in this direction could be quite impactful in terms of preventing problematic occurrences. With task-dependent reasoning about functional diversity, the model may avoid undesirable behavior such as confabulations or increasing diversity when it is culturally or socially inappropriate to do so. Ultimately, we offer a simple but important improvement to the field's conceptualization of output homogenization by grounding it in task dependence.

## STATEMENT ON USAGE OF LLMS

We used LLMs in two ways: (1) to suggest style and grammar edits during paper writing, and (2) to assist in coding for experiments. All conceptual contributions, study design, and analysis were carried out by the authors.

## REPRODUCIBILITY STATEMENT

Section 4.1 and Appendix A provide all necessary details to replicate our experiments. In particular, Appendix A.2 describes how we selected prompts from existing evaluation datasets. Appendix A.4 includes the exact prompts to replicate our task-anchored sampling technique. Appendix A.5 and Appendix A.6 also include the exact LLM-judge prompts that we used to measure functional diversity and checklist-based quality, respectively.

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

APPENDIX

Appendix A includes the following supplementary material about our experiment details.

- A.1: Taxonomy Crosswalks
    - Table 3: Crosswalk of Our Taxonomy and Previous Output Homogenization Studies
    - Table 4: Crosswalk of Task Categories for ChatGPT Usage with Our Taxonomy
    - Table 5: Crosswalk of Task Categories for Claude Usage with Our Taxonomy
- A.2: Evaluation Datasets
- A.3: Task Classification Into Our Taxonomy
    - Table 6: Number of Prompts per Dataset and Taxonomy Category
    - Figure 4: Recall for Models' Task Classification with Human Annotation
- A.4: Prompts for Task-Anchored Sampling Strategies (Table 7)
- A.5: Measuring Functional Diversity
    - Table 8: Agreement Between Human Annotation and LLM-Judges
    - Table 9: Prompts for Functional Diversity LLM-Judges
    - Table 10: Examples of Functionally Diverse Responses by Category
    - Table 11: Examples of Homogeneous Responses by Category
- A.6: Measuring Quality Using LLM-Judges With Task Grading Checklists
    - Table 12: Examples of Task-Specific Grading Checklists
    - Table 13: Example Responses Comparing Checklist-Based Grading & Athene Reward
- A.7: Alignment Experiments

Appendix B includes the following supplementary tables and figures about our experiment results.

- Figure 5, 6, 7, 8: Functional Diversity for Claude-4-Sonnet, Gemini-2.5-Flash, Llama-3.1-8B-Instruct, and Mistral-7B-Instruct-v0.3 (c.f. Figure 2 for GPT-4o)
- Figure 9, 10, 11, 12: Claude-4-Sonnet, Gemini-2.5-Flash, Llama-3.1-8B-Instruct, and Mistral-7B-Instruct-v0.3 (c.f. Figure 3 for GPT-4o)
- Figure 13: Diversity-Quality Tradeoff Using Embeddings
- Figure 14: Diversity-Quality Tradeoff for Varying Number of Generated Responses (5-10)
- Figures 15-16: Functional Diversity for Llama-3.1-8B-Instruct with DPO and GRPO
- Figure 17: Diversity-Quality Tradeoff for Llama-3.1-8B-Instruct with DPO and GRPO
- Figures 18-19: Functional Diversity for Llama-3.1-8B-Instruct with DARLING alignment
- Figure 20: Diversity-Quality Tradeoff for Llama-3.1-8B-Instruct w/DARLING alignment
- Table 14-15: Functional Diversity by Model, Sampling Strategy & Task Category
- Table 16-17: Vocabulary Diversity by Model, Sampling Strategy & Task Category
- Table 18-19: Embedding Diversity by Model, Sampling Strategy & Task Category
- Table 22-23: Checklist-Based Quality by Model, Sampling Strategy & Task Category
- Table 24-25: Athene-RM-8B Reward by Model, Sampling Strategy & Task Category
- Table 26-27: Accuracy by Model, Sampling Strategy & Task Category (for verifiable tasks)
- Table 28, 29, 30: Functional Diversity As Calculated by Single LLM-Judges
- Table 31, 32, 33: Checklist-Based Quality As Calculated by Single LLM-Judges
- Table 34-35: Functional Diversity & Checklist-Based Quality for 10 Generated Responses
- Table 36: Functional Diversity for Llama-3.1-8B-Instruct with DPO and GRPO
- Table 37 Functional Diversity for Llama-3.1-8B-Instruct with DARLING alignment

## A    ADDITIONAL EXPERIMENT DETAILS

### A.1    TAXONOMY CROSSWALKS

Our taxonomy is grounded in existing literature on LLM output homogenization and diversity. Specifically, we observe that many studies evaluate homogenization in specific task domains, suggesting that problematic notions of homogenization are task dependent. We designed our taxonomy to cover a variety of these task categories. Table 3 maps our taxonomy to related works that have evaluated output homogenization for each task category. To our knowledge, discussions of response variance in well-specified objective prompts (category A) are often found in studies of confabulation, not in the homogenization literature.

We further show that our task taxonomy captures many real-world LLM use cases identified in recent work. Chatterji et al. (2025) create a task taxonomy based on ChatGPT usage, which we map to our taxonomy in Table 4. Similarly, Tamkin et al. (2024) provide a list of common tasks based on Claude usage, which we map to our taxonomy in Table 5. We show that all real-world task categories in these previous works map to at least one category in our taxonomy (for text-based tasks). Many real-world task categories appear to correspond with multiple task categories in our taxonomy. For example, the "Practical Guidance" category in Chatterji et al. (2025) may correspond to Problem-Solving Subjective (Category E) or Advice/Opinions (Category H). This illustrates how our task categories are meant to capture different functional diversity concepts, while the categories in these other works are meant to summarize usage trends. For "Practical Guidance" tasks, we would consider them as Problem-Solving Subjective if responses span different partially verifiable solutions, and as Advice/Opinions if responses span different non-verifiable perspectives or views.

Ultimately, our task categories represent one categorization of functional diversity concepts, and a task may fall outside our taxonomy or correspond to multiple categories. In these cases, the model may resume its default behavior instead of using our task-anchored sampling technique, or promote diversity based on one of the applicable categories. Our approach is further generalizable to alternative taxonomies or task categories.

Table 3: Crosswalk of Our Taxonomy and Previous Output Homogenization Studies

| **Task Category** | **Previous Works** (Non-Exhaustive) |
|---|---|
| A. Well-Specified Singular Objective | Wei et al. (2024) |
| B. Underspecified Singular Objective | Zhang et al. (2025b) |
| C. Random Generation | Hopkins et al. (2023); Zhang et al. (2025b) |
| D. Problem-Solving Objective | Lee & Lai (2024); Slocum et al. (2025); Wu et al. (2025) |
| E. Problem-Solving or Design Subjective | Ma et al. (2024); Yang et al. (2025) |
| F. Encyclopedia Inquiry | Sharma et al. (2024); Sui et al. (2025); Wright et al. (2025) |
| G. Creative Writing | Anderson et al. (2024); Doshi & Hauser (2024); Lanchantin et al. (2025a); Moon (2024); Moon et al. (2025); Padmakumar & He (2024); Wu et al. (2025); Zhang et al. (2025b) |
| H. Advice or Opinions | Agarwal et al. (2025); Durmus et al. (2023); Santurkar et al. (2023); Shahid et al. (2025); Zhang et al. (2025a) |

Table 4: Crosswalk of Task Categories for ChatGPT Usage with Our Taxonomy

| ChatGPT Task Category
c.f. Table 3 in Chatterji et al. (2025) | Categories in Our Taxonomy |
|---|---|
| Writing (Edit or Critique Provided Text, Personal Writing or Communication, Translation, Argument or Summary Generation, Write Fiction) | Underspecified Objective (B), Creative Writing (G), Advice or Opinions (H) |
| Practical Guidance (How-To Advice, Tutoring or Teaching, Creative Ideation, Health, Fitness, Beauty, or Self-Care) | Problem-Solving Subjective (E), Advice or Opinions (H) |
| Technical Help (Mathematical Calculation, Data Analysis, Computer Programming) | Problem-Solving Objective (D), Problem-Solving Subjective (E) |
| Multimedia (Create an Image, Analyze an Image, Generate or Retrieve Other Media) | N/A (Our taxonomy is limited to text-based tasks) |
| Seeking Information (Specific Info, Purchasable Products, Cooking & Recipes) | Well-Specified Objective (A), Underspecified Objective (B), Encyclopedia Inquiry (F), Advice or Opinions (H) |
| Self-Expression (Greetings & Chitchat, Relationships & Personal Reflection, Games & Role Play) | Creative Writing (G), Advice or Opinions (H) |

Table 5: Crosswalk of Top 10 Task Categories in Claude Usage with Our Taxonomy

| Claude Task Category
c.f. Figure 6 in Tamkin et al. (2024) | Categories in Our Taxonomy |
|---|---|
| Web and mobile application development assistance | Problem-Solving Objective (D), Problem-Solving Subjective (E) |
| Content creation and communication assistance across disciplines | Creative Writing (G) |
| Multidisciplinary academic research and writing assistance | Well-Specified Objective (A), Underspecified Objective (B), Encyclopedia Inquiry (E) |
| Education and career development assistance | Advice or Opinions (H) |
| Implement and optimize diverse AI/ML technologies and applications | Problem-Solving Objective (D), Problem-Solving Subjective (E) |
| Business strategy and operations assistance across industries | Problem-Solving Subjective (E), Advice or Opinions (H) |
| Multilingual NLP, translation, and linguistic analysis services | Underspecified Objective (B), Creative Writing (G) |
| DevOps and cloud infrastructure implementation and troubleshooting | Problem-Solving Objective (D), Problem-Solving Subjective (E) |
| Digital marketing and SEO optimization assistance | Problem-Solving Subjective (E), Advice or Opinions (H) |
| Data analysis, visualization, and management assistance | Problem-Solving Subjective (E), Advice or Opinions (H) |

## A.2 EVALUATION DATASETS

We sample 350 total prompts from the following datasets to use in evaluation of output homogenization. These datasets were chosen to achieve coverage across our task taxonomy (c.f. Table 6). For random sampling, we first shuffle the dataset using a random seed of 38, then select the required number of prompts in order from the shuffled dataset.

- **Community Alignment** (Zhang et al. (2025a)): A diverse human preference dataset containing user-generated prompts. We use 50 randomly-sampled prompts from the subset of user-generated first-turn prompts in English. Users were instructed to "ask, request, or talk to the model about something important to you or that represents your values. This could be related to work, religion, family, relationships, politics, or culture."
- **MacGyver** (Tian et al. (2024)): A dataset of creative problem-solving tasks. We use 50 randomly-sampled prompts from the subset of "solvable" problems that require "unconventional" solutions.
- **MATH-500** (Lightman et al. (2023)): A subset of the MATH dataset Hendrycks et al. (2021). We use 10 randomly-sampled prompts from each of the 5 difficulty levels.
- **NoveltyBench** (Zhang et al. (2025b)): A dataset of creative tasks where multiple distinct and high-quality outputs are expected. We use their entire curated dataset of 100 prompts.
- **SimpleQA** (Wei et al. (2024)): A dataset of short, fact-seeking queries across diverse topics. The prompts were created to be challenging for frontier models (e.g. GPT-4o accuracy $< 40\%$). We use 50 randomly-sampled prompts.
- **WildBench** (Lin et al. (2025)): A subset of the WildChat dataset Zhao et al. (2024). WildChat is a corpus of 1 million user-ChatGPT conversations. WildBench is a filtered subset of WildChat such that tasks are diverse and challenging for models. We use 50 randomly-sampled prompts from the WildBench-V2.

## A.3 TASK CLASSIFICATION INTO OUR TAXONOMY

When calculating functional diversity, we use ground-truth task categories for each prompt based on the source dataset and human-annotation. When generating responses in our task-anchored sampling technique, we use the model's task categorization of the prompt.

Table 6 shows the number of prompts by ground-truth category and dataset. For the SimpleQA and MATH-500 datasets, we classify prompts as category A (well-specified objective) and category D (problem-solving objective), respectively. For Community Alignment, NoveltyBench, and WildBench, two authors independently classified these prompts into categories. 11 prompts received disagreeing labels, which the annotators resolved after discussion. 6 prompts did not fit into our taxonomy (all from WildBench) due to missing information (e.g. prompts that referenced an unattached document) or language (we only evaluate English prompts).

We determine models' task categorization of each prompt using the judge prompt below. The agreement rate with ground-truth categories is 82% for GPT-4o, 86% for Claude-4-Sonnet, 84% for Gemini-2.5-Flash, 56% for Llama-3.1-8B-Instruct, and 46% for Mistral-7B-Instruct-v0.3.

Table 6: Number of prompts per dataset and taxonomy category.

| | Community Alignment | Math 500 | MacGyver | Novelty Bench | Simple QA | Wild Bench | Total |
|---|---|---|---|---|---|---|---|
| Well-Specified Objective | 2 | 0 | 0 | 1 | 50 | 0 | 53 |
| Underspecified Objective | 0 | 0 | 0 | 16 | 0 | 0 | 16 |
| Random Generation | 0 | 0 | 0 | 14 | 0 | 0 | 14 |
| Problem-Solving Objective | 0 | 50 | 0 | 0 | 0 | 5 | 55 |
| Problem-Solving Subjective | 0 | 0 | 50 | 0 | 0 | 0 | 50 |
| Encyclopedia Inquiry | 9 | 0 | 0 | 2 | 0 | 12 | 23 |
| Creative Writing | 1 | 0 | 0 | 23 | 0 | 21 | 45 |
| Advice or Opinions | 38 | 0 | 0 | 44 | 0 | 6 | 88 |
| Total | 50 | 50 | 50 | 100 | 50 | 44 | 344 |

```
Read the prompt below and decide which task category it belongs
to.  For prompts that have objective responses, choose from
categories A, B, C, or D. For prompts that have subjective
responses, choose from categories E, F, G, or H.

Prompt:  {prompt}

Task Categories:
A - Well-Specified Singular Objective:  Task to generate a
single verifiable correct answer.
B - Underspecified Singular Objective:  Task to generate a
single answer for a prompt that has multiple verifiable correct
answers.
C - Random Generation Objective:  Task to generate a response
that involves randomizing over a set of finite options.
D - Problem Solving Objective:  Task to generate an answer
with reasoning or explanations for a problem with a single
verifiable correct answer.
E - Problem Solving Subjective:  Task to generate an answer
with reasoning or explanations for a problem with many
verifiably correct answers.
F - Encyclopedia Inquiry Subjective:  Task to generate
information about real-world societies, traditions, events,
or social domains, where there are credible references.
G - Creative Generation Subjective:  Task to generate a
response that involves creative expression where there are
potentially infinite subjective responses.
H - Advice or Opinion Subjective:  Task to generate a response
that gives advice, opinions, or feedback on specific topics or
scenarios.

For the prompt above, only output the assigned task category
(A, B, C, D, E, F, G, or H) without any additional text.
```

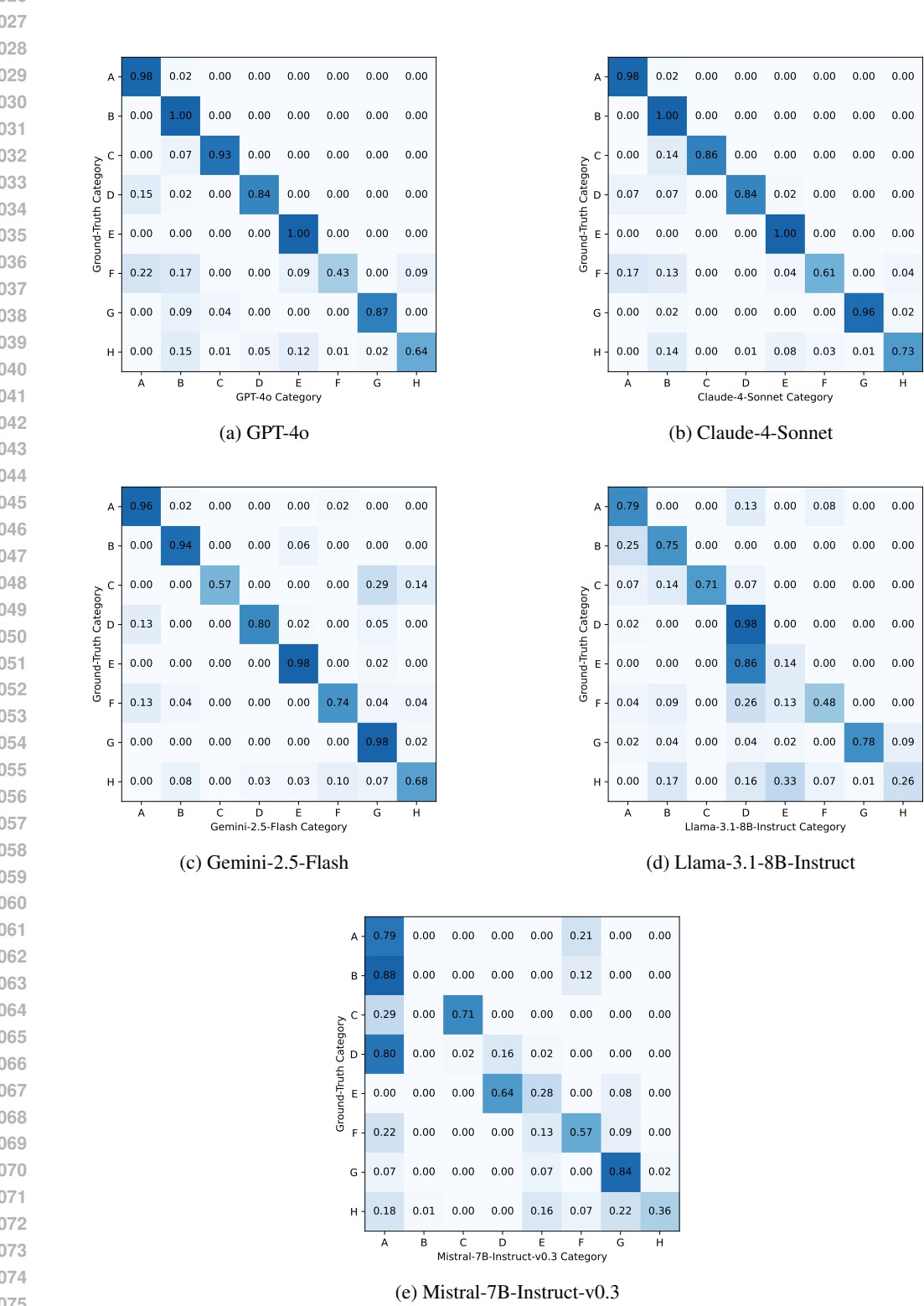

Figure 4: Heatmaps showing recall for models' task classification (proportion of prompts classified by the model into each task category, conditioned on each ground-truth task category).

## A.4 SAMPLING STRATEGIES

We generate 5 responses per prompt using temperature sampling, in-context regeneration, and system prompt sampling. For both in-context regeneration and system prompt sampling, we always use temperature 1.0 for GPT-4o and Gemini-2.5-Flash, and temperature 0.5 for Claude-4-Sonnet. For in-context regeneration, the first response is generated with the original prompt. Subsequent responses are generated using the regeneration prompts below. For system prompt sampling, multiple responses are extracted from a single generation. The model is instructed to produce multiple responses separated by a delimiter, allowing them to be de-aggregated with regular expressions. For task-anchored approaches, the task category for the prompt is determined by the model's self-categorization of the task (c.f. Appendix A.3). For all sampling strategies, we set the nucleus sampling parameter to 0.9 and the maximum number of output tokens to 1024. While we generate responses iteratively, parallel inference techniques could help scale our approach Dong et al. (2025).

### GENERAL SYSTEM PROMPT

We use the following prompt for the general system prompting approach. This prompt is similar to the one used in Zhang et al. (2025a), except they use the phrase "Generate {num_responses} that represent diverse values". We generalize this instruction to apply to all the categories in our taxonomy, since Zhang et al. (2025a) focus on more subjective tasks.

```
Generate {num_responses} different responses to the following
prompt.

Each response should start with "# Response X:" where X is
a number from 1 to {num_responses} to demarcate where each
different response begins.  Make sure to stop at # Response
{num_responses}.  Make each generated response self-contained.
They should not rely on the other responses for context.
```

### TASK-ANCHORED SYSTEM PROMPTS

We use the following template for task-anchored system prompting. Table 7 lists the task-specific part of the system prompt that we use for each task category.

```
{Task-Anchored System Prompt}

Each response should start with "# Response X:" where X is
a number from 1 to {num_responses} to demarcate where each
different response begins.  Make sure to stop at # Response
{num_responses}.  Make each generated response self-contained.
They should not rely on the other responses for context.
```

### GENERAL IN-CONTEXT REGENERATION PROMPT

Zhang et al. (2025b) use the following prompt for in-context regeneration. We call this a "general" prompt because there is no task dependence.

```
Can you generate a different response?
```

### TASK-ANCHORED IN-CONTEXT REGENERATION PROMPTS

We use the following template for task-anchored in-context regeneration. Table 7 lists the task-specific part of the prompt that we use for each task category.

```
{Task-Anchored In-Context Regeneration Prompt}

Do not include any starting phrases or reasons for why your new
response is different.  Your response should be self-contained,
as if the prompt was the first thing that I asked.

Remember, the prompt is:  {prompt}
```

Table 7: Prompts for Task-Anchored Sampling Strategies

| Category | Task-Anchored System Prompt | Task-Anchored In-Context Regeneration Prompt |
| --- | --- | --- |
| Well-Specified Objective | The following prompt has a single correct answer. Generate {num_responses} responses. If relevant, slight variation in wording is allowed but the answer should remain the same. | Can you generate a different response? The prompt has a single correct answer, so your answer should remain the same. If relevant, slight variation in wording is allowed. |
| Underspecified Objective | The following prompt is underspecified and has many correct answers. Generate {num_responses} responses, each with a different correct answer. | Can you generate a different correct answer? The prompt is underspecified and has many correct answers. |
| Random Generation | The following prompt is asking you to randomize over a set of finite options. Generate {num_responses} responses, each with a different pseudo-random option. | Can you generate a different pseudo-random response? The prompt is asking you to randomize over a set of finite options. |
| Problem-Solving Objective | The following problem has a single correct answer, but can be solved using different problem-solving strategies. Generate {num_responses} different solutions, each with a different problem-solving strategy. | Can you solve the problem using a different strategy? The problem has a single correct answer, but can be solved using different problem-solving strategies. |
| Problem-Solving Subjective | The following problem has multiple correct answers, and may be solved using different problem-solving strategies. Generate {num_responses} different solutions, each with a different answer or problem-solving strategy. | Can you solve the problem using a different strategy? The problem has multiple correct answers, and may be solved using different problem-solving strategies. |
| Encyclopedia Inquiry | The following prompt is asking for information about the real-world, where there may be different factual perspectives. Your response must be grounded in credible references though references do not need to be mentioned. Generate {num_responses} responses that reflect different perspectives. | Can you generate a new response that reflects a different factual perspective? The prompt is asking for information about the real-world, where there may be different factual perspectives. Your response must be grounded in credible references though references do not need to be mentioned. |
| Creative Writing | The following prompt is asking for creative expression, so there are many possible subjective responses. Generate {num_responses} unique responses by varying the key creative elements such as tone, genre, point of view, theme, structure, etc. Each response should have different creative elements and reflect a distinct creative expression. | Can you generate a new response with different creative elements? The prompt is asking for creative expression, so there are many possible subjective responses. Your new response should change the key creative elements such as tone, genre, point of view, theme, structure, etc. |
| Advice or Opinions | The following prompt is asking for advice or opinions, so there are many possible subjective responses. Generate {num_responses} unique responses where each response expresses a different viewpoint or perspective. | Can you generate a new response with a different viewpoint or perspective? The prompt is asking for advice or opinions, so there are many possible subjective responses. |

## A.5 MEASURING FUNCTIONAL DIVERSITY

We use the following prompt template to evaluate functional diversity using LLM-judges. Table 9 lists the task-specific parts of the prompt that we use for each task category.

We validate the LLM-judges on a stratified random sample of 225 response pairs across models, tasks, and sampling strategies. Two authors independently labeled these prompts for functional diversity. Table 8 reports the inter-rater agreement between annotators and each LLM-judge. When compared to the ensemble of LLM-judges, annotator 1 achieved 77.7% agreement and annotator 2 achieved 79.0% agreement. This is comparable to Zhang et al. (2025b)'s evaluation of functional diversity in creative tasks, where a fine-tuned judge achieved 79% agreement with human annotators.

```
For the given prompt and two responses, determine if the
responses are functionally equivalent.  Functional equivalence
means a user who has seen one response would find the other
response to be redundant.

{Task-Anchored Functional Diversity Definition}

###
Prompt:  {prompt}
Response 1:  {response 1}
Response 2:  {response 2}
###

Are the responses functionally equivalent?
{Task-Anchored Diversity Judge Options}
Only output YES or NO.
```

To compare with functional diversity, we also measure diversity using three general diversity metrics that do not have task-dependence. For embedding diversity, we generate response embeddings using the gemini-embedding-001 model (with 3072-dimensional embeddings). For compression diversity, we use gZip to compress the concatenated text of response pairs Shaib et al. (2024).

**Definition A.1** (Vocabulary Diversity). Given two responses $y_a, y_b \in \mathcal{Y}$, let $\mathcal{V}_a$ and $\mathcal{V}_b$ denote the sets of unique words in $y_a$ and $y_b$, respectively. The *vocabulary diversity* between $y_a$ and $y_b$ is

$$d_{\text{vocab}}(y_a, y_b) := 1 - \frac{|\mathcal{V}_a \cap \mathcal{V}_b|}{|\mathcal{V}_a \cup \mathcal{V}_b|},$$

where $|\mathcal{V}_a \cap \mathcal{V}_b|$ is the number of shared words and $|\mathcal{V}_a \cup \mathcal{V}_b|$ is the total number of unique words in both responses.

**Definition A.2** (Embedding Diversity). Given two responses $y_a, y_b \in \mathcal{Y}$, let $e(y)$ denote the embedding vector for response $y$. The *embedding diversity* between $y_a$ and $y_b$ is

$$d_{\text{embed}}(y_a, y_b) := 1 - \cos(e(y_a), e(y_b)),$$

where $\cos(e(y_a), e(y_b))$ is the cosine similarity between the embedding vectors of $y_a$ and $y_b$.

**Definition A.3** (Compression Diversity). Given two responses $y_a, y_b \in \mathcal{Y}$, the *compression diversity* between $y_a$ and $y_b$ is

$$d_{\text{compression}}(y_a, y_b) := \frac{\text{compressed size of } y_a || y_b}{\text{size of } y_a || y_b},$$

where $y_a || y_b$ is the concatenated text of responses $y_a$ and $y_b$.

Table 8: Annotator Agreement

| | Annotator 1 | Annotator 2 | GPT-4o | Claude-4-Sonnet | Gemini-2.5-Flash |
|---|---|---|---|---|---|
| Annotator 1 | - | 79.9% | 75.0% | 77.2% | 77.2% |
| Annotator 2 | 79.9% | - | 79.0% | 77.7% | 80.4% |
| GPT-4o | 75.0% | 79.0% | - | 90.6% | 88.8% |
| Claude-4-Sonnet | 77.2% | 77.7% | 90.6% | - | 93.8% |
| Gemini-2.5-Flash | 77.2% | 80.4% | 88.8% | 93.8% | - |

Table 9: Prompts for Functional Diversity LLM-Judge

| Category | Task-Anchored Functional Diversity Def. | Diversity Judge Options |
|---|---|---|
| Well-Specified Objective | The prompt has a single correct answer. Responses are functionally equivalent if they represent the same answer. | Output YES if the responses represent the same answer. Output NO if the responses represent different answers. |
| Underspecified Objective | The prompt is underspecified and has many correct answers. Responses are functionally equivalent if they represent the same answer. | Output YES if the responses represent the same answer. Output NO if the responses represent different answers. |
| Random Generation | The prompt is asking for a random response over a set of finite options. Responses are functionally equivalent if they represent the same pseudo-random option. | Output YES if the responses represent the same pseudo-random option. Output NO if the responses represent different pseudo-random options. |
| Problem-Solving Objective | The prompt involves solving a problem with a single correct answer, but it can be solved using different problem-solving strategies. Responses are functionally equivalent if they represent the same problem-solving strategy. | Output YES if the responses represent the same problem-solving strategy. Output NO if the responses represent different problem-solving strategies. |
| Problem-Solving Subjective | The prompt involves solving a problem with multiple correct answers, and may be solved using different problem-solving strategies. Responses are functionally equivalent if they represent the same answer and problem-solving strategy. | Output YES if the responses represent the same answer and problem-solving strategy. Output NO if the responses represent different answers or problem-solving strategies. |
| Encyclopedia Inquiry | The prompt is asking for information about the real-world, where there may be different factual perspectives. Responses are functionally equivalent if they represent similar factual perspectives. | Output YES if the responses represent similar perspectives. Output NO if the responses represent different perspectives. |
| Creative Writing | The prompt is asking for creative expression where there are many possible subjective responses. Responses are functionally equivalent if the key creative elements (such as tone, genre, point of view, theme, structure, etc.) are the same. | Output YES if the responses have similar key creative elements. Output NO if the responses have different key creative elements. |
| Advice or Opinions | The prompt is asking for advice or opinions. Responses are functionally equivalent if they express the same viewpoint or perspective, even if they are worded differently. | Output YES if the responses have similar perspectives. Output NO if the responses have different perspectives. |

Table 10: Examples of Functionally Diverse Responses by Category

| Category | Prompt | Response 1 | Response 2 |
|---|---|---|---|
| Well-Specified Objective | What is the name of the project that won the 2014 ACM Software System Award? | The project that won the **2014 ACM Software System Award** is **Apache Hadoop**. Apache Hadoop is an open-source framework for distributed storage and processing of large datasets, which has had a significant impact on big data processing and analytics. | The project that won the 2014 ACM Software System Award is **The Eclipse Project**. |
| Underspecified Objective | Name a Harry Potter Book. | Harry Potter and the Sorcerer's Stone | Harry Potter and the Half-Blood Prince |
| Random Generation | Pick a number between 1 and 100. Now tell me what you picked. | I picked the number 42. | I picked the number 87. |
| Problem-Solving Objective | If $2^8 = 4^x$, what is the value of $x$? | We know that $4^x = (2^2)^x$. Using the property of exponents $(a^m)^n = a^{m \cdot n}$, this becomes: $$4^x = 2^{2x}.$$ The original equation $2^8 = 4^x$ now becomes: $$2^8 = 2^{2x}.$$ Since the bases are the same, we equate the exponents: $$8 = 2x.$$ Solve for $x$ by dividing both sides by 2: $$x = 4.$$ | We start with the equation $2^8 = 4^x$. Take the natural logarithm (ln) of both sides: $$\ln(2^8) = \ln(4^x).$$ Using the logarithmic rule $\ln(a^b) = b \cdot \ln(a)$, this becomes: $$8 \cdot \ln(2) = x \cdot \ln(4).$$ Express $\ln(4)$ as $\ln(2^2)$, which simplifies to $2 \cdot \ln(2)$. Substituting this back: $$8 \cdot \ln(2) = x \cdot (2 \cdot \ln(2)).$$ Cancel $\ln(2)$ from both sides: $$8 = 2x.$$ Solve for $x$: $$x = 4.$$ |
| Problem-Solving Subjective | You need to tighten a loose door knob but you don't have a screwdriver. The available items are a round-tipped hairpin, a brittle plastic fork, a flimsy nail file, a pair of rusted pliers, a thin rubber band, and a butter knife. The door knob screws are not the standard slot or Phillips head, but are special star head screws which cannot be tightened using a butter knife or hairpin due to the unusual shape and the risk of stripping the screw head. How can you tighten the door knob? | You can use the pair of rusted pliers as a workaround tool. Although they are rusted, they might still have enough grip to grab onto the edges of the star head screw. Position the pliers carefully around the head of the screw and twist it in the tightening direction. Apply steady pressure and try to avoid slipping. Clean the rust off the pliers beforehand, if possible, to improve their grip. | Since none of the tools are directly suitable for a star head screw, you can try securing the loose door knob in place temporarily. Use the thin rubber band to wrap tightly around the door knob base, compressing it against the door. This won't tighten the screws but can stabilize the knob until you acquire the correct star head screwdriver for a permanent fix. |
| Encyclopedia Inquiry | Did any major countries reinstate the death penalty after prohibiting it? | While it is not common, some nations have reinstated the death penalty after abolishing it. For example, Pakistan imposed a moratorium on executions in 2008, effectively halting the use of the death penalty. However, in 2014, the country lifted the moratorium following a devastating terrorist attack, and executions resumed. This demonstrates how security concerns can influence such decisions. | Historically, a few countries have reinstated the death penalty after initially banning or suspending it. The United States offers an example at the state level, where capital punishment was temporarily halted in the 1970s after the Supreme Court's Furman v. Georgia decision but reinstated later following the Gregg v. Georgia ruling in 1976. Nationally, reinstatements are usually tied to shifting societal attitudes or judicial rulings. |
| Creative Writing | Write a short love poem with 4 lines. | Your laughter blooms like spring's embrace, A gentle warmth I cannot replace. In your eyes, a world so true, Forever, my heart beats for you. | The ocean hums your name to me, Its waves reflect eternity. My soul is bound, your love's my art, Forever etched within my heart. |
| Advice or Opinions | I am naming my Indian son what one name should I name him? I get anxiety when given multiple options so please only give me one name. | Consider naming your son "Kabir." This name is associated with wisdom and spirituality, inspired by the renowned Indian poet and saint. | You might like the name "Rohan." It is simple, easy to pronounce globally, and represents growth and ascent. |

Table 11: Examples of Homogeneous Responses by Category

| Category | Prompt | Response 1 | Response 2 |
|---|---|---|---|
| Well-Specified Objective | What is the name of the project that won the 2014 ACM Software System Award? | The project that won the **2014 ACM Software System Award** is **The Coq Proof Assistant**. Coq is a formal proof management system, providing a formal language to write mathematical definitions, executable algorithms, and theorems, together with tools to verify the correctness of these proofs. | The project that won the 2014 ACM Software System Award is **Coq**, a formal proof management system. Coq is widely used in the field of computer science for formalizing mathematical proofs and for developing software with guarantees of correctness. |
| Underspecified Objective | Name a Harry Potter Book. | Harry Potter and the Sorcerer's Stone | Harry Potter and the Sorcerer's Stone (or Harry Potter and the Philosopher's Stone in the UK). |
| Random Generation | Pick a number between 1 and 100. Now tell me what you picked. | Sure! I'll pick the number **42**. | Sure! I pick 42. |
| Problem-Solving Objective | If $2^8 = 4^x$, what is the value of $x$? | We know that $4^x = (2^2)^x$. Using the property of exponents $(a^m)^n = a^{m \cdot n}$, this becomes: $$4^x = 2^{2x}.$$ The original equation $2^8 = 4^x$ now becomes: $$2^8 = 2^{2x}.$$ Since the bases are the same, we equate the exponents: $$8 = 2x.$$ Solve for $x$ by dividing both sides by 2: $$x = 4.$$ | Step 1: Express $4^x$ in terms of base 2 Since $4 = 2^2$, we can rewrite $4^x$ as: $$4^x = (2^2)^x$$ Using the property of exponents $(a^m)^n = a^{m \cdot n}$, this becomes: $$4^x = 2^{2x}$$ Step 2: Substitute into the equation The equation now becomes: $$2^8 = 2^{2x}$$ Step 3: Equating exponents Since the bases are the same (2), we can equate the exponents: $$8 = 2x$$ Step 4: Solve for $x$ Divide both sides of the equation by 2: $$x = \frac{8}{2} = 4$$ Final Answer: $$\boxed{4}$$ |
| Problem-Solving Subjective | You need to tighten a loose door knob but you don't have a screwdriver. The available items are a round-tipped hairpin, a brittle plastic fork, a flimsy nail file, a pair of rusted pliers, a thin rubber band, and a butter knife. The door knob screws are not the standard slot or Phillips head, but are special star head screws which cannot be tightened using a butter knife or hairpin due to the unusual shape and the risk of stripping the screw head. How can you tighten the door knob? | Wrap the thin rubber band around the head of the screw to improve grip. Use the tips of the rusted pliers to carefully turn the screw, pressing firmly to ensure the pliers engage with the rubber band and the screw head without slipping. Tighten the screw gradually while maintaining steady pressure. | Use the rusted pliers to grip the edges of the star head screw. Apply steady pressure and carefully turn the pliers to tighten the screw. If additional grip is needed, place the thin rubber band between the screw head and the pliers to increase friction and prevent slipping. |
| Encyclopedia Inquiry | Did any major countries reinstate the death penalty after prohibiting it? | The Philippines reinstated the death penalty in 1993 after having abolished it in 1987, before abolishing it again in 2006. Pakistan reinstated executions in 2014 after a six-year moratorium, primarily in response to terrorism. Sri Lanka has made moves toward resuming executions for drug-related offenses despite not carrying any out since 1976. In the United States, individual states like New York have reinstated the death penalty after previously prohibiting it, though some of these efforts were later reversed or deemed unconstitutional. | The Philippines reinstated the death penalty in 1993 after abolishing it in 1987, but it was abolished again in 2006. Pakistan lifted a six-year moratorium on the death penalty in 2014 and resumed executions, citing security concerns. In the United States, some states reinstated the death penalty after earlier bans, such as New York in the 1990s, though in some cases, these reinstatements were later overturned. Sri Lanka has expressed intentions to resume executions despite not carrying out any since 1976. |
| Creative Writing | Write a short love poem with 4 lines. | Beneath the stars, your gaze feels near, A whispered promise, soft and clear. In your embrace, my heart takes flight, You are my day, my moon, my night. | Beneath the stars, your eyes take flight, A gentle glow, my endless light. With every breath, my heart you keep, A love eternal, vast and deep. |
| Advice or Opinions | I am naming my Indian son what one name should I name him? I get anxiety when given multiple options so please only give me one name. | Sure! A beautiful and meaningful Indian name for your son could be **Arjun**. It is rooted in Indian heritage, symbolizes bravery and focus, and has a timeless charm. | How about the name **Arjun**? It is a strong, classic Indian name with roots in the Mahabharata, symbolizing bravery, focus, and virtue. |

## A.6 MEASURING CHECKLIST-BASED QUALITY

We follow the approach in Lin et al. (2025) and Wei et al. (2025) to evaluate quality using LLM-judges with task-specific grading checklists. First, the LLM-judge generates a grading checklist for each prompt with key factors that should be present in high-quality responses (Checklist Creation Prompt). Two authors independently verified that these checklists were reasonable for each prompt. Then, the LLM-judge is given a response to grade along with the previously generated checklist (Checklist-Based Quality Judge Prompt). We grade responses on a Likert scale from 1 to 5, where 1 indicates the response is very poor and does not meet any criteria on the checklist, whereas 5 indicates the response is very good and meets all the criteria on the checklist. Table 12 includes examples of the grading checklists generated by LLM-judges.

CHECKLIST CREATION PROMPT

```
You are an expert evaluator of LLM response quality.  Your task
is to generate a quality checklist that can be used to evaluate
response quality for a given user prompt.

# Instructions
First, identify 3 to 5 key factors that should be present in
high-quality responses for the given prompt.  Since a prompt
may have many different high-quality responses, the key factors
should be broad enough to cover varied high-quality responses
while still being specific to the prompt.

Then, turn each key factor into a yes/no question for a
checklist to evaluate response quality.  Your questions should
be concise and specific to the user prompt.  Avoid creating
duplicate, cumbersome, or vague questions.  Do not create more
than 5 questions.

# User Prompt
{prompt}

# Output Format
First, output the key factors you identified in a short
paragraph.  Then, output your quality checklist of yes/no
questions in the following format, starting with "Here is my
quality checklist:".  Separate each question with newlines.  Do
not output any other text after the checklist.

Here is my quality checklist:
1.  Question 1
2.  Question 2
...
```

CHECKLIST-BASED QUALITY JUDGE PROMPT

```
You are an expert evaluator of LLM response quality.  Your task
is to grade the given LLM response to a user prompt based on
the quality checklist below.

# User Prompt
{prompt}

# Response To Grade
{response}

# Quality Checklist
{checklist}

# Instructions
First, answer each question in the checklist with "yes" or
"no", based on the response to grade.

Then, assign a quality score between 1 to 5 for the response,
based on the following criteria:
- Score 1:  The response is very poor and does not meet any
criteria on the checklist.
- Score 2:  The response is poor and meets few criteria on the
checklist.
- Score 3:  The response is fair and meets some criteria on the
checklist.
- Score 4:  The response is good and meets most criteria on the
checklist.
- Score 5:  The response is very good and meets all criteria on
the checklist.

# Output Format
First, output your answers to the quality checklist as a
numbered list.  Each answer should start with "yes" or "no"
and have a short justification (1-2 sentences).

Then, output your quality score for the response.  Your output
should end with the following sentence, where <score> is the
quality score you assigned:  "Based on the quality checklist,
my quality score for the response is <score>."
```

Table 12: Examples of Task-Specific Grading Checklists

| LLM-Judge | Prompt | Grading Checklist Generated by LLM-Judge |
|---|---|---|
| GPT-4o | Tell me a story in five sentences about a boy and his dog. | 1. Does the response present a clear and cohesive narrative within five sentences? 2. Are the boy and his dog central to the story, with a clear focus on their interaction or relationship? 3. Does the story elicit some form of engagement (e.g., emotional resonance, curiosity, or intrigue)? 4. Does the story conclude in a satisfying or meaningful way within the constraints of five sentences? 5. Is the story creative and free from overused cliches? |
| Claude-4-Sonnet | Tell me a dad joke. | 1. Does the response contain an actual joke? 2. Is the joke appropriate for all ages and family-friendly? 3. Does the joke follow typical dad joke characteristics (puns, wordplay, or simple humor)? 4. Is the response concise and focused on delivering the joke? |
| Gemini-2.5-Flash | Write a short love poem with 4 lines. | 1. Is the response structured as a poem? 2. Does the poem express a theme of love? 3. Is the poem exactly four lines long? |
| GPT-4o | Why is Issac Newton famous? | 1. Does the response mention at least one of Isaac Newton's major contributions, such as the laws of motion, the law of universal gravitation, or calculus? 2. Is the information provided accurate and free from factual errors? 3. Does the response maintain relevance to the prompt, focusing on why Isaac Newton is famous? 4. Is the response clear and easy to understand by the intended audience? |
| Claude-4-Sonnet | If $2^8 = 4^x$, what is the value of $x$? | 1. Does the response correctly rewrite the equation using the same base (either base 2 or base 4)? 2. Does the response properly apply exponent rules to solve for x? 3. Does the response show clear, logical steps that lead to the solution? 4. Does the response arrive at the correct answer (x = 4)? 5. Does the response verify the solution by checking it against the original equation? |
| Gemini-2.5-Flash | Who was the first scientist to isolate cardiolipin? | 1. Does the response identify the first scientist to isolate cardiolipin? 2. Does the response accurately name the scientist as M. C. Pangborn? 3. Is the information provided factually correct? 4. Is the response concise and directly answer the question? |
| GPT-4o | I am going to join a software company next week as a software engineer. What are the things that I should take care of before joining? | 1. Does the response provide actionable steps to prepare technically or brush up on relevant skills for the role? 2. Does the response address logistical preparation, such as ensuring necessary documentation or setup is in place? 3. Does the response offer advice on professional behavior or mindset to adopt for a successful onboarding experience? 4. Are the suggestions relevant for a software engineer and tailored to the context of joining a new company? 5. Is the advice practical and feasible for someone to implement before their first day? |
| Claude-4-Sonnet | Pick a number between 1 and 100. Now tell me what you picked. | 1. Does the response include a specific number between 1 and 100 (inclusive)? 2. Does the response clearly state what number was picked? 3. Does the response directly address the user's request without unnecessary complexity or evasion? |
| Gemini-2.5-Flash | Write an apology letter to mother and father that I had run away in fear from getting beaten up by my friend. | 1. Does the letter sincerely apologize for running away? 2. Does the letter clearly state that the reason for running away was fear of being beaten by a friend? 3. Does the letter acknowledge the parents' potential worry or hurt caused by the action? 4. Does the letter offer any reassurance to the parents about the future or the child's safety? |

Table 13: Examples of "Very Good" Responses Under the Checklist-Based Grading Approach With Varying Athene-RM 8B Reward

| Athene Reward | Prompt/Response |
|---|---|
| | Prompt (Category A): What species of Polymixia fish is named after research scientist Gloria Hollister? |
| 0.82 | The species of Polymixia fish named after research scientist Gloria Hollister is *Polymixia hollisterae*. |
| 0.63 | The fish species *Polymixia hollisterae* commemorates the contributions of Gloria Hollister to science. |
| | Prompt (Category B): List a capital city in Africa. |
| -0.20 | Cairo is the capital city of Egypt. |
| 0.12 | Addis Ababa is the capital city of Ethiopia. |
| | Prompt (Category C): Pick a number between 1 and 100. Now tell me what you picked. |
| 0.35 | I picked the number 42. |
| -0.25 | I picked the number 6. |
| | Prompt (Category D): How many positive whole-number divisors does 196 have? |
| 1.07 | First, perform the prime factorization of 196. Dividing 196 by 2 repeatedly gives $2^2$, and dividing the remaining quotient (49) results in $7^2$. Thus, $196 = 2^2 \times 7^2$. To find the number of divisors, use the formula $(e_1 + 1)(e_2 + 1)$, where $e_1$ and $e_2$ are the exponents of the prime factors. Here, $(2+1)(2+1) = 9$. Hence, 196 has **9 divisors**. |
| 0.34 | Recognize that 196 is a perfect square: $14 \times 14$. Break 14 into its prime factors: $14 = 2 \times 7$, so $196 = (2 \times 7) \times (2 \times 7) = 2^2 \times 7^2$.
Using the rule for finding divisors $((e_1 + 1)(e_2 + 1)...)$, where $e_1, e_2$ are the exponents in the prime factorization: $(2+1)(2+1) = 3 \times 3 = 9$.
Thus, 196 has **9 positive whole-number divisors**. |
| | Prompt (Category E): Your necklace has tangled into a very tight knot, and the hotel does not provide any tools for untangling. Available tools include a shampoo, a fork, a safety pin, a pen, a hair brush, a toothpick, a hair dryer, a credit card and hand cream. Constraint: The necklace is made of very fine and delicate chain, so using a safety pin or credit card may damage it. How do you untangle it using just these items? |
| -0.14 | Rub a tiny amount of hand cream onto the knot to reduce friction between the links. Place the chain flat on a table, holding it steady. Use the prongs of the fork to carefully tease the loops apart, starting with the loosest part of the knot. Work with patience, rotating the fork as needed to avoid pulling too hard on the chain. If the knot is still tight, use the pen to hold one side steady while you maneuver the fork through the loops for additional leverage. |
| -0.47 | Start by applying a small amount of shampoo or hand cream to the knot. This will act as a lubricant, making the chain slippery and easier to untangle. Next, use the toothpick to gently prod at the knot. Work carefully to loosen the knot without pulling or using too much force, as the chain is delicate. If the knot remains stubborn, gently warm the chain using the hair dryer on a low setting to make the metal more flexible, then repeat the process with the toothpick until the knot comes undone. |
| | Prompt (Category F): Can you list 3 influential persons from the nineteenth century? |
| 0.11 | 1. Charles Darwin – Naturalist and biologist known for his theory of evolution by natural selection. 2. Queen Victoria – Monarch of the United Kingdom who oversaw the Victorian Era. 3. Karl Marx – Philosopher, economist, and revolutionary socialist who co-wrote "The Communist Manifesto." |
| 0.25 | 1. Abraham Lincoln – 16th President of the United States who led the country during the Civil War and abolished slavery. 2. Florence Nightingale – Founder of modern nursing and a key figure in improving healthcare. 3. Leo Tolstoy – Russian novelist famous for works like "War and Peace" and "Anna Karenina." |
| | Prompt (Category G): Tell me a riddle. |
| 0.07 | The more of me you take, the more you leave behind. What am I? **Answer:** Footsteps. |
| 0.92 | What has cities, but no houses; forests, but no trees; and rivers, but no water? **Answer:** A map. |
| | Prompt (Category H): Suggest a birthday gift for a friend's 25th birthday. Just one. |
| 0.13 | A cooking class voucher or experience for two. This can be a fun way for them to learn something new and enjoy a memorable experience, either solo or with a loved one. |
| 0.53 | A personalized star map that shows the alignment of the stars on the day they were born. It's a unique and sentimental way to celebrate their 25th birthday. |

## A.7 ALIGNMENT EXPERIMENTS

We run exploratory alignment experiments to further show the usefulness of our task-dependent framework for evaluating and reducing homogenization. All alignment experiments use *Llama-3.1-8B-Instruct* and the *Athene-RM-8B* reward model, and LLM-judge metrics are only calculated with GPT-4o. First, we demonstrate that functional diversity does not collapse and sometimes increases after preference tuning with Direct Preference Optimization (DPO) (Rafailov et al., 2023) and Group Relative Policy Optimization (GRPO) (Shao et al., 2024) (Figures 15-17). Second, we show how task-dependent evaluation clarifies the impact of diversity-promoting methods in alignment by evaluating DARLING (Diversity-Aware Reinforcement Learning) (Li et al., 2025b) (Figures 18-20). We further explore how DARLING could be modified to account for task-dependence.

**DPO and GRPO**   We run online DPO and GRPO following the training recipe for non-verifiable rewards in Lanchantin et al. (2025b) and Li et al. (2025b), respectively. In particular, we use a learning rate of $1e-6$, batch size of 32, and train for 1000 steps. At each step, we generate 8 responses per prompt with temperature 1.0 and 1024 max tokens. For DPO, preference pairs are constructed based on the responses with the maximum and minimum reward. For GRPO, all 8 responses are used to calculate advantage. We explore two values of $\beta$: 0.01 and 0.1 for DPO, and 0.001 and 0.01 for GRPO.

We also explore using two datasets for preference tuning: Wildchat (Zhao et al., 2024) and Ultrafeedback (Cui et al., 2023). For Wildchat, we use 10,000 randomly sampled prompts. The majority of these prompts (5,535) corresponded to category E (Creative Writing), based on task classification by GPT-4o. Thus, for Ultrafeedback, we try a stratified random sample of 10,000 prompts based on each task category in our taxonomy and GPT-4o as the task classification judge. Specifically, we sample 2,500 prompts for each task category, excluding category E (Problem-Solving Subjective) and combining categories B and C (Underspecified Objective and Random Generation) due to prompt availability. In both datasets, we exclude prompts with more than 512 tokens.

**DARLING**   Li et al. (2025b) propose DARLING (Diversity-Aware Reinforcement Learning), which modifies the GRPO reward to jointly reinforce diversity and quality. Specifically, they scale reward by the diversity $d(y_i|y_1, \ldots, y_n)$ of a generation $y_i$, which they define as the average pairwise distance between $y_i$ and all other generations $y_j$ ($j \neq i$), normalized to be between 0 and 1. They implement their method using "semantic uniqueness" as their distance metric, which represents a general notion of functional diversity (not task-dependent). They fine-tune a *ModernBERT-base* model to predict semantic uniqueness based on 1000 human annotations from NoveltyBench (Zhang et al., 2025b).

We explore modifying DARLING to account for task-dependence by using GPT-4o as a task-dependent functional diversity judge, in place of the fine-tuned ModernBert classifier. For prompts in category A (Well-Specified Objective), we also modify DARLING to scale the reward by $1 - d(y_i|y_1, \ldots, y_n)$, which promotes homogenization instead of diversity for category A.

## B  ADDITIONAL EXPERIMENT RESULTS

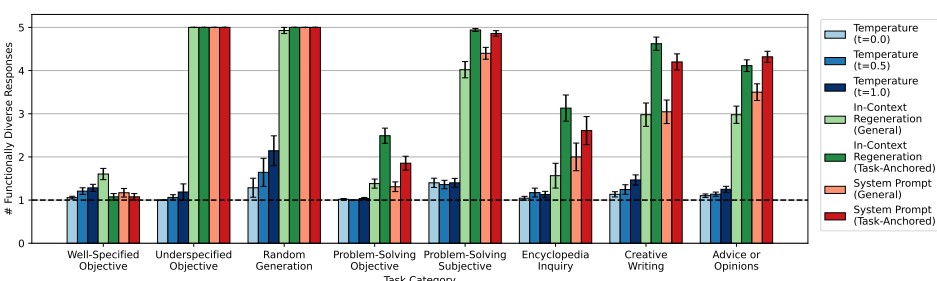

Figure 5: Number of functionally diverse responses generated by **Claude-4-Sonnet** for each sampling strategy and task category (c.f. Figure 2).

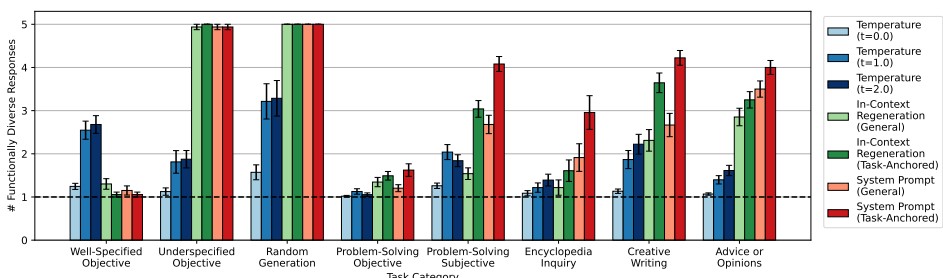

Figure 6: Number of functionally diverse responses generated by **Gemini-2.5-Flash** for each sampling strategy and task category (c.f. Figure 2).

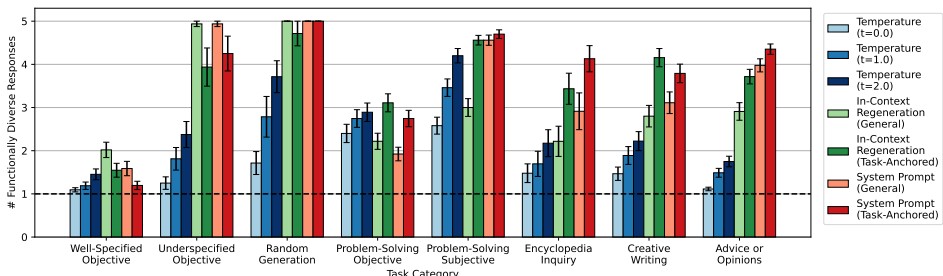

Figure 7: Number of functionally diverse responses generated by **Llama-3.1-8B-Instruct** (c.f. Figure 2).

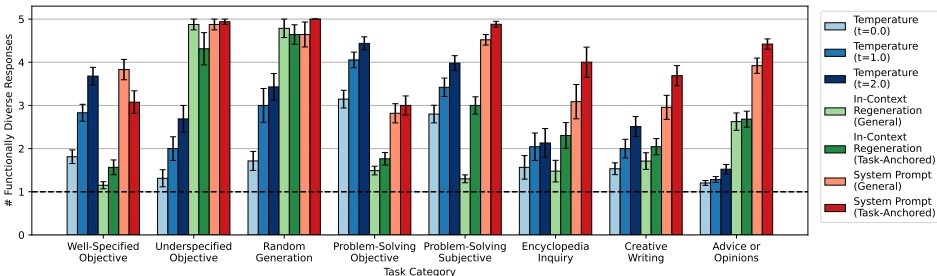

Figure 8: Number of functionally diverse responses generated by **Mistral-7B-Instruct-v0.3** for each sampling strategy and task category (c.f. Figure 2).

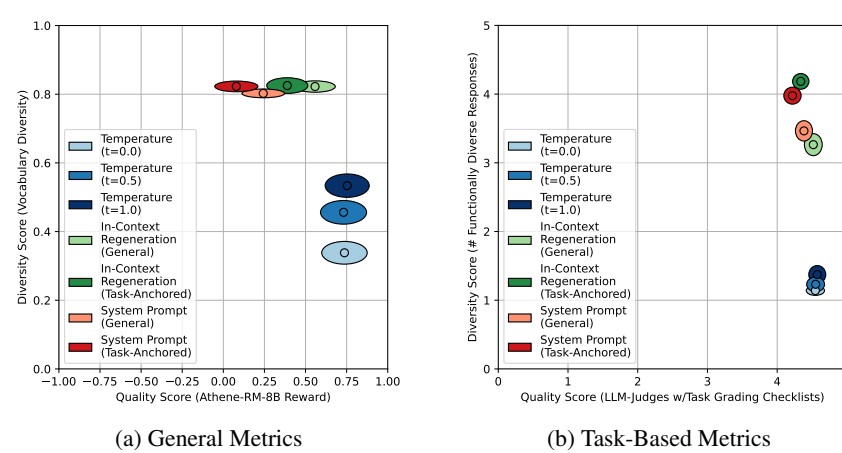

(a) General Metrics       (b) Task-Based Metrics

Figure 9: Diversity-quality tradeoff under general vs task-based metrics for **Claude-4-Sonnet**.

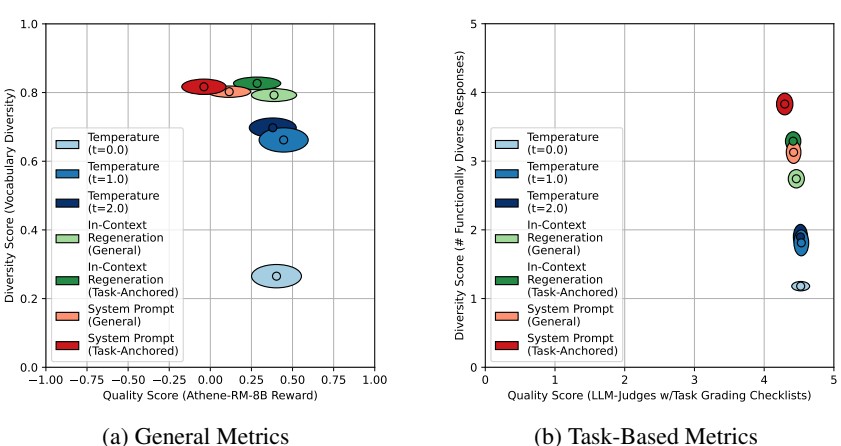

(a) General Metrics       (b) Task-Based Metrics

Figure 10: Diversity-quality tradeoff under general vs task-based metrics for **Gemini-2.5-Flash**.

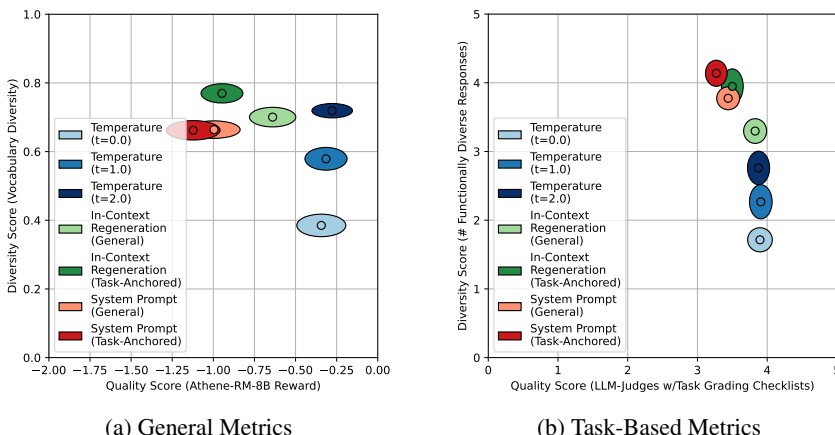

(a) General Metrics       (b) Task-Based Metrics

Figure 11: Diversity-quality tradeoff under general vs task-based metrics for **Llama-3.1-8B-Instruct**.

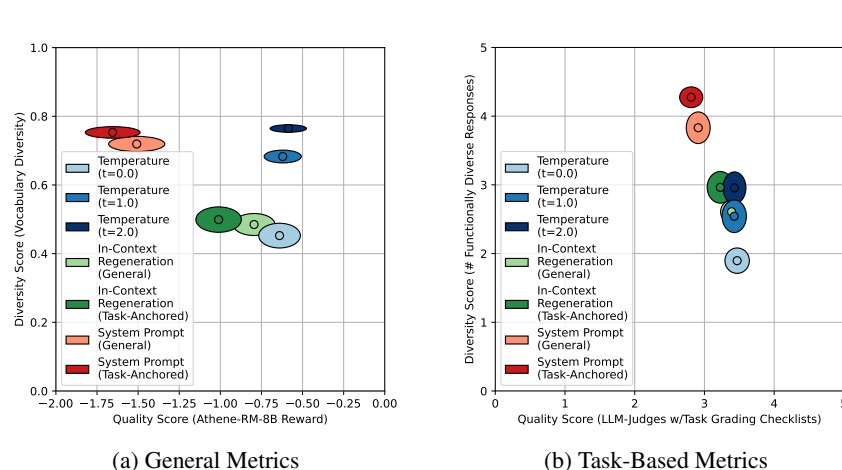

(a) General Metrics       (b) Task-Based Metrics

Figure 12: Diversity-quality tradeoff under general vs task-based metrics for **Mistral-7B-Instruct-v0.3**.

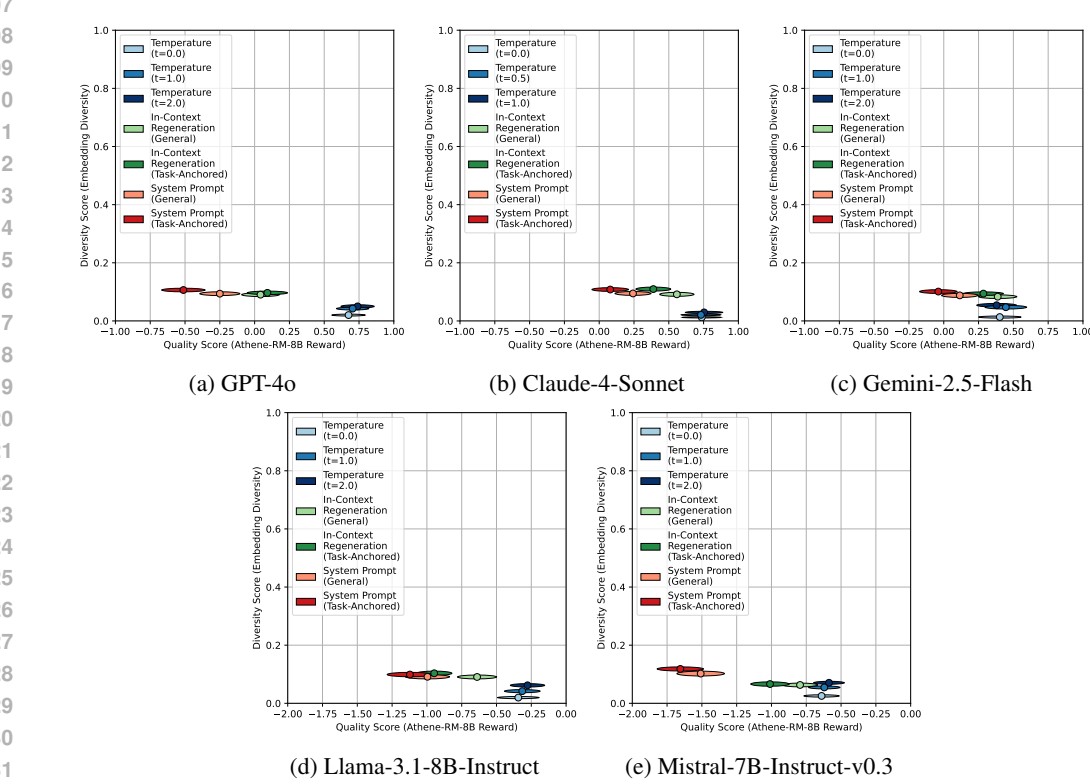

(a) GPT-4o     (b) Claude-4-Sonnet     (c) Gemini-2.5-Flash

(d) Llama-3.1-8B-Instruct     (e) Mistral-7B-Instruct-v0.3

Figure 13: Diversity-quality tradeoff using embedding diversity.

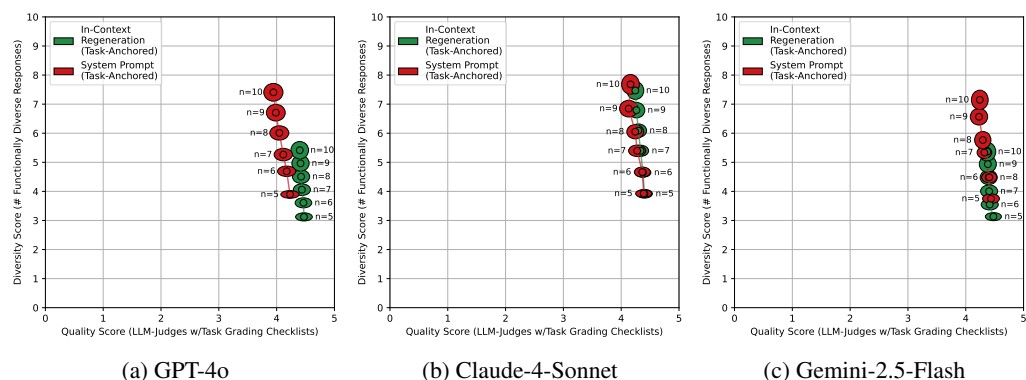

(a) GPT-4o  (b) Claude-4-Sonnet  (c) Gemini-2.5-Flash

Figure 14: Diversity-quality tradeoff for varying number of generated responses ($n = 5$ to $n = 10$). Judge metrics based on GPT-4o only. The number of functionally diverse responses consistently increases with more generated responses. However, there appear to be small (statistically insignificant) decreases in checklist-based quality. The quality decrease is larger for system prompt sampling, possibly due to $n = 10$ approaching the max output length for a single generation.

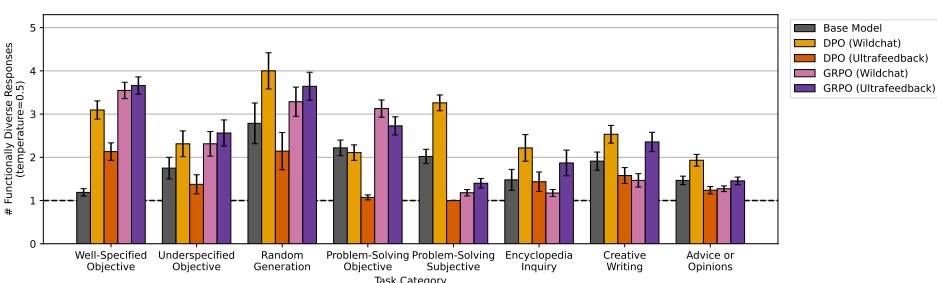

Figure 15: Number of functionally diverse responses generated by Llama-3.1-8B-Instruct, with and without preference alignment. DPO and GRPO results based on 1000 training steps and $\beta = 0.01$ and $\beta = 0.001$, respectively. Unlike prior results on token entropy (Lanchantin et al., 2025b), functional diversity does not collapse and sometimes increases post-alignment.

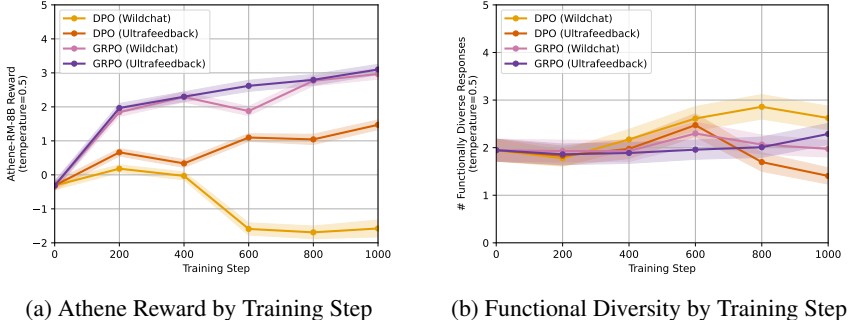

(a) Athene Reward by Training Step  (b) Functional Diversity by Training Step

Figure 16: During alignment of Llama-3.1-8B-Instruct, the reward generally increases without a collapse in functional diversity. DPO and GRPO results based on $\beta = 0.01$ and $\beta = 0.001$, respectively. Metrics avg. across all task categories except category A, where homogenization is desired.

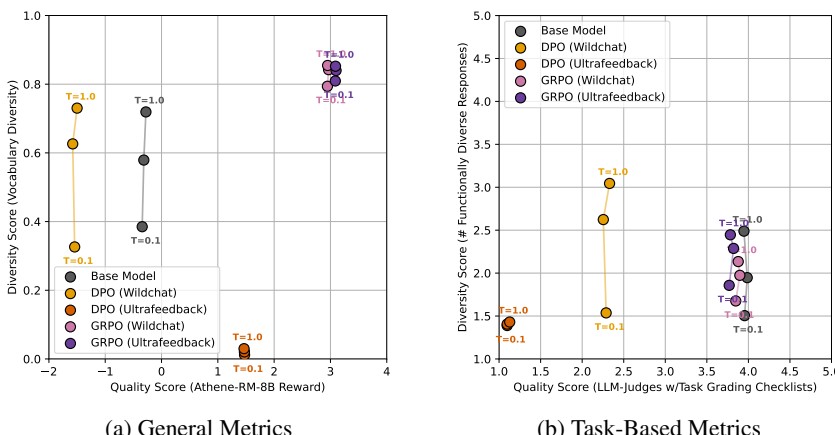

(a) General Metrics

(b) Task-Based Metrics

Figure 17: Diversity-quality tradeoff under general vs task-based metrics for Llama-3.1-8B-Instruct, with and without preference alignment. DPO and GRPO results based on 1000 training steps and $\beta = 0.01$ and $\beta = 0.001$, respectively. While DPO and GRPO generally improve reward quality, they do not always improve checklist-based quality. Metrics avg. across all task categories except category A, where homogenization is desired.

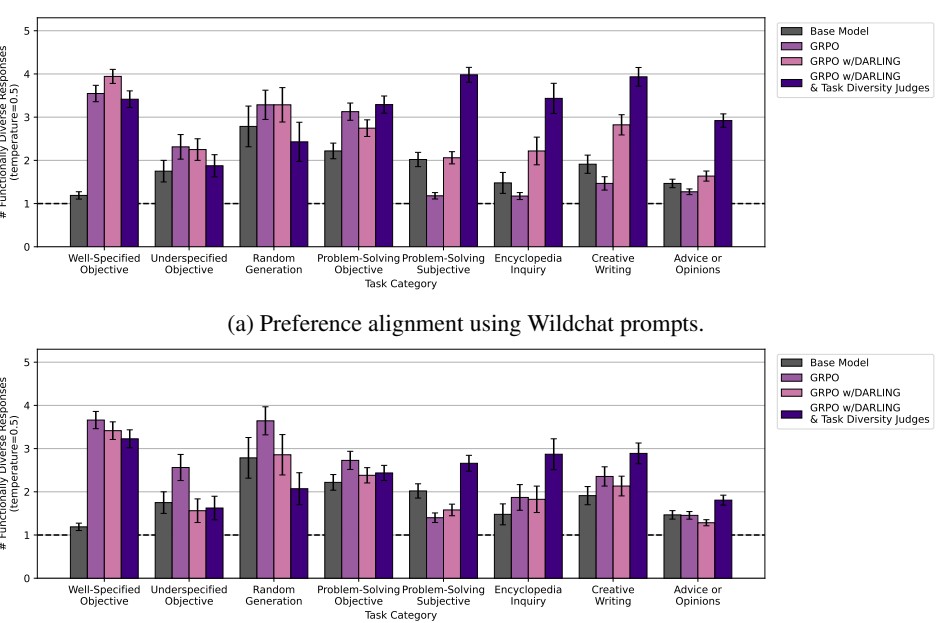

(a) Preference alignment using Wildchat prompts.

(b) Preference alignment using Ultrafeedback prompts.

Figure 18: Number of functionally diverse responses generated by Llama-3.1-8B-Instruct, after preference alignment with DARLING (Li et al., 2025b). GRPO and DARLING results with $\beta = 0.001$. DARLING with task diversity judges uses GPT-4o as the task-dependent functional diversity judge. DARLING generally maintains or improves functional diversity over GRPO, and task diversity judges generally provide further improvement. All alignment methods undesirably reduce homogenization for category A (Well-Specified Objective).

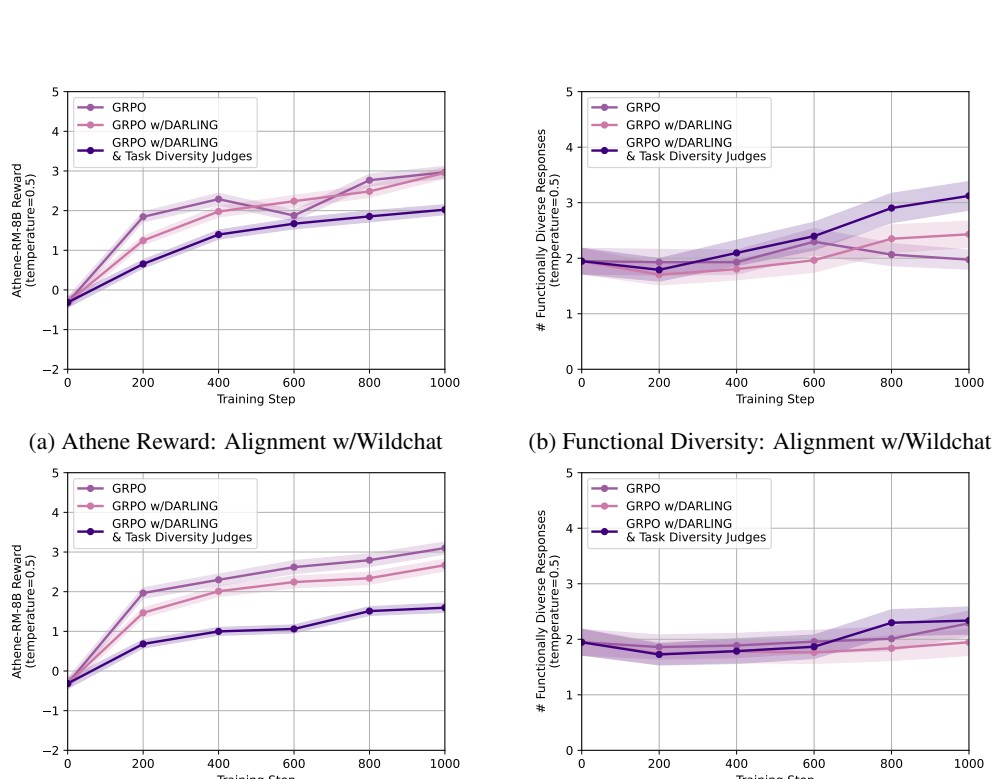

(a) Athene Reward: Alignment w/Wildchat

(b) Functional Diversity: Alignment w/Wildchat

(c) Athene Reward: Alignment w/Ultrafeedback

(d) Functional Diversity: Alignment w/Ultrafeedback

Figure 19: During alignment of Llama-3.1-8B-Instruct with DARLING using **Wildchat prompts**, both the reward and functional diversity generally increase. GRPO and DARLING use $\beta = 0.001$. Metrics avg. across all task categories except category A, where homogenization is desired.

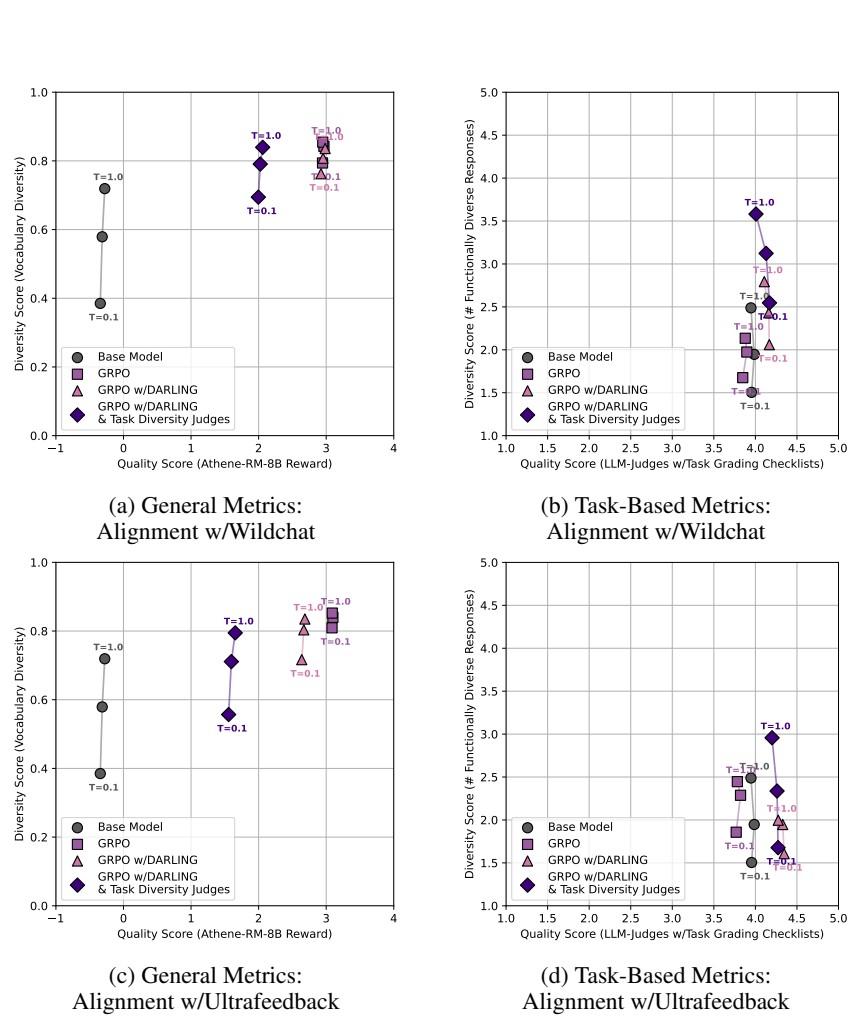

Figure 20: Diversity-quality tradeoff under general vs task-based metrics for Llama-3.1-8B-Instruct, after preference alignment with DARLING (Li et al., 2025b). GRPO and DARLING results based on 1000 training steps and $\beta = 0.001$. While general metrics do not show improvements, task-based metrics show that DARLING improves both diversity and quality compared to GRPO.

Table 14: # of Functionally Diverse Responses by Model, Sampling Strategy, and Task Category.

| Model | Sampling Strategy | A | B | C | D | E | F | G | H |
|-------|-------------------|---|---|---|---|---|---|---|---|
| gpt-4o | Temperature (t=0.0) | 1.57 (0.11) | 1.19 (0.10) | 1.86 (0.31) | 1.33 (0.09) | 1.22 (0.07) | 1.13 (0.10) | 1.18 (0.07) | 1.08 (0.03) |
| gpt-4o | Temperature (t=1.0) | 2.11 (0.19) | 1.50 (0.18) | 3.14 (0.46) | 1.27 (0.10) | 1.30 (0.10) | 1.13 (0.13) | 1.40 (0.13) | 1.19 (0.06) |
| gpt-4o | Temperature (t=2.0) | 2.38 (0.21) | 1.69 (0.25) | 3.14 (0.42) | 1.31 (0.10) | 1.32 (0.10) | 1.09 (0.09) | 1.89 (0.19) | 1.24 (0.06) |
| gpt-4o | In-Context Regeneration (General) | 2.19 (0.22) | 4.94 (0.06) | 5.00 (0.00) | 1.24 (0.10) | 1.84 (0.16) | 1.35 (0.24) | 2.44 (0.26) | 2.84 (0.20) |
| gpt-4o | In-Context Regeneration (Task-Anchored) | 1.06 (0.03) | 5.00 (0.00) | 5.00 (0.00) | 1.31 (0.07) | 2.82 (0.22) | 1.74 (0.28) | 3.31 (0.25) | 3.20 (0.20) |
| gpt-4o | System Prompt (General) | 1.94 (0.22) | 5.00 (0.00) | 5.00 (0.00) | 1.10 (0.06) | 3.80 (0.19) | 2.30 (0.39) | 2.96 (0.27) | 3.65 (0.19) |
| gpt-4o | System Prompt (Task-Anchored) | 1.00 (0.00) | 5.00 (0.00) | 5.00 (0.00) | 1.69 (0.13) | 4.60 (0.11) | 2.95 (0.37) | 4.13 (0.20) | 4.20 (0.15) |
| claude-4-sonnet | Temperature (t=0.0) | 1.06 (0.03) | 1.00 (0.00) | 1.29 (0.22) | 1.02 (0.02) | 1.40 (0.11) | 1.04 (0.04) | 1.13 (0.06) | 1.10 (0.04) |
| claude-4-sonnet | Temperature (t=0.5) | 1.21 (0.08) | 1.06 (0.06) | 1.64 (0.32) | 1.00 (0.00) | 1.36 (0.10) | 1.17 (0.10) | 1.24 (0.11) | 1.14 (0.05) |
| claude-4-sonnet | Temperature (t=1.0) | 1.28 (0.08) | 1.19 (0.19) | 2.14 (0.35) | 1.04 (0.03) | 1.40 (0.10) | 1.13 (0.07) | 1.47 (0.12) | 1.25 (0.07) |
| claude-4-sonnet | In-Context Regeneration (General) | 1.60 (0.13) | 5.00 (0.00) | 4.93 (0.07) | 1.38 (0.11) | 4.02 (0.19) | 1.57 (0.29) | 2.98 (0.27) | 2.98 (0.20) |
| claude-4-sonnet | In-Context Regeneration (Task-Anchored) | 1.08 (0.08) | 5.00 (0.00) | 5.00 (0.00) | 2.49 (0.18) | 4.94 (0.03) | 3.13 (0.30) | 4.62 (0.15) | 4.11 (0.14) |
| claude-4-sonnet | System Prompt (General) | 1.17 (0.10) | 5.00 (0.00) | 5.00 (0.00) | 1.31 (0.11) | 4.40 (0.14) | 2.00 (0.32) | 3.05 (0.27) | 3.50 (0.19) |
| claude-4-sonnet | System Prompt (Task-Anchored) | 1.08 (0.08) | 5.00 (0.00) | 5.00 (0.00) | 1.85 (0.16) | 4.86 (0.06) | 2.61 (0.33) | 4.20 (0.19) | 4.32 (0.13) |
| gemini-2.5-flash | Temperature (t=0.0) | 1.25 (0.07) | 1.12 (0.09) | 1.57 (0.17) | 1.02 (0.02) | 1.26 (0.06) | 1.09 (0.06) | 1.13 (0.05) | 1.07 (0.03) |
| gemini-2.5-flash | Temperature (t=1.0) | 2.55 (0.21) | 1.81 (0.26) | 3.21 (0.41) | 1.13 (0.06) | 2.04 (0.17) | 1.22 (0.11) | 1.87 (0.21) | 1.40 (0.10) |
| gemini-2.5-flash | Temperature (t=2.0) | 2.68 (0.20) | 1.88 (0.20) | 3.29 (0.41) | 1.05 (0.04) | 1.84 (0.14) | 1.39 (0.14) | 2.22 (0.23) | 1.61 (0.12) |
| gemini-2.5-flash | In-Context Regeneration (General) | 1.30 (0.12) | 4.94 (0.06) | 5.00 (0.00) | 1.35 (0.11) | 1.54 (0.13) | 1.22 (0.18) | 2.31 (0.25) | 2.85 (0.20) |
| gemini-2.5-flash | In-Context Regeneration (Task-Anchored) | 1.06 (0.06) | 5.00 (0.00) | 5.00 (0.00) | 1.49 (0.10) | 3.04 (0.19) | 1.61 (0.25) | 3.64 (0.23) | 3.25 (0.19) |
| gemini-2.5-flash | System Prompt (General) | 1.15 (0.11) | 4.94 (0.06) | 5.00 (0.00) | 1.20 (0.08) | 2.68 (0.21) | 1.91 (0.32) | 2.67 (0.27) | 3.50 (0.19) |
| gemini-2.5-flash | System Prompt (Task-Anchored) | 1.06 (0.06) | 4.94 (0.06) | 5.00 (0.00) | 1.62 (0.15) | 4.08 (0.17) | 2.96 (0.39) | 4.22 (0.17) | 4.00 (0.16) |

Table 15: # of Functionally Diverse Responses by Model, Sampling Strategy, and Task Category.

(Continued from Table 14)

| Model | Sampling Strategy | A | B | C | D | E | F | G | H |
|-------|-------------------|---|---|---|---|---|---|---|---|
| Llama-3.1-8B-Instruct | Temperature (t=0.0) | 1.09 (0.05) | 1.25 (0.14) | 1.71 (0.27) | 2.40 (0.21) | 2.58 (0.20) | 1.48 (0.22) | 1.47 (0.15) | 1.11 (0.04) |
| Llama-3.1-8B-Instruct | Temperature (t=0.5) | 1.19 (0.09) | 1.81 (0.26) | 2.79 (0.47) | 2.75 (0.20) | 3.46 (0.20) | 1.70 (0.29) | 1.89 (0.21) | 1.49 (0.10) |
| Llama-3.1-8B-Instruct | Temperature (t=1.0) | 1.45 (0.13) | 2.38 (0.30) | 3.71 (0.37) | 2.89 (0.21) | 4.20 (0.17) | 2.17 (0.31) | 2.22 (0.22) | 1.75 (0.12) |
| Llama-3.1-8B-Instruct | In-Context Regeneration (General) | 2.02 (0.18) | 4.94 (0.06) | 5.00 (0.00) | 2.22 (0.19) | 3.00 (0.21) | 2.22 (0.35) | 2.80 (0.25) | 2.91 (0.20) |
| Llama-3.1-8B-Instruct | In-Context Regeneration (Task-Anchored) | 1.55 (0.16) | 3.94 (0.44) | 4.71 (0.29) | 3.11 (0.21) | 4.56 (0.11) | 3.43 (0.36) | 4.16 (0.21) | 3.72 (0.17) |
| Llama-3.1-8B-Instruct | System Prompt (General) | 1.59 (0.17) | 4.94 (0.06) | 5.00 (0.00) | 1.92 (0.16) | 4.56 (0.12) | 2.91 (0.43) | 3.11 (0.25) | 3.98 (0.15) |
| Llama-3.1-8B-Instruct | System Prompt (Task-Anchored) | 1.20 (0.10) | 4.25 (0.40) | 5.00 (0.00) | 2.75 (0.19) | 4.70 (0.10) | 4.13 (0.30) | 3.79 (0.21) | 4.35 (0.12) |
| Mistral-7B-Instruct-v0.3 | Temperature (t=0.0) | 1.81 (0.16) | 1.31 (0.20) | 1.71 (0.22) | 3.15 (0.21) | 2.80 (0.20) | 1.57 (0.27) | 1.53 (0.14) | 1.20 (0.06) |
| Mistral-7B-Instruct-v0.3 | Temperature (t=0.5) | 2.83 (0.20) | 2.00 (0.27) | 3.00 (0.39) | 4.05 (0.18) | 3.42 (0.21) | 2.04 (0.32) | 2.00 (0.22) | 1.28 (0.07) |
| Mistral-7B-Instruct-v0.3 | Temperature (t=1.0) | 3.68 (0.20) | 2.69 (0.31) | 3.43 (0.31) | 4.44 (0.15) | 3.98 (0.17) | 2.13 (0.33) | 2.51 (0.23) | 1.52 (0.11) |
| Mistral-7B-Instruct-v0.3 | In-Context Regeneration (General) | 1.15 (0.08) | 4.88 (0.12) | 4.79 (0.21) | 1.49 (0.10) | 1.30 (0.09) | 1.48 (0.25) | 1.71 (0.19) | 2.62 (0.20) |
| Mistral-7B-Instruct-v0.3 | In-Context Regeneration (Task-Anchored) | 1.57 (0.17) | 4.31 (0.37) | 4.64 (0.23) | 1.76 (0.14) | 3.00 (0.20) | 2.30 (0.30) | 2.04 (0.19) | 2.68 (0.19) |
| Mistral-7B-Instruct-v0.3 | System Prompt (General) | 3.83 (0.24) | 4.88 (0.12) | 4.64 (0.29) | 2.82 (0.22) | 4.52 (0.12) | 3.09 (0.40) | 2.96 (0.28) | 3.92 (0.18) |
| Mistral-7B-Instruct-v0.3 | System Prompt (Task-Anchored) | 3.08 (0.26) | 4.94 (0.06) | 5.00 (0.00) | 3.00 (0.22) | 4.88 (0.07) | 4.00 (0.35) | 3.69 (0.23) | 4.42 (0.12) |

Table 16: Vocabulary Diversity by Model, Sampling Strategy, and Task Category.

| Model | Sampling Strategy | A | B | C | D | E | F | G | H |
|---|---|---|---|---|---|---|---|---|---|
| gpt-4o | Temperature (t=0.0) | 0.29 (0.02) | 0.31 (0.05) | 0.26 (0.07) | 0.41 (0.02) | 0.53 (0.01) | 0.53 (0.03) | 0.51 (0.04) | 0.48 (0.02) |
| gpt-4o | Temperature (t=1.0) | 0.49 (0.03) | 0.52 (0.06) | 0.68 (0.07) | 0.63 (0.01) | 0.72 (0.00) | 0.72 (0.01) | 0.75 (0.02) | 0.71 (0.01) |
| gpt-4o | Temperature (t=2.0) | 0.54 (0.02) | 0.58 (0.04) | 0.71 (0.05) | 0.66 (0.01) | 0.75 (0.00) | 0.75 (0.01) | 0.79 (0.02) | 0.74 (0.01) |
| gpt-4o | In-Context Regeneration (General) | 0.61 (0.03) | 0.96 (0.03) | 0.95 (0.03) | 0.52 (0.02) | 0.72 (0.01) | 0.75 (0.01) | 0.84 (0.02) | 0.83 (0.01) |
| gpt-4o | In-Context Regeneration (Task-Anchored) | 0.53 (0.02) | 0.96 (0.03) | 0.96 (0.03) | 0.53 (0.01) | 0.72 (0.01) | 0.76 (0.02) | 0.88 (0.01) | 0.86 (0.01) |
| gpt-4o | System Prompt (General) | 0.68 (0.02) | 0.78 (0.04) | 0.77 (0.08) | 0.55 (0.01) | 0.78 (0.01) | 0.82 (0.01) | 0.86 (0.01) | 0.85 (0.00) |
| gpt-4o | System Prompt (Task-Anchored) | 0.51 (0.02) | 0.81 (0.05) | 0.75 (0.08) | 0.65 (0.02) | 0.79 (0.01) | 0.80 (0.03) | 0.87 (0.01) | 0.86 (0.01) |
| claude-4-sonnet | Temperature (t=0.0) | 0.20 (0.03) | 0.07 (0.04) | 0.16 (0.06) | 0.28 (0.02) | 0.53 (0.02) | 0.43 (0.04) | 0.48 (0.04) | 0.42 (0.02) |
| claude-4-sonnet | Temperature (t=0.5) | 0.35 (0.02) | 0.17 (0.05) | 0.29 (0.07) | 0.35 (0.03) | 0.63 (0.01) | 0.57 (0.04) | 0.62 (0.03) | 0.56 (0.02) |
| claude-4-sonnet | Temperature (t=1.0) | 0.45 (0.02) | 0.23 (0.05) | 0.46 (0.07) | 0.42 (0.02) | 0.68 (0.01) | 0.63 (0.04) | 0.68 (0.03) | 0.63 (0.02) |
| claude-4-sonnet | In-Context Regeneration (General) | 0.74 (0.01) | 0.78 (0.05) | 0.92 (0.02) | 0.73 (0.01) | 0.78 (0.00) | 0.83 (0.01) | 0.87 (0.01) | 0.86 (0.01) |
| claude-4-sonnet | In-Context Regeneration (Task-Anchored) | 0.62 (0.01) | 0.75 (0.06) | 0.87 (0.06) | 0.76 (0.01) | 0.79 (0.01) | 0.82 (0.01) | 0.92 (0.01) | 0.87 (0.01) |
| claude-4-sonnet | System Prompt (General) | 0.71 (0.01) | 0.81 (0.01) | 0.81 (0.04) | 0.70 (0.01) | 0.78 (0.01) | 0.80 (0.01) | 0.86 (0.01) | 0.85 (0.00) |
| claude-4-sonnet | System Prompt (Task-Anchored) | 0.68 (0.01) | 0.81 (0.01) | 0.81 (0.06) | 0.75 (0.01) | 0.81 (0.00) | 0.82 (0.01) | 0.90 (0.01) | 0.86 (0.00) |
| gemini-2.5-flash | Temperature (t=0.0) | 0.07 (0.02) | 0.12 (0.05) | 0.25 (0.07) | 0.20 (0.02) | 0.35 (0.02) | 0.34 (0.03) | 0.31 (0.03) | 0.29 (0.02) |
| gemini-2.5-flash | Temperature (t=1.0) | 0.38 (0.04) | 0.44 (0.08) | 0.70 (0.07) | 0.52 (0.02) | 0.76 (0.00) | 0.74 (0.02) | 0.76 (0.03) | 0.71 (0.02) |
| gemini-2.5-flash | Temperature (t=2.0) | 0.44 (0.04) | 0.44 (0.07) | 0.77 (0.04) | 0.56 (0.02) | 0.77 (0.00) | 0.78 (0.01) | 0.79 (0.03) | 0.77 (0.01) |
| gemini-2.5-flash | In-Context Regeneration (General) | 0.76 (0.02) | 0.98 (0.02) | 0.89 (0.04) | 0.63 (0.01) | 0.69 (0.01) | 0.74 (0.02) | 0.80 (0.03) | 0.83 (0.01) |
| gemini-2.5-flash | In-Context Regeneration (Task-Anchored) | 0.60 (0.02) | 0.95 (0.03) | 0.91 (0.04) | 0.69 (0.01) | 0.71 (0.01) | 0.77 (0.02) | 0.89 (0.01) | 0.87 (0.01) |
| gemini-2.5-flash | System Prompt (General) | 0.67 (0.01) | 0.85 (0.05) | 0.86 (0.03) | 0.66 (0.01) | 0.74 (0.01) | 0.81 (0.01) | 0.85 (0.01) | 0.85 (0.00) |
| gemini-2.5-flash | System Prompt (Task-Anchored) | 0.57 (0.02) | 0.92 (0.05) | 0.73 (0.07) | 0.71 (0.01) | 0.77 (0.01) | 0.83 (0.01) | 0.89 (0.01) | 0.87 (0.00) |

Table 17: Vocabulary Diversity by Model, Sampling Strategy, and Task Category.

(Continued from Table 16)

| Model | Sampling Strategy | A | B | C | D | E | F | G | H |
|---|---|---|---|---|---|---|---|---|---|
| Llama-3.1-8B-Instruct | Temperature (t=0.0) | 0.15 (0.03) | 0.07 (0.02) | 0.24 (0.06) | 0.45 (0.03) | 0.54 (0.01) | 0.47 (0.04) | 0.46 (0.04) | 0.47 (0.02) |
| Llama-3.1-8B-Instruct | Temperature (t=0.5) | 0.43 (0.03) | 0.32 (0.05) | 0.48 (0.09) | 0.64 (0.02) | 0.66 (0.01) | 0.63 (0.02) | 0.65 (0.03) | 0.67 (0.01) |
| Llama-3.1-8B-Instruct | Temperature (t=1.0) | 0.64 (0.02) | 0.57 (0.04) | 0.74 (0.05) | 0.72 (0.01) | 0.74 (0.00) | 0.74 (0.02) | 0.77 (0.02) | 0.75 (0.01) |
| Llama-3.1-8B-Instruct | In-Context Regeneration (General) | 0.58 (0.02) | 0.86 (0.05) | 0.94 (0.03) | 0.53 (0.02) | 0.52 (0.02) | 0.64 (0.04) | 0.73 (0.03) | 0.68 (0.02) |
| Llama-3.1-8B-Instruct | In-Context Regeneration (Task-Anchored) | 0.52 (0.02) | 0.89 (0.04) | 0.87 (0.07) | 0.60 (0.01) | 0.66 (0.01) | 0.73 (0.03) | 0.84 (0.02) | 0.80 (0.01) |
| Llama-3.1-8B-Instruct | System Prompt (General) | 0.65 (0.02) | 0.73 (0.03) | 0.72 (0.04) | 0.42 (0.02) | 0.64 (0.02) | 0.66 (0.04) | 0.71 (0.03) | 0.76 (0.01) |
| Llama-3.1-8B-Instruct | System Prompt (Task-Anchored) | 0.58 (0.02) | 0.66 (0.03) | 0.49 (0.06) | 0.57 (0.03) | 0.68 (0.01) | 0.72 (0.03) | 0.76 (0.02) | 0.76 (0.01) |
| Mistral-7B-Instruct-v0.3 | Temperature (t=0.0) | 0.33 (0.03) | 0.26 (0.06) | 0.38 (0.06) | 0.49 (0.03) | 0.55 (0.01) | 0.50 (0.04) | 0.53 (0.04) | 0.46 (0.02) |
| Mistral-7B-Instruct-v0.3 | Temperature (t=0.5) | 0.60 (0.02) | 0.60 (0.03) | 0.74 (0.02) | 0.70 (0.01) | 0.67 (0.01) | 0.67 (0.02) | 0.72 (0.02) | 0.67 (0.01) |
| Mistral-7B-Instruct-v0.3 | Temperature (t=1.0) | 0.72 (0.01) | 0.75 (0.01) | 0.82 (0.02) | 0.76 (0.01) | 0.73 (0.01) | 0.75 (0.01) | 0.80 (0.01) | 0.75 (0.01) |
| Mistral-7B-Instruct-v0.3 | In-Context Regeneration (General) | 0.46 (0.02) | 0.73 (0.02) | 0.63 (0.06) | 0.27 (0.03) | 0.28 (0.02) | 0.43 (0.04) | 0.51 (0.03) | 0.53 (0.02) |
| Mistral-7B-Instruct-v0.3 | In-Context Regeneration (Task-Anchored) | 0.42 (0.02) | 0.59 (0.04) | 0.47 (0.08) | 0.31 (0.03) | 0.38 (0.02) | 0.51 (0.04) | 0.63 (0.03) | 0.59 (0.02) |
| Mistral-7B-Instruct-v0.3 | System Prompt (General) | 0.64 (0.02) | 0.80 (0.02) | 0.81 (0.03) | 0.45 (0.03) | 0.72 (0.02) | 0.74 (0.02) | 0.72 (0.03) | 0.80 (0.01) |
| Mistral-7B-Instruct-v0.3 | System Prompt (Task-Anchored) | 0.63 (0.02) | 0.77 (0.02) | 0.79 (0.02) | 0.52 (0.03) | 0.76 (0.02) | 0.81 (0.01) | 0.78 (0.02) | 0.84 (0.01) |

Table 18: Embedding Diversity by Model, Sampling Strategy, and Task Category.

| Model | Sampling Strategy | A | B | C | D | E | F | G | H |
|---|---|---|---|---|---|---|---|---|---|
| gpt-4o | Temperature (t=0.0) | 0.02 (0.00) | 0.03 (0.01) | 0.02 (0.01) | 0.01 (0.00) | 0.01 (0.00) | 0.02 (0.00) | 0.03 (0.00) | 0.02 (0.00) |
| gpt-4o | Temperature (t=1.0) | 0.04 (0.00) | 0.04 (0.01) | 0.07 (0.01) | 0.03 (0.00) | 0.03 (0.00) | 0.04 (0.00) | 0.06 (0.01) | 0.04 (0.00) |
| gpt-4o | Temperature (t=2.0) | 0.05 (0.00) | 0.06 (0.01) | 0.07 (0.02) | 0.03 (0.00) | 0.03 (0.00) | 0.04 (0.00) | 0.07 (0.01) | 0.04 (0.00) |
| gpt-4o | In-Context Regeneration (General) | 0.07 (0.01) | 0.15 (0.01) | 0.16 (0.01) | 0.04 (0.00) | 0.04 (0.00) | 0.05 (0.01) | 0.10 (0.01) | 0.09 (0.01) |
| gpt-4o | In-Context Regeneration (Task-Anchored) | 0.03 (0.00) | 0.15 (0.01) | 0.16 (0.01) | 0.03 (0.00) | 0.05 (0.00) | 0.06 (0.01) | 0.13 (0.01) | 0.11 (0.01) |
| gpt-4o | System Prompt (General) | 0.05 (0.00) | 0.14 (0.01) | 0.13 (0.01) | 0.03 (0.00) | 0.06 (0.00) | 0.08 (0.01) | 0.12 (0.01) | 0.11 (0.00) |
| gpt-4o | System Prompt (Task-Anchored) | 0.02 (0.00) | 0.13 (0.01) | 0.12 (0.01) | 0.06 (0.00) | 0.08 (0.00) | 0.10 (0.01) | 0.14 (0.01) | 0.12 (0.00) |
| claude-4-sonnet | Temperature (t=0.0) | 0.01 (0.00) | 0.01 (0.00) | 0.01 (0.00) | 0.01 (0.00) | 0.01 (0.00) | 0.01 (0.00) | 0.02 (0.00) | 0.01 (0.00) |
| claude-4-sonnet | Temperature (t=0.5) | 0.02 (0.00) | 0.01 (0.00) | 0.02 (0.00) | 0.01 (0.00) | 0.02 (0.00) | 0.02 (0.00) | 0.04 (0.00) | 0.02 (0.00) |
| claude-4-sonnet | Temperature (t=1.0) | 0.03 (0.00) | 0.02 (0.00) | 0.05 (0.01) | 0.01 (0.00) | 0.02 (0.00) | 0.02 (0.00) | 0.05 (0.00) | 0.03 (0.00) |
| claude-4-sonnet | In-Context Regeneration (General) | 0.07 (0.00) | 0.13 (0.01) | 0.14 (0.01) | 0.04 (0.00) | 0.06 (0.00) | 0.07 (0.01) | 0.11 (0.01) | 0.10 (0.00) |
| claude-4-sonnet | In-Context Regeneration (Task-Anchored) | 0.03 (0.00) | 0.12 (0.01) | 0.14 (0.01) | 0.06 (0.00) | 0.08 (0.00) | 0.09 (0.01) | 0.16 (0.01) | 0.13 (0.00) |
| claude-4-sonnet | System Prompt (General) | 0.05 (0.00) | 0.16 (0.01) | 0.12 (0.01) | 0.04 (0.00) | 0.07 (0.00) | 0.07 (0.01) | 0.12 (0.01) | 0.10 (0.00) |
| claude-4-sonnet | System Prompt (Task-Anchored) | 0.04 (0.00) | 0.16 (0.01) | 0.11 (0.01) | 0.06 (0.00) | 0.08 (0.00) | 0.08 (0.01) | 0.14 (0.01) | 0.12 (0.00) |
| gemini-2.5-flash | Temperature (t=0.0) | 0.01 (0.00) | 0.01 (0.01) | 0.02 (0.01) | 0.01 (0.00) | 0.01 (0.00) | 0.02 (0.00) | 0.02 (0.00) | 0.01 (0.00) |
| gemini-2.5-flash | Temperature (t=1.0) | 0.04 (0.01) | 0.04 (0.01) | 0.09 (0.02) | 0.02 (0.00) | 0.03 (0.00) | 0.04 (0.00) | 0.07 (0.01) | 0.04 (0.00) |
| gemini-2.5-flash | Temperature (t=2.0) | 0.05 (0.01) | 0.05 (0.01) | 0.10 (0.01) | 0.02 (0.00) | 0.04 (0.00) | 0.05 (0.01) | 0.07 (0.01) | 0.05 (0.00) |
| gemini-2.5-flash | In-Context Regeneration (General) | 0.08 (0.00) | 0.14 (0.01) | 0.15 (0.01) | 0.03 (0.00) | 0.04 (0.00) | 0.05 (0.01) | 0.10 (0.01) | 0.09 (0.01) |
| gemini-2.5-flash | In-Context Regeneration (Task-Anchored) | 0.04 (0.00) | 0.14 (0.01) | 0.15 (0.01) | 0.04 (0.00) | 0.04 (0.00) | 0.06 (0.01) | 0.13 (0.01) | 0.10 (0.00) |
| gemini-2.5-flash | System Prompt (General) | 0.04 (0.00) | 0.13 (0.01) | 0.13 (0.01) | 0.03 (0.00) | 0.05 (0.00) | 0.07 (0.01) | 0.11 (0.01) | 0.10 (0.00) |
| gemini-2.5-flash | System Prompt (Task-Anchored) | 0.02 (0.00) | 0.13 (0.01) | 0.12 (0.01) | 0.05 (0.00) | 0.07 (0.00) | 0.10 (0.01) | 0.14 (0.01) | 0.11 (0.00) |

Table 19: Embedding Diversity by Model, Sampling Strategy, and Task Category.

| Model | Sampling Strategy | A | B | C | D | E | F | G | H |
|---|---|---|---|---|---|---|---|---|---|
| | (Continued from Table 18) | | | | | | | | |
| Llama-3.1-8B-Instruct | Temperature (t=0.0) | 0.01 (0.00) | 0.01 (0.00) | 0.03 (0.01) | 0.02 (0.00) | 0.02 (0.00) | 0.02 (0.00) | 0.03 (0.00) | 0.02 (0.00) |
| Llama-3.1-8B-Instruct | Temperature (t=0.5) | 0.03 (0.00) | 0.04 (0.01) | 0.05 (0.01) | 0.04 (0.00) | 0.03 (0.00) | 0.03 (0.00) | 0.06 (0.01) | 0.04 (0.00) |
| Llama-3.1-8B-Instruct | Temperature (t=1.0) | 0.06 (0.00) | 0.08 (0.01) | 0.08 (0.01) | 0.04 (0.00) | 0.04 (0.00) | 0.05 (0.01) | 0.08 (0.01) | 0.06 (0.00) |
| Llama-3.1-8B-Instruct | In-Context Regeneration (General) | 0.07 (0.00) | 0.14 (0.01) | 0.16 (0.01) | 0.04 (0.00) | 0.04 (0.00) | 0.07 (0.01) | 0.11 (0.01) | 0.09 (0.01) |
| Llama-3.1-8B-Instruct | In-Context Regeneration (Task-Anchored) | 0.05 (0.01) | 0.12 (0.01) | 0.15 (0.01) | 0.05 (0.00) | 0.06 (0.00) | 0.09 (0.01) | 0.14 (0.01) | 0.11 (0.00) |
| Llama-3.1-8B-Instruct | System Prompt (General) | 0.07 (0.00) | 0.14 (0.01) | 0.11 (0.01) | 0.03 (0.00) | 0.07 (0.00) | 0.08 (0.01) | 0.10 (0.01) | 0.10 (0.00) |
| Llama-3.1-8B-Instruct | System Prompt (Task-Anchored) | 0.04 (0.00) | 0.12 (0.01) | 0.10 (0.01) | 0.06 (0.00) | 0.08 (0.00) | 0.11 (0.01) | 0.12 (0.01) | 0.11 (0.00) |
| Mistral-7B-Instruct-v0.3 | Temperature (t=0.0) | 0.03 (0.00) | 0.03 (0.01) | 0.03 (0.01) | 0.03 (0.00) | 0.02 (0.00) | 0.02 (0.00) | 0.03 (0.00) | 0.02 (0.00) |
| Mistral-7B-Instruct-v0.3 | Temperature (t=0.5) | 0.06 (0.00) | 0.07 (0.01) | 0.08 (0.01) | 0.05 (0.00) | 0.04 (0.00) | 0.04 (0.00) | 0.06 (0.01) | 0.04 (0.00) |
| Mistral-7B-Instruct-v0.3 | Temperature (t=1.0) | 0.08 (0.00) | 0.10 (0.01) | 0.10 (0.01) | 0.06 (0.00) | 0.04 (0.00) | 0.05 (0.00) | 0.08 (0.01) | 0.05 (0.00) |
| Mistral-7B-Instruct-v0.3 | In-Context Regeneration (General) | 0.04 (0.00) | 0.14 (0.01) | 0.11 (0.01) | 0.02 (0.00) | 0.02 (0.00) | 0.04 (0.01) | 0.06 (0.01) | 0.06 (0.00) |
| Mistral-7B-Instruct-v0.3 | In-Context Regeneration (Task-Anchored) | 0.03 (0.00) | 0.11 (0.01) | 0.08 (0.01) | 0.02 (0.00) | 0.03 (0.00) | 0.06 (0.01) | 0.09 (0.01) | 0.07 (0.00) |
| Mistral-7B-Instruct-v0.3 | System Prompt (General) | 0.07 (0.00) | 0.16 (0.01) | 0.13 (0.01) | 0.04 (0.00) | 0.08 (0.00) | 0.09 (0.01) | 0.11 (0.01) | 0.11 (0.00) |
| Mistral-7B-Instruct-v0.3 | System Prompt (Task-Anchored) | 0.06 (0.01) | 0.16 (0.01) | 0.14 (0.01) | 0.04 (0.00) | 0.10 (0.00) | 0.12 (0.01) | 0.12 (0.01) | 0.14 (0.00) |

Table 20: Compression Diversity by Model, Sampling Strategy, and Task Category.

| Model | Sampling Strategy | A | B | C | D | E | F | G | H |
|---|---|---|---|---|---|---|---|---|---|
| gpt-4o | Temperature (t=0.0) | 0.52 (0.01) | 0.61 (0.05) | 0.90 (0.19) | 0.29 (0.01) | 0.36 (0.00) | 0.38 (0.01) | 0.49 (0.02) | 0.43 (0.02) |
| gpt-4o | Temperature (t=1.0) | 0.57 (0.01) | 0.69 (0.07) | 0.94 (0.14) | 0.34 (0.01) | 0.42 (0.00) | 0.44 (0.01) | 0.55 (0.02) | 0.49 (0.02) |
| gpt-4o | Temperature (t=2.0) | 0.58 (0.01) | 0.74 (0.07) | 0.91 (0.12) | 0.36 (0.01) | 0.43 (0.00) | 0.45 (0.01) | 0.57 (0.03) | 0.50 (0.02) |
| gpt-4o | In-Context Regeneration (General) | 0.96 (0.07) | 1.51 (0.12) | 2.34 (0.30) | 0.42 (0.03) | 0.45 (0.00) | 0.49 (0.02) | 0.69 (0.06) | 0.73 (0.04) |
| gpt-4o | In-Context Regeneration (Task-Anchored) | 0.70 (0.02) | 1.55 (0.12) | 2.29 (0.31) | 0.37 (0.02) | 0.44 (0.01) | 0.54 (0.06) | 0.70 (0.05) | 0.73 (0.04) |
| gpt-4o | System Prompt (General) | 0.66 (0.01) | 0.97 (0.12) | 1.89 (0.52) | 0.41 (0.01) | 0.51 (0.00) | 0.55 (0.01) | 0.67 (0.04) | 0.63 (0.02) |
| gpt-4o | System Prompt (Task-Anchored) | 0.72 (0.02) | 1.37 (0.18) | 2.09 (0.34) | 0.45 (0.02) | 0.51 (0.00) | 0.58 (0.01) | 0.61 (0.02) | 0.63 (0.02) |
| claude-4-sonnet | Temperature (t=0.0) | 0.38 (0.01) | 0.83 (0.15) | 1.29 (0.53) | 0.28 (0.01) | 0.38 (0.00) | 0.36 (0.01) | 0.44 (0.01) | 0.43 (0.02) |
| claude-4-sonnet | Temperature (t=0.5) | 0.42 (0.01) | 0.81 (0.13) | 1.32 (0.52) | 0.29 (0.01) | 0.41 (0.00) | 0.40 (0.01) | 0.48 (0.01) | 0.46 (0.02) |
| claude-4-sonnet | Temperature (t=1.0) | 0.44 (0.01) | 0.81 (0.13) | 1.36 (0.52) | 0.30 (0.01) | 0.42 (0.00) | 0.41 (0.01) | 0.50 (0.02) | 0.48 (0.02) |
| claude-4-sonnet | In-Context Regeneration (General) | 0.55 (0.01) | 1.08 (0.15) | 1.72 (0.51) | 0.39 (0.01) | 0.45 (0.00) | 0.48 (0.01) | 0.57 (0.02) | 0.58 (0.03) |
| claude-4-sonnet | In-Context Regeneration (Task-Anchored) | 0.51 (0.01) | 1.11 (0.14) | 1.93 (0.50) | 0.38 (0.01) | 0.45 (0.00) | 0.46 (0.01) | 0.58 (0.02) | 0.59 (0.03) |
| claude-4-sonnet | System Prompt (General) | 0.51 (0.00) | 0.60 (0.02) | 1.14 (0.50) | 0.42 (0.01) | 0.48 (0.00) | 0.50 (0.01) | 0.59 (0.02) | 0.55 (0.01) |
| claude-4-sonnet | System Prompt (Task-Anchored) | 0.55 (0.01) | 0.59 (0.01) | 1.51 (0.53) | 0.42 (0.01) | 0.49 (0.00) | 0.51 (0.01) | 0.60 (0.02) | 0.55 (0.01) |
| gemini-2.5-flash | Temperature (t=0.0) | 0.57 (0.02) | 1.49 (0.22) | 1.41 (0.51) | 0.25 (0.00) | 0.30 (0.00) | 0.34 (0.01) | 0.41 (0.02) | 0.54 (0.06) |
| gemini-2.5-flash | Temperature (t=1.0) | 0.62 (0.02) | 1.55 (0.20) | 1.45 (0.50) | 0.31 (0.01) | 0.39 (0.00) | 0.42 (0.01) | 0.53 (0.03) | 0.62 (0.05) |
| gemini-2.5-flash | Temperature (t=2.0) | 0.63 (0.01) | 1.62 (0.21) | 1.42 (0.50) | 0.32 (0.01) | 0.40 (0.00) | 0.43 (0.01) | 0.55 (0.03) | 0.64 (0.05) |
| gemini-2.5-flash | In-Context Regeneration (General) | 0.92 (0.03) | 1.89 (0.14) | 2.25 (0.49) | 0.34 (0.01) | 0.39 (0.00) | 0.45 (0.02) | 0.59 (0.04) | 0.71 (0.06) |
| gemini-2.5-flash | In-Context Regeneration (Task-Anchored) | 0.77 (0.02) | 1.90 (0.17) | 1.99 (0.50) | 0.34 (0.01) | 0.38 (0.00) | 0.46 (0.02) | 0.55 (0.02) | 0.69 (0.05) |
| gemini-2.5-flash | System Prompt (General) | 0.63 (0.01) | 1.21 (0.13) | 1.39 (0.52) | 0.37 (0.01) | 0.45 (0.00) | 0.51 (0.01) | 0.61 (0.02) | 0.56 (0.01) |
| gemini-2.5-flash | System Prompt (Task-Anchored) | 0.70 (0.01) | 1.80 (0.16) | 0.93 (0.16) | 0.37 (0.01) | 0.44 (0.00) | 0.54 (0.01) | 0.58 (0.02) | 0.59 (0.03) |

Table 21: Compression Diversity by Model, Sampling Strategy, and Task Category.

| Model | Sampling Strategy | A | B | C | D | E | F | G | H |
|---|---|---|---|---|---|---|---|---|---|
| | | (Continued from Table 20) | | | | | | | |
| Llama-3.1-8B-Instruct | Temperature (t=0.0) | 0.57 (0.02) | 0.63 (0.05) | 0.82 (0.18) | 0.27 (0.01) | 0.34 (0.00) | 0.32 (0.02) | 0.43 (0.03) | 0.36 (0.01) |
| Llama-3.1-8B-Instruct | Temperature (t=0.5) | 0.64 (0.01) | 0.67 (0.05) | 0.90 (0.18) | 0.30 (0.01) | 0.37 (0.00) | 0.37 (0.01) | 0.49 (0.03) | 0.41 (0.01) |
| Llama-3.1-8B-Instruct | Temperature (t=1.0) | 0.68 (0.01) | 0.74 (0.05) | 1.01 (0.19) | 0.36 (0.01) | 0.40 (0.00) | 0.42 (0.02) | 0.51 (0.02) | 0.44 (0.01) |
| Llama-3.1-8B-Instruct | In-Context Regeneration (General) | 0.69 (0.02) | 1.31 (0.14) | 2.38 (0.30) | 0.30 (0.01) | 0.35 (0.01) | 0.41 (0.03) | 0.58 (0.05) | 0.49 (0.03) |
| Llama-3.1-8B-Instruct | In-Context Regeneration (Task-Anchored) | 0.66 (0.02) | 1.41 (0.14) | 2.38 (0.36) | 0.27 (0.01) | 0.41 (0.01) | 0.43 (0.03) | 0.57 (0.05) | 0.52 (0.03) |
| Llama-3.1-8B-Instruct | System Prompt (General) | 0.52 (0.01) | 0.56 (0.02) | 0.67 (0.07) | 0.33 (0.01) | 0.44 (0.01) | 0.45 (0.02) | 0.55 (0.02) | 0.51 (0.01) |
| Llama-3.1-8B-Instruct | System Prompt (Task-Anchored) | 0.62 (0.02) | 0.59 (0.03) | 0.77 (0.06) | 0.34 (0.02) | 0.45 (0.01) | 0.48 (0.01) | 0.54 (0.02) | 0.50 (0.01) |
| Mistral-7B-Instruct-v0.3 | Temperature (t=0.0) | 0.44 (0.01) | 0.41 (0.01) | 0.47 (0.02) | 0.32 (0.01) | 0.37 (0.00) | 0.36 (0.01) | 0.44 (0.01) | 0.38 (0.01) |
| Mistral-7B-Instruct-v0.3 | Temperature (t=0.5) | 0.51 (0.01) | 0.48 (0.02) | 0.54 (0.03) | 0.36 (0.01) | 0.41 (0.00) | 0.41 (0.01) | 0.50 (0.01) | 0.44 (0.01) |
| Mistral-7B-Instruct-v0.3 | Temperature (t=1.0) | 0.54 (0.01) | 0.52 (0.02) | 0.59 (0.03) | 0.39 (0.01) | 0.43 (0.00) | 0.44 (0.01) | 0.53 (0.02) | 0.46 (0.01) |
| Mistral-7B-Instruct-v0.3 | In-Context Regeneration (General) | 0.50 (0.01) | 0.52 (0.02) | 0.90 (0.31) | 0.31 (0.02) | 0.32 (0.01) | 0.37 (0.02) | 0.45 (0.02) | 0.42 (0.01) |
| Mistral-7B-Instruct-v0.3 | In-Context Regeneration (Task-Anchored) | 0.48 (0.01) | 0.49 (0.02) | 0.56 (0.06) | 0.33 (0.02) | 0.33 (0.01) | 0.35 (0.02) | 0.42 (0.01) | 0.40 (0.01) |
| Mistral-7B-Instruct-v0.3 | System Prompt (General) | 0.57 (0.01) | 0.62 (0.01) | 0.62 (0.02) | 0.37 (0.01) | 0.49 (0.01) | 0.54 (0.01) | 0.59 (0.02) | 0.57 (0.01) |
| Mistral-7B-Instruct-v0.3 | System Prompt (Task-Anchored) | 0.60 (0.01) | 0.62 (0.02) | 0.66 (0.02) | 0.40 (0.01) | 0.50 (0.01) | 0.56 (0.01) | 0.54 (0.01) | 0.58 (0.01) |

Table 22: Checklist-Based Quality by Model, Sampling Strategy, and Task Category.

| Model | Sampling strategy | A | B | C | D | E | F | G | H |
|---|---|---|---|---|---|---|---|---|---|
| gpt-4o | Temperature (t=0.0) | 3.76 (0.11) | 4.61 (0.15) | 4.61 (0.14) | 4.00 (0.15) | 4.62 (0.06) | 4.40 (0.19) | 4.76 (0.05) | 4.78 (0.04) |
| gpt-4o | Temperature (t=1.0) | 3.74 (0.12) | 4.65 (0.16) | 4.58 (0.17) | 3.98 (0.15) | 4.66 (0.06) | 4.41 (0.19) | 4.78 (0.05) | 4.78 (0.03) |
| gpt-4o | Temperature (t=2.0) | 3.66 (0.12) | 4.73 (0.11) | 4.52 (0.18) | 3.99 (0.16) | 4.63 (0.06) | 4.37 (0.20) | 4.74 (0.06) | 4.77 (0.04) |
| gpt-4o | In-Context Regeneration (General) | 3.37 (0.13) | 4.71 (0.14) | 4.30 (0.24) | 3.92 (0.14) | 4.42 (0.07) | 4.11 (0.19) | 4.66 (0.07) | 4.14 (0.09) |
| gpt-4o | In-Context Regeneration (Task-Anchored) | 3.54 (0.11) | 4.81 (0.09) | 4.33 (0.23) | 3.97 (0.13) | 4.26 (0.09) | 4.00 (0.20) | 4.63 (0.06) | 4.13 (0.08) |
| gpt-4o | System Prompt (General) | 3.52 (0.13) | 4.56 (0.13) | 4.30 (0.25) | 3.83 (0.16) | 4.01 (0.08) | 3.82 (0.18) | 4.70 (0.05) | 4.34 (0.07) |
| gpt-4o | System Prompt (Task-Anchored) | 3.48 (0.12) | 4.72 (0.14) | 4.32 (0.26) | 3.47 (0.17) | 3.66 (0.08) | 3.39 (0.22) | 4.45 (0.09) | 4.12 (0.08) |
| claude-4-sonnet | Temperature (t=0.0) | 3.05 (0.15) | 4.70 (0.11) | 4.12 (0.33) | 4.29 (0.13) | 4.79 (0.04) | 4.43 (0.17) | 4.65 (0.11) | 4.85 (0.03) |
| claude-4-sonnet | Temperature (t=0.5) | 3.09 (0.14) | 4.68 (0.11) | 4.19 (0.30) | 4.24 (0.13) | 4.75 (0.04) | 4.45 (0.17) | 4.69 (0.11) | 4.86 (0.03) |
| claude-4-sonnet | Temperature (t=1.0) | 3.09 (0.14) | 4.67 (0.12) | 4.23 (0.28) | 4.33 (0.12) | 4.76 (0.04) | 4.45 (0.17) | 4.73 (0.09) | 4.85 (0.03) |
| claude-4-sonnet | In-Context Regeneration (General) | 3.19 (0.12) | 4.63 (0.13) | 4.33 (0.27) | 4.30 (0.12) | 4.59 (0.07) | 4.36 (0.16) | 4.72 (0.11) | 4.70 (0.04) |
| claude-4-sonnet | In-Context Regeneration (Task-Anchored) | 3.14 (0.13) | 4.62 (0.15) | 4.52 (0.16) | 3.94 (0.11) | 4.22 (0.07) | 4.16 (0.16) | 4.53 (0.10) | 4.38 (0.06) |
| claude-4-sonnet | System Prompt (General) | 3.14 (0.14) | 4.43 (0.12) | 4.43 (0.18) | 4.30 (0.11) | 4.22 (0.07) | 4.17 (0.20) | 4.60 (0.11) | 4.53 (0.06) |
| claude-4-sonnet | System Prompt (Task-Anchored) | 3.26 (0.14) | 4.35 (0.12) | 4.37 (0.21) | 4.17 (0.11) | 4.03 (0.08) | 3.79 (0.21) | 4.60 (0.07) | 4.24 (0.07) |
| gemini-2.5-flash | Temperature (t=0.0) | 3.45 (0.12) | 4.70 (0.16) | 4.35 (0.24) | 4.13 (0.14) | 4.80 (0.04) | 4.31 (0.19) | 4.81 (0.05) | 4.56 (0.08) |
| gemini-2.5-flash | Temperature (t=1.0) | 3.37 (0.11) | 4.81 (0.10) | 4.41 (0.15) | 4.07 (0.14) | 4.81 (0.04) | 4.32 (0.18) | 4.73 (0.07) | 4.57 (0.07) |
| gemini-2.5-flash | Temperature (t=2.0) | 3.35 (0.12) | 4.85 (0.09) | 4.45 (0.14) | 4.05 (0.14) | 4.79 (0.04) | 4.23 (0.17) | 4.73 (0.06) | 4.54 (0.08) |
| gemini-2.5-flash | In-Context Regeneration (General) | 3.13 (0.13) | 4.82 (0.09) | 4.26 (0.20) | 4.13 (0.13) | 4.79 (0.03) | 4.21 (0.17) | 4.69 (0.10) | 4.32 (0.08) |
| gemini-2.5-flash | In-Context Regeneration (Task-Anchored) | 3.31 (0.11) | 4.88 (0.10) | 4.43 (0.16) | 3.93 (0.13) | 4.70 (0.05) | 4.12 (0.17) | 4.52 (0.08) | 4.33 (0.08) |
| gemini-2.5-flash | System Prompt (General) | 3.45 (0.12) | 4.82 (0.06) | 4.21 (0.18) | 4.42 (0.09) | 4.42 (0.07) | 3.95 (0.21) | 4.68 (0.07) | 4.44 (0.06) |
| gemini-2.5-flash | System Prompt (Task-Anchored) | 3.40 (0.12) | 4.89 (0.07) | 4.33 (0.20) | 4.26 (0.11) | 4.41 (0.06) | 3.49 (0.22) | 4.41 (0.10) | 4.26 (0.07) |

Table 23: Checklist-Based Quality by Model, Sampling Strategy, and Task Category.

| Model | Sampling strategy | A | B | C | D | E | F | G | H |
|---|---|---|---|---|---|---|---|---|---|
| | (Continued from Table 22) | | | | | | | | |
| Llama-3.1-8B-Instruct | Temperature (t=0.0) | 1.99 (0.09) | 4.54 (0.20) | 4.23 (0.27) | 3.09 (0.18) | 3.01 (0.13) | 3.57 (0.26) | 4.31 (0.13) | 4.52 (0.07) |
| Llama-3.1-8B-Instruct | Temperature (t=0.5) | 1.98 (0.09) | 4.59 (0.15) | 4.18 (0.25) | 3.02 (0.17) | 3.10 (0.13) | 3.70 (0.24) | 4.23 (0.13) | 4.53 (0.06) |
| Llama-3.1-8B-Instruct | Temperature (t=1.0) | 1.97 (0.08) | 4.48 (0.18) | 4.22 (0.23) | 2.93 (0.18) | 3.00 (0.12) | 3.61 (0.24) | 4.40 (0.09) | 4.50 (0.06) |
| Llama-3.1-8B-Instruct | In-Context Regeneration (General) | 2.21 (0.09) | 4.68 (0.13) | 4.12 (0.29) | 3.04 (0.17) | 2.85 (0.13) | 3.44 (0.25) | 4.37 (0.10) | 4.28 (0.08) |
| Llama-3.1-8B-Instruct | In-Context Regeneration (Task-Anchored) | 2.07 (0.09) | 4.59 (0.18) | 4.13 (0.23) | 2.62 (0.16) | 2.49 (0.10) | 2.93 (0.22) | 3.78 (0.14) | 3.97 (0.08) |
| Llama-3.1-8B-Instruct | System Prompt (General) | 2.22 (0.12) | 4.19 (0.17) | 4.05 (0.29) | 2.79 (0.19) | 2.38 (0.09) | 2.98 (0.19) | 3.95 (0.12) | 3.74 (0.09) |
| Llama-3.1-8B-Instruct | System Prompt (Task-Anchored) | 2.72 (0.15) | 4.42 (0.13) | 3.88 (0.27) | 2.51 (0.17) | 2.23 (0.08) | 2.48 (0.18) | 3.77 (0.15) | 3.61 (0.10) |
| Mistral-7B-Instruct-v0.3 | Temperature (t=0.0) | 2.58 (0.13) | 4.05 (0.25) | 3.37 (0.23) | 1.90 (0.15) | 3.22 (0.13) | 3.63 (0.24) | 3.75 (0.15) | 4.37 (0.07) |
| Mistral-7B-Instruct-v0.3 | Temperature (t=0.5) | 2.54 (0.12) | 3.97 (0.24) | 3.20 (0.27) | 1.83 (0.13) | 3.15 (0.13) | 3.62 (0.23) | 3.82 (0.14) | 4.39 (0.07) |
| Mistral-7B-Instruct-v0.3 | Temperature (t=1.0) | 2.41 (0.11) | 4.08 (0.21) | 3.29 (0.25) | 1.71 (0.13) | 3.12 (0.13) | 3.61 (0.22) | 3.86 (0.14) | 4.35 (0.07) |
| Mistral-7B-Instruct-v0.3 | In-Context Regeneration (General) | 2.64 (0.13) | 4.15 (0.13) | 3.44 (0.23) | 1.89 (0.14) | 2.95 (0.14) | 3.36 (0.23) | 3.75 (0.15) | 4.17 (0.07) |
| Mistral-7B-Instruct-v0.3 | In-Context Regeneration (Task-Anchored) | 2.57 (0.14) | 4.00 (0.25) | 3.60 (0.32) | 1.84 (0.15) | 2.70 (0.13) | 3.06 (0.22) | 3.45 (0.16) | 3.94 (0.08) |
| Mistral-7B-Instruct-v0.3 | System Prompt (General) | 2.48 (0.12) | 3.99 (0.21) | 3.07 (0.37) | 1.50 (0.10) | 2.24 (0.08) | 2.54 (0.18) | 3.45 (0.17) | 3.59 (0.09) |
| Mistral-7B-Instruct-v0.3 | System Prompt (Task-Anchored) | 2.55 (0.12) | 4.21 (0.17) | 3.13 (0.34) | 1.66 (0.13) | 2.08 (0.08) | 2.22 (0.17) | 3.12 (0.16) | 3.25 (0.11) |

Table 24: Athene-RM-8B Reward by Model, Sampling Strategy, and Task Category.

| Model | Sampling Strategy | A | B | C | D | E | F | G | H |
|---|---|---|---|---|---|---|---|---|---|
| gpt-4o | Temperature (t=0.0) | 0.37 (0.09) | 0.46 (0.13) | 0.78 (0.14) | 0.50 (0.12) | 0.28 (0.06) | 0.94 (0.21) | 0.81 (0.09) | 0.96 (0.08) |
| gpt-4o | Temperature (t=1.0) | 0.35 (0.08) | 0.43 (0.14) | 0.78 (0.12) | 0.47 (0.13) | 0.49 (0.06) | 0.97 (0.20) | 0.81 (0.10) | 0.98 (0.07) |
| gpt-4o | Temperature (t=2.0) | 0.34 (0.08) | 0.41 (0.12) | 0.80 (0.12) | 0.45 (0.14) | 0.56 (0.08) | 1.04 (0.21) | 0.85 (0.10) | 1.07 (0.07) |
| gpt-4o | In-Context Regeneration (General) | -0.38 (0.08) | -0.36 (0.11) | 0.04 (0.21) | 0.45 (0.11) | -0.40 (0.07) | 0.30 (0.19) | 0.65 (0.15) | -0.38 (0.10) |
| gpt-4o | In-Context Regeneration (Task-Anchored) | -0.08 (0.08) | -0.36 (0.10) | 0.05 (0.21) | 0.54 (0.12) | -0.24 (0.08) | 0.29 (0.24) | 0.61 (0.16) | -0.24 (0.10) |
| gpt-4o | System Prompt (General) | -0.05 (0.08) | -0.05 (0.16) | 0.15 (0.22) | 0.39 (0.13) | -1.12 (0.08) | -0.64 (0.18) | 0.22 (0.13) | -0.70 (0.10) |
| gpt-4o | System Prompt (Task-Anchored) | -0.19 (0.09) | -0.25 (0.19) | -0.13 (0.24) | -0.02 (0.12) | -1.18 (0.07) | -1.25 (0.20) | 0.16 (0.15) | -0.91 (0.11) |
| claude-4-sonnet | Temperature (t=0.0) | 0.42 (0.08) | 0.12 (0.20) | 0.69 (0.20) | 0.08 (0.15) | 1.10 (0.07) | 1.07 (0.16) | 0.96 (0.12) | 1.13 (0.07) |
| claude-4-sonnet | Temperature (t=0.5) | 0.43 (0.08) | 0.11 (0.21) | 0.68 (0.20) | 0.03 (0.15) | 1.08 (0.07) | 1.07 (0.17) | 0.97 (0.12) | 1.17 (0.07) |
| claude-4-sonnet | Temperature (t=1.0) | 0.41 (0.08) | 0.11 (0.21) | 0.67 (0.20) | 0.14 (0.14) | 1.09 (0.07) | 1.13 (0.15) | 0.96 (0.11) | 1.16 (0.07) |
| claude-4-sonnet | In-Context Regeneration (General) | 0.33 (0.08) | -0.06 (0.16) | 0.46 (0.16) | 0.25 (0.13) | 0.88 (0.07) | 0.89 (0.15) | 0.70 (0.11) | 0.78 (0.07) |
| claude-4-sonnet | In-Context Regeneration (Task-Anchored) | 0.26 (0.08) | -0.25 (0.19) | 0.42 (0.16) | -0.15 (0.11) | 0.76 (0.06) | 0.74 (0.15) | 0.48 (0.12) | 0.71 (0.07) |
| claude-4-sonnet | System Prompt (General) | 0.29 (0.08) | 0.20 (0.12) | 0.26 (0.21) | 0.39 (0.11) | -0.25 (0.08) | 0.28 (0.18) | 0.53 (0.12) | 0.29 (0.08) |
| claude-4-sonnet | System Prompt (Task-Anchored) | 0.28 (0.08) | 0.12 (0.15) | 0.29 (0.19) | 0.18 (0.11) | -0.35 (0.08) | -0.16 (0.16) | 0.50 (0.13) | -0.02 (0.10) |
| gemini-2.5-flash | Temperature (t=0.0) | 0.05 (0.10) | -0.30 (0.09) | 0.22 (0.17) | -0.20 (0.14) | 1.25 (0.15) | 0.65 (0.23) | 0.64 (0.14) | 0.56 (0.14) |
| gemini-2.5-flash | Temperature (t=1.0) | 0.12 (0.09) | -0.21 (0.10) | 0.37 (0.15) | -0.24 (0.15) | 1.27 (0.12) | 0.70 (0.26) | 0.67 (0.13) | 0.55 (0.13) |
| gemini-2.5-flash | Temperature (t=2.0) | 0.13 (0.09) | -0.31 (0.10) | 0.35 (0.15) | -0.26 (0.15) | 1.05 (0.13) | 0.69 (0.22) | 0.59 (0.12) | 0.54 (0.13) |
| gemini-2.5-flash | In-Context Regeneration (General) | -0.51 (0.09) | -0.64 (0.11) | 0.27 (0.14) | 0.04 (0.13) | 1.27 (0.09) | 0.71 (0.22) | 0.62 (0.14) | 0.45 (0.15) |
| gemini-2.5-flash | In-Context Regeneration (Task-Anchored) | -0.16 (0.09) | -0.62 (0.11) | 0.15 (0.15) | -0.28 (0.13) | 1.27 (0.09) | 0.56 (0.24) | 0.46 (0.15) | 0.44 (0.13) |
| gemini-2.5-flash | System Prompt (General) | 0.10 (0.08) | -0.28 (0.12) | -0.04 (0.23) | 0.24 (0.11) | -0.23 (0.07) | 0.16 (0.19) | 0.69 (0.11) | 0.26 (0.09) |
| gemini-2.5-flash | System Prompt (Task-Anchored) | -0.22 (0.08) | -0.56 (0.09) | 0.08 (0.20) | 0.12 (0.12) | 0.37 (0.09) | -0.62 (0.18) | 0.32 (0.16) | 0.02 (0.11) |

Table 25: Athene-RM-8B Reward by Model, Sampling Strategy, and Task Category.

| Model | Sampling Strategy | A | B | C | D | E | F | G | H |
|---|---|---|---|---|---|---|---|---|---|
| | (Continued from Table 24) | | | | | | | | |
| Llama-3.1-8B-Instruct | Temperature (t=0.0) | -0.84 (0.07) | 0.02 (0.10) | 0.03 (0.20) | -0.87 (0.18) | -0.91 (0.07) | -0.49 (0.27) | -0.22 (0.15) | 0.04 (0.08) |
| Llama-3.1-8B-Instruct | Temperature (t=0.5) | -0.81 (0.07) | 0.02 (0.12) | 0.03 (0.14) | -0.94 (0.17) | -0.90 (0.07) | -0.30 (0.18) | -0.16 (0.14) | 0.04 (0.07) |
| Llama-3.1-8B-Instruct | Temperature (t=1.0) | -0.95 (0.06) | -0.17 (0.16) | 0.14 (0.13) | -0.86 (0.15) | -0.76 (0.07) | -0.29 (0.17) | -0.06 (0.12) | 0.06 (0.07) |
| Llama-3.1-8B-Instruct | In-Context Regeneration (General) | -0.62 (0.06) | -0.34 (0.15) | -0.21 (0.23) | -0.92 (0.17) | -1.34 (0.08) | -0.87 (0.14) | -0.29 (0.14) | -0.51 (0.08) |
| Llama-3.1-8B-Instruct | In-Context Regeneration (Task-Anchored) | -0.74 (0.06) | -0.57 (0.12) | -0.09 (0.16) | -1.47 (0.16) | -1.59 (0.08) | -1.24 (0.13) | -0.75 (0.15) | -0.92 (0.09) |
| Llama-3.1-8B-Instruct | System Prompt (General) | -0.24 (0.08) | -0.36 (0.22) | -0.02 (0.19) | -0.83 (0.16) | -2.08 (0.09) | -1.53 (0.17) | -0.80 (0.17) | -1.35 (0.11) |
| Llama-3.1-8B-Instruct | System Prompt (Task-Anchored) | -0.14 (0.09) | -0.19 (0.19) | 0.00 (0.22) | -1.05 (0.15) | -2.24 (0.09) | -2.04 (0.19) | -0.86 (0.18) | -1.48 (0.12) |
| Mistral-7B-Instruct-v0.3 | Temperature (t=0.0) | -0.08 (0.07) | -0.08 (0.20) | -0.78 (0.15) | -1.30 (0.15) | -1.12 (0.08) | -0.45 (0.13) | -0.35 (0.13) | -0.39 (0.07) |
| Mistral-7B-Instruct-v0.3 | Temperature (t=0.5) | -0.06 (0.06) | -0.07 (0.16) | -0.84 (0.15) | -1.37 (0.13) | -1.06 (0.08) | -0.39 (0.11) | -0.29 (0.11) | -0.32 (0.06) |
| Mistral-7B-Instruct-v0.3 | Temperature (t=1.0) | -0.09 (0.06) | -0.17 (0.16) | -0.74 (0.14) | -1.40 (0.12) | -0.98 (0.07) | -0.36 (0.12) | -0.18 (0.10) | -0.28 (0.06) |
| Mistral-7B-Instruct-v0.3 | In-Context Regeneration (General) | -0.18 (0.07) | -0.33 (0.13) | -0.74 (0.20) | -1.32 (0.13) | -1.22 (0.09) | -0.76 (0.13) | -0.46 (0.12) | -0.73 (0.08) |
| Mistral-7B-Instruct-v0.3 | In-Context Regeneration (Task-Anchored) | -0.10 (0.07) | -0.48 (0.15) | -0.59 (0.21) | -1.38 (0.13) | -1.54 (0.11) | -1.01 (0.17) | -1.00 (0.12) | -1.07 (0.09) |
| Mistral-7B-Instruct-v0.3 | System Prompt (General) | -0.41 (0.08) | -0.49 (0.25) | -1.17 (0.22) | -1.81 (0.13) | -2.19 (0.08) | -2.06 (0.21) | -1.04 (0.17) | -1.81 (0.13) |
| Mistral-7B-Instruct-v0.3 | System Prompt (Task-Anchored) | -0.41 (0.08) | -0.33 (0.18) | -1.20 (0.25) | -1.86 (0.15) | -2.30 (0.09) | -2.46 (0.19) | -1.29 (0.16) | -2.15 (0.15) |

Table 26: Accuracy by Model, Sampling Strategy, and Evaluation Dataset.

(For Tasks with Singular Verifiable Rewards)

| Model | Sampling Strategy | Math-500 | Simple-QA |
|---|---|---|---|
| gpt-4o | Temperature (t=0.0) | 0.59 (0.06) | 0.37 (0.06) |
| gpt-4o | Temperature (t=1.0) | 0.59 (0.06) | 0.36 (0.06) |
| gpt-4o | Temperature (t=2.0) | 0.57 (0.06) | 0.38 (0.06) |
| gpt-4o | In-Context Regeneration (General) | 0.57 (0.07) | 0.30 (0.06) |
| gpt-4o | In-Context Regeneration (Task-Anchored) | 0.59 (0.07) | 0.35 (0.07) |
| gpt-4o | System Prompt (General) | 0.61 (0.07) | 0.37 (0.07) |
| gpt-4o | System Prompt (Task-Anchored) | 0.63 (0.07) | 0.28 (0.06) |
| claude-4-sonnet | Temperature (t=0.0) | 0.69 (0.06) | 0.17 (0.05) |
| claude-4-sonnet | Temperature (t=0.5) | 0.68 (0.06) | 0.17 (0.05) |
| claude-4-sonnet | Temperature (t=1.0) | 0.68 (0.06) | 0.18 (0.05) |
| claude-4-sonnet | In-Context Regeneration (General) | 0.71 (0.06) | 0.19 (0.05) |
| claude-4-sonnet | In-Context Regeneration (Task-Anchored) | 0.69 (0.06) | 0.17 (0.05) |
| claude-4-sonnet | System Prompt (General) | 0.70 (0.06) | 0.21 (0.06) |
| claude-4-sonnet | System Prompt (Task-Anchored) | 0.73 (0.06) | 0.21 (0.06) |
| gemini-2.5-flash | Temperature (t=0.0) | 0.63 (0.07) | 0.34 (0.06) |
| gemini-2.5-flash | Temperature (t=1.0) | 0.64 (0.06) | 0.27 (0.05) |
| gemini-2.5-flash | Temperature (t=2.0) | 0.63 (0.06) | 0.25 (0.05) |
| gemini-2.5-flash | In-Context Regeneration (General) | 0.65 (0.06) | 0.31 (0.06) |
| gemini-2.5-flash | In-Context Regeneration (Task-Anchored) | 0.62 (0.06) | 0.35 (0.07) |
| gemini-2.5-flash | System Prompt (General) | 0.75 (0.06) | 0.25 (0.06) |
| gemini-2.5-flash | System Prompt (Task-Anchored) | 0.73 (0.06) | 0.29 (0.06) |

Table 27: Accuracy by Model, Sampling Strategy, and Evaluation Dataset.

(For Tasks with Singular Verifiable Rewards)
(Continued from Table 26)

| Model | Sampling Strategy | Math-500 | Simple-QA |
|---|---|---|---|
| Llama-3.1-8B-Instruct | Temperature (t=0.0) | 0.40 (0.06) | 0.03 (0.02) |
| Llama-3.1-8B-Instruct | Temperature (t=1.0) | 0.36 (0.05) | 0.03 (0.02) |
| Llama-3.1-8B-Instruct | Temperature (t=2.0) | 0.36 (0.05) | 0.03 (0.02) |
| Llama-3.1-8B-Instruct | In-Context Regeneration (General) | 0.39 (0.06) | 0.03 (0.02) |
| Llama-3.1-8B-Instruct | In-Context Regeneration (Task-Anchored) | 0.33 (0.06) | 0.02 (0.01) |
| Llama-3.1-8B-Instruct | System Prompt (General) | 0.41 (0.07) | 0.05 (0.03) |
| Llama-3.1-8B-Instruct | System Prompt (Task-Anchored) | 0.41 (0.06) | 0.08 (0.04) |
| Mistral-7B-Instruct-v0.3 | Temperature (t=0.0) | 0.08 (0.03) | 0.04 (0.02) |
| Mistral-7B-Instruct-v0.3 | Temperature (t=1.0) | 0.08 (0.02) | 0.05 (0.03) |
| Mistral-7B-Instruct-v0.3 | Temperature (t=2.0) | 0.06 (0.02) | 0.06 (0.03) |
| Mistral-7B-Instruct-v0.3 | In-Context Regeneration (General) | 0.09 (0.04) | 0.02 (0.02) |
| Mistral-7B-Instruct-v0.3 | In-Context Regeneration (Task-Anchored) | 0.14 (0.05) | 0.06 (0.03) |
| Mistral-7B-Instruct-v0.3 | System Prompt (General) | 0.06 (0.03) | 0.06 (0.03) |
| Mistral-7B-Instruct-v0.3 | System Prompt (Task-Anchored) | 0.09 (0.03) | 0.05 (0.02) |

Table 28: # of Functionally Diverse Responses by Model, Sampling Strategy, and Task Category.

| Model | Sampling Strategy | A | B | C | D | E | F | G | H |
|---|---|---|---|---|---|---|---|---|---|
| | (Using Only GPT-4o as the Functional Diversity Judge) | | | | | | | | |
| gpt-4o | Temperature (t=0.0) | 1.58 (0.11) | 1.25 (0.11) | 1.64 (0.27) | 1.11 (0.06) | 1.08 (0.04) | 1.13 (0.10) | 1.33 (0.12) | 1.09 (0.03) |
| gpt-4o | Temperature (t=1.0) | 2.09 (0.19) | 1.38 (0.15) | 2.86 (0.46) | 1.05 (0.04) | 1.10 (0.05) | 1.13 (0.13) | 1.58 (0.17) | 1.22 (0.06) |
| gpt-4o | Temperature (t=2.0) | 2.38 (0.21) | 1.75 (0.27) | 2.93 (0.40) | 1.11 (0.05) | 1.08 (0.05) | 1.09 (0.09) | 2.09 (0.20) | 1.26 (0.06) |
| gpt-4o | In-Context Regeneration (General) | 2.17 (0.22) | 5.00 (0.00) | 5.00 (0.00) | 1.22 (0.10) | 1.66 (0.13) | 1.39 (0.24) | 2.56 (0.26) | 2.90 (0.20) |
| gpt-4o | In-Context Regeneration (Task-Anchored) | 1.08 (0.04) | 5.00 (0.00) | 5.00 (0.00) | 1.09 (0.05) | 2.16 (0.20) | 1.78 (0.29) | 3.58 (0.25) | 3.25 (0.20) |
| gpt-4o | System Prompt (General) | 1.92 (0.22) | 5.00 (0.00) | 5.00 (0.00) | 1.10 (0.08) | 3.46 (0.22) | 2.35 (0.39) | 3.29 (0.27) | 3.66 (0.19) |
| gpt-4o | System Prompt (Task-Anchored) | 1.00 (0.00) | 5.00 (0.00) | 5.00 (0.00) | 1.37 (0.10) | 4.36 (0.14) | 2.86 (0.37) | 4.42 (0.17) | 4.25 (0.14) |
| claude-4-sonnet | Temperature (t=0.0) | 1.06 (0.03) | 1.00 (0.00) | 1.00 (0.00) | 1.02 (0.02) | 1.12 (0.05) | 1.00 (0.00) | 1.13 (0.06) | 1.09 (0.03) |
| claude-4-sonnet | Temperature (t=0.5) | 1.21 (0.08) | 1.06 (0.06) | 1.29 (0.16) | 1.00 (0.00) | 1.10 (0.05) | 1.04 (0.04) | 1.36 (0.14) | 1.14 (0.05) |
| claude-4-sonnet | Temperature (t=1.0) | 1.28 (0.08) | 1.19 (0.19) | 1.86 (0.27) | 1.02 (0.02) | 1.18 (0.07) | 1.04 (0.04) | 1.62 (0.17) | 1.24 (0.07) |
| claude-4-sonnet | In-Context Regeneration (General) | 1.58 (0.13) | 5.00 (0.00) | 4.43 (0.39) | 1.22 (0.09) | 3.26 (0.22) | 1.52 (0.29) | 3.33 (0.28) | 2.99 (0.20) |
| claude-4-sonnet | In-Context Regeneration (Task-Anchored) | 1.08 (0.08) | 5.00 (0.00) | 4.79 (0.21) | 1.71 (0.15) | 4.82 (0.07) | 2.52 (0.32) | 4.60 (0.15) | 3.99 (0.15) |
| claude-4-sonnet | System Prompt (General) | 1.19 (0.11) | 5.00 (0.00) | 4.71 (0.29) | 1.27 (0.12) | 4.08 (0.16) | 1.91 (0.33) | 3.75 (0.25) | 3.51 (0.19) |
| claude-4-sonnet | System Prompt (Task-Anchored) | 1.08 (0.08) | 5.00 (0.00) | 5.00 (0.00) | 1.56 (0.15) | 4.74 (0.10) | 2.43 (0.34) | 4.44 (0.18) | 4.30 (0.14) |
| gemini-2.5-flash | Temperature (t=0.0) | 1.25 (0.07) | 1.12 (0.09) | 1.57 (0.17) | 1.00 (0.00) | 1.08 (0.04) | 1.09 (0.06) | 1.16 (0.05) | 1.08 (0.03) |
| gemini-2.5-flash | Temperature (t=1.0) | 2.53 (0.21) | 1.81 (0.26) | 3.07 (0.38) | 1.05 (0.04) | 1.34 (0.10) | 1.13 (0.07) | 2.00 (0.21) | 1.41 (0.10) |
| gemini-2.5-flash | Temperature (t=2.0) | 2.70 (0.21) | 1.88 (0.20) | 3.14 (0.39) | 1.00 (0.00) | 1.24 (0.09) | 1.26 (0.09) | 2.38 (0.24) | 1.66 (0.13) |
| gemini-2.5-flash | In-Context Regeneration (General) | 1.28 (0.12) | 4.94 (0.06) | 5.00 (0.00) | 1.15 (0.09) | 1.34 (0.12) | 1.22 (0.18) | 2.78 (0.27) | 2.85 (0.20) |
| gemini-2.5-flash | In-Context Regeneration (Task-Anchored) | 1.08 (0.08) | 5.00 (0.00) | 4.93 (0.07) | 1.20 (0.08) | 2.18 (0.18) | 1.48 (0.24) | 3.98 (0.21) | 3.16 (0.20) |
| gemini-2.5-flash | System Prompt (General) | 1.17 (0.11) | 4.94 (0.06) | 5.00 (0.00) | 1.15 (0.09) | 2.46 (0.22) | 1.91 (0.32) | 3.22 (0.27) | 3.45 (0.19) |
| gemini-2.5-flash | System Prompt (Task-Anchored) | 1.06 (0.06) | 4.94 (0.06) | 5.00 (0.00) | 1.53 (0.16) | 3.62 (0.20) | 2.91 (0.39) | 4.31 (0.19) | 3.94 (0.17) |

Table 29: # of Functionally Diverse Responses by Model, Sampling Strategy, and Task Category.

| Model | Sampling Strategy | A | B | C | D | E | F | G | H |
|-------|-------------------|---|---|---|---|---|---|---|---|
| | (Using Only Claude-4-Sonnet as the Functional Diversity Judge) | | | | | | | | |
| gpt-4o | Temperature (t=0.0) | 1.57 (0.11) | 1.19 (0.10) | 1.86 (0.31) | 1.40 (0.11) | 1.20 (0.06) | 1.09 (0.09) | 1.11 (0.06) | 1.08 (0.03) |
| gpt-4o | Temperature (t=1.0) | 2.11 (0.19) | 1.44 (0.16) | 3.14 (0.46) | 1.31 (0.11) | 1.36 (0.10) | 1.13 (0.13) | 1.38 (0.13) | 1.18 (0.05) |
| gpt-4o | Temperature (t=2.0) | 2.38 (0.21) | 1.69 (0.25) | 3.14 (0.42) | 1.29 (0.09) | 1.36 (0.11) | 1.09 (0.09) | 1.87 (0.20) | 1.24 (0.06) |
| gpt-4o | In-Context Regeneration (General) | 2.19 (0.22) | 4.94 (0.06) | 5.00 (0.00) | 1.38 (0.14) | 1.82 (0.16) | 1.35 (0.24) | 2.33 (0.24) | 2.82 (0.20) |
| gpt-4o | In-Context Regeneration (Task-Anchored) | 1.08 (0.04) | 5.00 (0.00) | 5.00 (0.00) | 1.36 (0.08) | 2.92 (0.21) | 1.65 (0.26) | 3.36 (0.24) | 3.18 (0.20) |
| gpt-4o | System Prompt (General) | 1.94 (0.22) | 5.00 (0.00) | 5.00 (0.00) | 1.10 (0.06) | 3.86 (0.19) | 2.30 (0.39) | 2.82 (0.26) | 3.64 (0.19) |
| gpt-4o | System Prompt (Task-Anchored) | 1.06 (0.06) | 5.00 (0.00) | 5.00 (0.00) | 1.85 (0.14) | 4.68 (0.09) | 2.91 (0.35) | 4.07 (0.20) | 4.16 (0.15) |
| Model | Sampling strategy | A | B | C | D | E | F | G | H |
| claude-4-sonnet | Temperature (t=0.0) | 1.06 (0.03) | 1.00 (0.00) | 1.29 (0.22) | 1.00 (0.00) | 1.40 (0.11) | 1.04 (0.04) | 1.11 (0.06) | 1.10 (0.04) |
| claude-4-sonnet | Temperature (t=0.5) | 1.21 (0.08) | 1.06 (0.06) | 1.64 (0.32) | 1.02 (0.02) | 1.38 (0.12) | 1.17 (0.10) | 1.18 (0.10) | 1.11 (0.04) |
| claude-4-sonnet | Temperature (t=1.0) | 1.28 (0.08) | 1.19 (0.19) | 2.36 (0.37) | 1.04 (0.03) | 1.48 (0.12) | 1.13 (0.10) | 1.36 (0.12) | 1.22 (0.07) |
| claude-4-sonnet | In-Context Regeneration (General) | 1.62 (0.14) | 5.00 (0.00) | 4.93 (0.07) | 1.40 (0.10) | 4.08 (0.19) | 1.52 (0.27) | 2.91 (0.27) | 2.90 (0.20) |
| claude-4-sonnet | In-Context Regeneration (Task-Anchored) | 1.08 (0.08) | 5.00 (0.00) | 5.00 (0.00) | 2.58 (0.18) | 4.96 (0.03) | 3.26 (0.31) | 4.62 (0.15) | 4.30 (0.12) |
| claude-4-sonnet | System Prompt (General) | 1.17 (0.10) | 5.00 (0.00) | 5.00 (0.00) | 1.33 (0.11) | 4.52 (0.13) | 2.00 (0.32) | 2.91 (0.27) | 3.49 (0.20) |
| claude-4-sonnet | System Prompt (Task-Anchored) | 1.06 (0.06) | 5.00 (0.00) | 5.00 (0.00) | 1.84 (0.15) | 4.88 (0.06) | 2.35 (0.31) | 4.09 (0.19) | 4.25 (0.13) |
| gemini-2.5-flash | Temperature (t=0.0) | 1.25 (0.07) | 1.12 (0.09) | 1.57 (0.17) | 1.02 (0.02) | 1.28 (0.06) | 1.09 (0.06) | 1.11 (0.05) | 1.08 (0.03) |
| gemini-2.5-flash | Temperature (t=1.0) | 2.55 (0.21) | 1.81 (0.26) | 3.21 (0.41) | 1.11 (0.06) | 2.22 (0.18) | 1.30 (0.15) | 1.98 (0.20) | 1.40 (0.09) |
| gemini-2.5-flash | Temperature (t=2.0) | 2.62 (0.19) | 1.94 (0.19) | 3.29 (0.41) | 1.07 (0.04) | 2.40 (0.18) | 1.35 (0.13) | 2.18 (0.23) | 1.60 (0.12) |
| gemini-2.5-flash | In-Context Regeneration (General) | 1.32 (0.13) | 4.94 (0.06) | 5.00 (0.00) | 1.33 (0.10) | 1.66 (0.14) | 1.22 (0.18) | 2.27 (0.25) | 2.84 (0.20) |
| gemini-2.5-flash | In-Context Regeneration (Task-Anchored) | 1.06 (0.06) | 5.00 (0.00) | 5.00 (0.00) | 1.62 (0.12) | 3.64 (0.19) | 1.70 (0.25) | 3.53 (0.22) | 3.41 (0.18) |
| gemini-2.5-flash | System Prompt (General) | 1.23 (0.13) | 4.94 (0.06) | 5.00 (0.00) | 1.19 (0.07) | 2.84 (0.21) | 1.78 (0.33) | 2.36 (0.26) | 3.48 (0.19) |
| gemini-2.5-flash | System Prompt (Task-Anchored) | 1.09 (0.07) | 5.00 (0.00) | 5.00 (0.00) | 1.64 (0.14) | 4.28 (0.16) | 2.83 (0.38) | 4.22 (0.16) | 3.97 (0.16) |

Table 30: # of Functionally Diverse Responses by Model, Sampling Strategy, and Task Category.

| Model | Sampling Strategy | A | B | C | D | E | F | G | H |
|---|---|---|---|---|---|---|---|---|---|
| | (Using Only Gemini-2.5-Flash as the Functional Diversity Judge) | | | | | | | | |
| gpt-4o | Temperature (t=0.0) | 1.58 (0.11) | 1.19 (0.10) | 1.93 (0.35) | 1.49 (0.13) | 1.72 (0.13) | 1.17 (0.10) | 1.16 (0.05) | 1.07 (0.03) |
| gpt-4o | Temperature (t=1.0) | 2.08 (0.18) | 1.50 (0.18) | 3.14 (0.46) | 1.49 (0.14) | 1.96 (0.17) | 1.17 (0.14) | 1.27 (0.09) | 1.19 (0.06) |
| gpt-4o | Temperature (t=2.0) | 2.38 (0.20) | 1.62 (0.26) | 3.14 (0.42) | 1.56 (0.14) | 1.94 (0.19) | 1.22 (0.13) | 1.62 (0.14) | 1.25 (0.06) |
| gpt-4o | In-Context Regeneration (General) | 2.19 (0.22) | 4.94 (0.06) | 5.00 (0.00) | 1.33 (0.11) | 2.12 (0.18) | 1.43 (0.25) | 1.69 (0.17) | 2.67 (0.20) |
| gpt-4o | In-Context Regeneration (Task-Anchored) | 1.06 (0.03) | 5.00 (0.00) | 5.00 (0.00) | 1.36 (0.08) | 3.26 (0.20) | 1.87 (0.29) | 2.60 (0.23) | 3.14 (0.20) |
| gpt-4o | System Prompt (General) | 1.92 (0.22) | 5.00 (0.00) | 5.00 (0.00) | 1.17 (0.09) | 3.88 (0.18) | 2.26 (0.38) | 2.44 (0.24) | 3.62 (0.19) |
| gpt-4o | System Prompt (Task-Anchored) | 1.00 (0.00) | 4.94 (0.06) | 5.00 (0.00) | 1.78 (0.16) | 4.58 (0.11) | 2.86 (0.37) | 3.93 (0.22) | 4.18 (0.15) |
| claude-4-sonnet | Temperature (t=0.0) | 1.06 (0.03) | 1.00 (0.00) | 1.29 (0.22) | 1.11 (0.06) | 1.76 (0.16) | 1.13 (0.10) | 1.16 (0.06) | 1.11 (0.04) |
| claude-4-sonnet | Temperature (t=0.5) | 1.21 (0.08) | 1.06 (0.06) | 1.71 (0.35) | 1.09 (0.05) | 2.08 (0.16) | 1.17 (0.10) | 1.29 (0.11) | 1.15 (0.04) |
| claude-4-sonnet | Temperature (t=1.0) | 1.28 (0.08) | 1.19 (0.19) | 2.14 (0.35) | 1.13 (0.05) | 2.12 (0.16) | 1.13 (0.07) | 1.33 (0.11) | 1.24 (0.07) |
| claude-4-sonnet | In-Context Regeneration (General) | 1.64 (0.13) | 5.00 (0.00) | 4.93 (0.07) | 1.45 (0.11) | 4.18 (0.18) | 1.74 (0.30) | 2.44 (0.26) | 2.88 (0.20) |
| claude-4-sonnet | In-Context Regeneration (Task-Anchored) | 1.08 (0.08) | 5.00 (0.00) | 4.93 (0.07) | 2.69 (0.18) | 4.90 (0.04) | 3.30 (0.34) | 4.09 (0.20) | 4.00 (0.14) |
| claude-4-sonnet | System Prompt (General) | 1.17 (0.10) | 5.00 (0.00) | 5.00 (0.00) | 1.42 (0.13) | 4.42 (0.13) | 2.35 (0.37) | 2.64 (0.27) | 3.57 (0.19) |
| claude-4-sonnet | System Prompt (Task-Anchored) | 1.11 (0.08) | 5.00 (0.00) | 5.00 (0.00) | 1.91 (0.16) | 4.82 (0.07) | 2.74 (0.33) | 3.91 (0.21) | 4.27 (0.13) |
| gemini-2.5-flash | Temperature (t=0.0) | 1.25 (0.07) | 1.12 (0.09) | 1.57 (0.17) | 1.04 (0.03) | 1.30 (0.07) | 1.09 (0.06) | 1.13 (0.05) | 1.07 (0.03) |
| gemini-2.5-flash | Temperature (t=1.0) | 2.55 (0.21) | 1.81 (0.26) | 3.21 (0.41) | 1.27 (0.10) | 2.48 (0.20) | 1.26 (0.11) | 1.58 (0.18) | 1.40 (0.10) |
| gemini-2.5-flash | Temperature (t=2.0) | 2.66 (0.21) | 1.88 (0.20) | 3.29 (0.41) | 1.09 (0.05) | 2.16 (0.17) | 1.48 (0.16) | 1.96 (0.21) | 1.58 (0.11) |
| gemini-2.5-flash | In-Context Regeneration (General) | 1.26 (0.11) | 4.94 (0.06) | 5.00 (0.00) | 1.45 (0.12) | 1.60 (0.14) | 1.26 (0.18) | 1.87 (0.21) | 2.81 (0.20) |
| gemini-2.5-flash | In-Context Regeneration (Task-Anchored) | 1.06 (0.06) | 4.94 (0.06) | 5.00 (0.00) | 1.65 (0.11) | 3.12 (0.19) | 1.65 (0.26) | 3.11 (0.23) | 3.06 (0.19) |
| gemini-2.5-flash | System Prompt (General) | 1.15 (0.11) | 4.94 (0.06) | 5.00 (0.00) | 1.20 (0.08) | 2.82 (0.21) | 1.96 (0.32) | 2.36 (0.25) | 3.53 (0.19) |
| gemini-2.5-flash | System Prompt (Task-Anchored) | 1.06 (0.06) | 4.94 (0.06) | 5.00 (0.00) | 1.62 (0.14) | 4.06 (0.17) | 2.96 (0.39) | 3.96 (0.18) | 4.02 (0.16) |

Table 31: Checklist-Based Quality by Model, Sampling Strategy, and Task Category.

(Using Only GPT-4o as the Checklist-Based Quality Judge)

| Model | Sampling Strategy | A | B | C | D | E | F | G | H |
|---|---|---|---|---|---|---|---|---|---|
| gpt-4o | Temperature (t=0.0) | 3.70 (0.16) | 4.76 (0.19) | 4.63 (0.20) | 4.29 (0.12) | 4.78 (0.06) | 4.58 (0.19) | 4.93 (0.03) | 4.89 (0.04) |
| gpt-4o | Temperature (t=1.0) | 3.69 (0.15) | 4.78 (0.17) | 4.61 (0.22) | 4.27 (0.12) | 4.85 (0.04) | 4.61 (0.18) | 4.92 (0.03) | 4.90 (0.03) |
| gpt-4o | Temperature (t=2.0) | 3.59 (0.17) | 4.83 (0.12) | 4.56 (0.23) | 4.27 (0.13) | 4.81 (0.05) | 4.57 (0.19) | 4.87 (0.06) | 4.90 (0.03) |
| gpt-4o | In-Context Regeneration (General) | 3.31 (0.15) | 4.66 (0.19) | 4.26 (0.27) | 4.20 (0.14) | 4.69 (0.07) | 4.45 (0.20) | 4.84 (0.06) | 4.36 (0.09) |
| gpt-4o | In-Context Regeneration (Task-Anchored) | 3.40 (0.16) | 4.70 (0.18) | 4.30 (0.28) | 4.24 (0.12) | 4.48 (0.10) | 4.37 (0.20) | 4.81 (0.06) | 4.42 (0.08) |
| gpt-4o | System Prompt (General) | 3.56 (0.17) | 4.69 (0.16) | 4.20 (0.29) | 3.98 (0.17) | 4.37 (0.09) | 4.23 (0.18) | 4.84 (0.06) | 4.64 (0.06) |
| gpt-4o | System Prompt (Task-Anchored) | 3.32 (0.16) | 4.70 (0.19) | 4.19 (0.31) | 3.76 (0.16) | 4.09 (0.09) | 3.72 (0.21) | 4.72 (0.08) | 4.44 (0.07) |
| claude-4-sonnet | Temperature (t=0.0) | 3.27 (0.15) | 4.92 (0.06) | 4.00 (0.36) | 4.33 (0.14) | 4.87 (0.04) | 4.67 (0.16) | 4.72 (0.11) | 4.95 (0.03) |
| claude-4-sonnet | Temperature (t=0.5) | 3.31 (0.14) | 4.90 (0.07) | 4.10 (0.33) | 4.22 (0.15) | 4.81 (0.05) | 4.65 (0.15) | 4.74 (0.12) | 4.94 (0.03) |
| claude-4-sonnet | Temperature (t=1.0) | 3.33 (0.14) | 4.88 (0.09) | 4.17 (0.31) | 4.34 (0.14) | 4.82 (0.05) | 4.63 (0.17) | 4.79 (0.10) | 4.93 (0.03) |
| claude-4-sonnet | In-Context Regeneration (General) | 3.40 (0.14) | 4.55 (0.19) | 4.16 (0.33) | 4.27 (0.14) | 4.72 (0.07) | 4.57 (0.15) | 4.80 (0.10) | 4.85 (0.04) |
| claude-4-sonnet | In-Context Regeneration (Task-Anchored) | 3.30 (0.15) | 4.53 (0.19) | 4.30 (0.25) | 3.88 (0.14) | 4.39 (0.08) | 4.50 (0.14) | 4.61 (0.10) | 4.61 (0.06) |
| claude-4-sonnet | System Prompt (General) | 3.37 (0.17) | 4.58 (0.13) | 4.30 (0.26) | 4.34 (0.12) | 4.51 (0.08) | 4.41 (0.20) | 4.72 (0.12) | 4.74 (0.05) |
| claude-4-sonnet | System Prompt (Task-Anchored) | 3.26 (0.18) | 4.59 (0.12) | 4.33 (0.27) | 4.16 (0.14) | 4.35 (0.09) | 4.04 (0.22) | 4.72 (0.08) | 4.54 (0.07) |
| gemini-2.5-flash | Temperature (t=0.0) | 3.04 (0.16) | 4.79 (0.14) | 4.31 (0.31) | 4.10 (0.15) | 4.83 (0.05) | 4.46 (0.21) | 4.91 (0.06) | 4.66 (0.08) |
| gemini-2.5-flash | Temperature (t=1.0) | 2.98 (0.16) | 4.83 (0.10) | 4.36 (0.24) | 4.03 (0.15) | 4.88 (0.04) | 4.50 (0.18) | 4.81 (0.07) | 4.67 (0.07) |
| gemini-2.5-flash | Temperature (t=2.0) | 2.97 (0.15) | 4.86 (0.10) | 4.33 (0.24) | 4.04 (0.15) | 4.83 (0.05) | 4.46 (0.16) | 4.86 (0.05) | 4.62 (0.08) |
| gemini-2.5-flash | In-Context Regeneration (General) | 2.91 (0.16) | 4.75 (0.16) | 4.13 (0.27) | 4.11 (0.14) | 4.85 (0.05) | 4.55 (0.16) | 4.80 (0.09) | 4.48 (0.08) |
| gemini-2.5-flash | In-Context Regeneration (Task-Anchored) | 2.87 (0.16) | 4.79 (0.16) | 4.31 (0.25) | 3.93 (0.14) | 4.79 (0.06) | 4.37 (0.18) | 4.61 (0.10) | 4.53 (0.08) |
| gemini-2.5-flash | System Prompt (General) | 3.35 (0.17) | 4.74 (0.15) | 4.09 (0.25) | 4.41 (0.11) | 4.65 (0.07) | 4.36 (0.21) | 4.82 (0.07) | 4.67 (0.05) |
| gemini-2.5-flash | System Prompt (Task-Anchored) | 3.14 (0.16) | 4.78 (0.17) | 4.40 (0.23) | 4.24 (0.14) | 4.68 (0.06) | 3.80 (0.25) | 4.56 (0.10) | 4.60 (0.06) |

Table 32: Checklist-Based Quality by Model, Sampling Strategy, and Task Category.

| Model | Sampling Strategy | A | B | C | D | E | F | G | H |
|---|---|---|---|---|---|---|---|---|---|
| | (Using Only Claude-4-Sonnet as the Checklist-Based Quality Judge) | | | | | | | | |
| gpt-4o | Temperature (t=0.0) | 3.37 (0.17) | 4.66 (0.14) | 4.59 (0.22) | 3.80 (0.17) | 4.44 (0.08) | 4.23 (0.23) | 4.54 (0.10) | 4.63 (0.07) |
| gpt-4o | Temperature (t=1.0) | 3.37 (0.17) | 4.66 (0.16) | 4.61 (0.20) | 3.75 (0.17) | 4.46 (0.08) | 4.18 (0.24) | 4.60 (0.09) | 4.62 (0.06) |
| gpt-4o | Temperature (t=2.0) | 3.28 (0.16) | 4.69 (0.12) | 4.54 (0.24) | 3.74 (0.18) | 4.46 (0.08) | 4.19 (0.24) | 4.60 (0.09) | 4.62 (0.06) |
| gpt-4o | In-Context Regeneration (General) | 2.93 (0.16) | 4.83 (0.09) | 4.30 (0.27) | 3.55 (0.17) | 4.15 (0.09) | 3.73 (0.24) | 4.43 (0.11) | 3.79 (0.11) |
| gpt-4o | In-Context Regeneration (Task-Anchored) | 3.14 (0.17) | 4.90 (0.05) | 4.33 (0.26) | 3.72 (0.16) | 4.02 (0.11) | 3.71 (0.24) | 4.40 (0.10) | 3.76 (0.11) |
| gpt-4o | System Prompt (General) | 3.06 (0.17) | 4.45 (0.14) | 4.30 (0.36) | 3.54 (0.18) | 3.65 (0.10) | 3.38 (0.24) | 4.56 (0.09) | 3.97 (0.10) |
| gpt-4o | System Prompt (Task-Anchored) | 3.16 (0.17) | 4.70 (0.12) | 4.30 (0.34) | 3.16 (0.18) | 3.36 (0.11) | 2.97 (0.23) | 4.19 (0.13) | 3.80 (0.11) |
| claude-4-sonnet | Temperature (t=0.0) | 3.11 (0.16) | 4.65 (0.12) | 4.30 (0.38) | 4.19 (0.15) | 4.71 (0.07) | 4.27 (0.23) | 4.58 (0.12) | 4.72 (0.06) |
| claude-4-sonnet | Temperature (t=0.5) | 3.18 (0.15) | 4.61 (0.12) | 4.37 (0.34) | 4.18 (0.15) | 4.64 (0.06) | 4.35 (0.20) | 4.62 (0.12) | 4.74 (0.05) |
| claude-4-sonnet | Temperature (t=1.0) | 3.23 (0.15) | 4.60 (0.12) | 4.40 (0.32) | 4.24 (0.15) | 4.66 (0.06) | 4.34 (0.22) | 4.64 (0.11) | 4.71 (0.05) |
| claude-4-sonnet | In-Context Regeneration (General) | 3.13 (0.15) | 4.67 (0.11) | 4.50 (0.29) | 4.18 (0.14) | 4.49 (0.08) | 4.13 (0.21) | 4.60 (0.12) | 4.47 (0.07) |
| claude-4-sonnet | In-Context Regeneration (Task-Anchored) | 3.26 (0.16) | 4.71 (0.12) | 4.61 (0.22) | 3.80 (0.13) | 4.11 (0.09) | 3.87 (0.23) | 4.46 (0.10) | 4.15 (0.09) |
| claude-4-sonnet | System Prompt (General) | 3.10 (0.16) | 4.29 (0.16) | 4.56 (0.24) | 4.17 (0.14) | 3.95 (0.08) | 3.93 (0.25) | 4.52 (0.11) | 4.30 (0.08) |
| claude-4-sonnet | System Prompt (Task-Anchored) | 3.21 (0.16) | 4.22 (0.15) | 4.39 (0.27) | 3.99 (0.13) | 3.75 (0.10) | 3.48 (0.22) | 4.56 (0.08) | 3.93 (0.10) |
| gemini-2.5-flash | Temperature (t=0.0) | 2.99 (0.18) | 4.64 (0.20) | 4.31 (0.24) | 3.94 (0.16) | 4.65 (0.08) | 3.95 (0.28) | 4.65 (0.08) | 4.33 (0.10) |
| gemini-2.5-flash | Temperature (t=1.0) | 3.02 (0.16) | 4.74 (0.12) | 4.47 (0.13) | 3.88 (0.16) | 4.64 (0.08) | 3.94 (0.25) | 4.62 (0.08) | 4.34 (0.09) |
| gemini-2.5-flash | Temperature (t=2.0) | 3.00 (0.16) | 4.83 (0.11) | 4.51 (0.16) | 3.83 (0.16) | 4.63 (0.07) | 3.85 (0.25) | 4.56 (0.08) | 4.34 (0.09) |
| gemini-2.5-flash | In-Context Regeneration (General) | 2.68 (0.15) | 4.81 (0.11) | 4.20 (0.26) | 3.93 (0.16) | 4.66 (0.07) | 3.70 (0.26) | 4.57 (0.11) | 4.03 (0.11) |
| gemini-2.5-flash | In-Context Regeneration (Task-Anchored) | 2.79 (0.15) | 4.95 (0.04) | 4.37 (0.22) | 3.68 (0.15) | 4.52 (0.09) | 3.68 (0.25) | 4.41 (0.08) | 3.98 (0.10) |
| gemini-2.5-flash | System Prompt (General) | 3.03 (0.14) | 4.85 (0.08) | 4.06 (0.33) | 4.23 (0.13) | 4.11 (0.10) | 3.46 (0.26) | 4.52 (0.10) | 4.13 (0.09) |
| gemini-2.5-flash | System Prompt (Task-Anchored) | 2.95 (0.16) | 4.96 (0.04) | 4.14 (0.27) | 4.04 (0.12) | 4.06 (0.10) | 3.06 (0.24) | 4.28 (0.13) | 3.93 (0.10) |

Table 33: Checklist-Based Quality by Model, Sampling Strategy, and Task Category.

(Using Only Gemini-2.5-Flash as the Checklist-Based Quality Judge)

| Model | Sampling Strategy | A | B | C | D | E | F | G | H |
|---|---|---|---|---|---|---|---|---|---|
| gpt-4o | Temperature (t=0.0) | 4.21 (0.14) | 4.41 (0.25) | 4.63 (0.20) | 3.89 (0.19) | 4.64 (0.09) | 4.38 (0.22) | 4.80 (0.07) | 4.82 (0.06) |
| gpt-4o | Temperature (t=1.0) | 4.18 (0.14) | 4.50 (0.23) | 4.50 (0.22) | 3.91 (0.19) | 4.66 (0.09) | 4.44 (0.21) | 4.82 (0.06) | 4.83 (0.05) |
| gpt-4o | Temperature (t=2.0) | 4.10 (0.14) | 4.67 (0.15) | 4.46 (0.23) | 3.94 (0.20) | 4.62 (0.08) | 4.34 (0.21) | 4.74 (0.08) | 4.79 (0.05) |
| gpt-4o | In-Context Regeneration (General) | 3.89 (0.18) | 4.65 (0.18) | 4.36 (0.31) | 4.01 (0.18) | 4.41 (0.10) | 4.16 (0.22) | 4.72 (0.09) | 4.26 (0.10) |
| gpt-4o | In-Context Regeneration (Task-Anchored) | 4.09 (0.15) | 4.84 (0.07) | 4.36 (0.30) | 3.96 (0.17) | 4.28 (0.12) | 3.91 (0.23) | 4.68 (0.09) | 4.20 (0.10) |
| gpt-4o | System Prompt (General) | 3.95 (0.16) | 4.55 (0.18) | 4.39 (0.30) | 3.95 (0.20) | 4.00 (0.13) | 3.83 (0.26) | 4.71 (0.08) | 4.42 (0.08) |
| gpt-4o | System Prompt (Task-Anchored) | 3.97 (0.17) | 4.75 (0.16) | 4.47 (0.30) | 3.52 (0.20) | 3.52 (0.12) | 3.49 (0.29) | 4.45 (0.11) | 4.11 (0.10) |
| claude-4-sonnet | Temperature (t=0.0) | 2.78 (0.22) | 4.51 (0.26) | 4.07 (0.37) | 4.34 (0.15) | 4.80 (0.06) | 4.36 (0.20) | 4.65 (0.13) | 4.90 (0.04) |
| claude-4-sonnet | Temperature (t=0.5) | 2.78 (0.22) | 4.54 (0.25) | 4.09 (0.36) | 4.33 (0.14) | 4.80 (0.08) | 4.34 (0.21) | 4.70 (0.13) | 4.91 (0.04) |
| claude-4-sonnet | Temperature (t=1.0) | 2.72 (0.22) | 4.55 (0.25) | 4.13 (0.34) | 4.41 (0.13) | 4.80 (0.08) | 4.37 (0.20) | 4.75 (0.10) | 4.91 (0.04) |
| claude-4-sonnet | In-Context Regeneration (General) | 3.06 (0.18) | 4.66 (0.18) | 4.33 (0.27) | 4.45 (0.13) | 4.57 (0.10) | 4.37 (0.20) | 4.77 (0.12) | 4.77 (0.05) |
| claude-4-sonnet | In-Context Regeneration (Task-Anchored) | 2.87 (0.21) | 4.64 (0.19) | 4.67 (0.22) | 4.12 (0.12) | 4.16 (0.10) | 4.11 (0.19) | 4.51 (0.12) | 4.38 (0.08) |
| claude-4-sonnet | System Prompt (General) | 2.96 (0.21) | 4.42 (0.23) | 4.44 (0.22) | 4.37 (0.15) | 4.21 (0.11) | 4.18 (0.24) | 4.57 (0.14) | 4.57 (0.07) |
| claude-4-sonnet | System Prompt (Task-Anchored) | 3.30 (0.20) | 4.22 (0.24) | 4.39 (0.23) | 4.36 (0.12) | 3.98 (0.11) | 3.83 (0.24) | 4.52 (0.09) | 4.25 (0.08) |
| gemini-2.5-flash | Temperature (t=0.0) | 4.33 (0.16) | 4.69 (0.18) | 4.41 (0.30) | 4.35 (0.15) | 4.92 (0.04) | 4.50 (0.17) | 4.87 (0.06) | 4.70 (0.09) |
| gemini-2.5-flash | Temperature (t=1.0) | 4.12 (0.16) | 4.88 (0.09) | 4.41 (0.22) | 4.29 (0.15) | 4.91 (0.03) | 4.54 (0.16) | 4.77 (0.08) | 4.69 (0.08) |
| gemini-2.5-flash | Temperature (t=2.0) | 4.09 (0.16) | 4.85 (0.08) | 4.50 (0.20) | 4.28 (0.16) | 4.92 (0.03) | 4.38 (0.17) | 4.78 (0.06) | 4.67 (0.08) |
| gemini-2.5-flash | In-Context Regeneration (General) | 3.81 (0.18) | 4.89 (0.05) | 4.44 (0.26) | 4.36 (0.15) | 4.88 (0.05) | 4.37 (0.17) | 4.71 (0.11) | 4.47 (0.09) |
| gemini-2.5-flash | In-Context Regeneration (Task-Anchored) | 4.27 (0.15) | 4.91 (0.09) | 4.61 (0.18) | 4.17 (0.15) | 4.79 (0.06) | 4.30 (0.17) | 4.54 (0.10) | 4.46 (0.08) |
| gemini-2.5-flash | System Prompt (General) | 3.96 (0.18) | 4.86 (0.08) | 4.50 (0.23) | 4.63 (0.11) | 4.50 (0.10) | 4.04 (0.23) | 4.71 (0.08) | 4.53 (0.07) |
| gemini-2.5-flash | System Prompt (Task-Anchored) | 4.11 (0.17) | 4.94 (0.04) | 4.46 (0.22) | 4.50 (0.12) | 4.47 (0.08) | 3.62 (0.25) | 4.40 (0.12) | 4.26 (0.08) |

Table 34: # of Functionally Diverse Responses by Model, Sampling Strategy, and Task Category.

(Based on $n = 10$ generated responses. Using Only GPT-4o as the Functional Diversity Judge)

| Model | Sampling Strategy | A | B | C | D | E | F | G | H |
|---|---|---|---|---|---|---|---|---|---|
| gpt-4o | In-Context Regeneration (General) | 3.58 (0.43) | 9.06 (0.62) | 10.00 (0.00) | 1.24 (0.17) | 1.86 (0.22) | 2.30 (0.61) | 4.02 (0.54) | 5.12 (0.46) |
| gpt-4o | In-Context Regeneration (Task-Anchored) | 1.06 (0.03) | 9.88 (0.09) | 10.00 (0.00) | 1.07 (0.04) | 2.72 (0.30) | 2.57 (0.66) | 6.11 (0.57) | 5.57 (0.45) |
| gpt-4o | System Prompt (General) | 2.74 (0.49) | 9.56 (0.26) | 10.00 (0.00) | 1.26 (0.16) | 6.20 (0.33) | 3.78 (0.83) | 5.76 (0.61) | 7.01 (0.42) |
| gpt-4o | System Prompt (Task-Anchored) | 1.00 (0.00) | 10.00 (0.00) | 10.00 (0.00) | 1.87 (0.20) | 7.70 (0.27) | 6.05 (0.86) | 8.13 (0.48) | 8.10 (0.32) |
| claude-4-sonnet | In-Context Regeneration (General) | 2.13 (0.29) | 9.56 (0.27) | 9.07 (0.68) | 1.24 (0.12) | 6.04 (0.45) | 2.30 (0.59) | 6.20 (0.58) | 5.73 (0.43) |
| claude-4-sonnet | In-Context Regeneration (Task-Anchored) | 1.19 (0.17) | 9.81 (0.19) | 9.86 (0.14) | 2.31 (0.27) | 9.40 (0.12) | 4.61 (0.74) | 8.98 (0.32) | 7.32 (0.35) |
| claude-4-sonnet | System Prompt (General) | 1.34 (0.23) | 9.44 (0.35) | 9.36 (0.64) | 1.38 (0.20) | 6.68 (0.44) | 2.74 (0.73) | 6.77 (0.58) | 7.01 (0.41) |
| claude-4-sonnet | System Prompt (Task-Anchored) | 1.17 (0.17) | 9.62 (0.31) | 10.00 (0.00) | 2.12 (0.32) | 9.24 (0.19) | 5.57 (0.85) | 8.68 (0.42) | 8.53 (0.28) |
| gemini-2.5-flash | In-Context Regeneration (General) | 1.58 (0.28) | 9.69 (0.20) | 9.29 (0.64) | 1.16 (0.09) | 1.86 (0.31) | 1.52 (0.41) | 4.51 (0.60) | 5.16 (0.45) |
| gemini-2.5-flash | In-Context Regeneration (Task-Anchored) | 1.21 (0.17) | 9.75 (0.19) | 9.93 (0.07) | 1.27 (0.12) | 3.12 (0.35) | 1.87 (0.54) | 5.96 (0.55) | 5.66 (0.43) |
| gemini-2.5-flash | System Prompt (General) | 1.04 (0.03) | 9.44 (0.33) | 9.36 (0.64) | 1.11 (0.09) | 3.86 (0.38) | 3.91 (0.78) | 6.02 (0.58) | 7.06 (0.40) |
| gemini-2.5-flash | System Prompt (Task-Anchored) | 1.15 (0.15) | 9.69 (0.25) | 10.00 (0.00) | 1.63 (0.22) | 6.92 (0.33) | 5.43 (0.75) | 8.31 (0.45) | 8.05 (0.33) |

Table 35: Checklist-Based Quality by Model, Sampling Strategy, and Task Category.

(Based on $n = 10$ generated responses. Using Only GPT-4o as the Checklist-Based Quality Judge.)

| Model | Sampling Strategy | A | B | C | D | E | F | G | H |
|---|---|---|---|---|---|---|---|---|---|
| gpt-4o | In-Context Regeneration (General) | 3.21 (0.14) | 4.63 (0.19) | 4.26 (0.29) | 3.89 (0.15) | 4.61 (0.07) | 4.40 (0.19) | 4.92 (0.02) | 4.33 (0.10) |
| gpt-4o | In-Context Regeneration (Task-Anchored) | 3.47 (0.16) | 4.44 (0.25) | 4.23 (0.28) | 4.27 (0.12) | 4.55 (0.09) | 4.22 (0.19) | 4.78 (0.06) | 4.31 (0.09) |
| gpt-4o | System Prompt (General) | 3.62 (0.16) | 4.59 (0.18) | 4.22 (0.29) | 3.60 (0.19) | 3.82 (0.10) | 3.77 (0.20) | 4.64 (0.09) | 4.35 (0.08) |
| gpt-4o | System Prompt (Task-Anchored) | 3.45 (0.16) | 4.53 (0.21) | 4.19 (0.29) | 3.33 (0.16) | 3.71 (0.11) | 3.23 (0.18) | 4.56 (0.11) | 4.06 (0.10) |
| claude-4-sonnet | In-Context Regeneration (General) | 3.32 (0.14) | 4.54 (0.19) | 4.17 (0.33) | 4.24 (0.13) | 4.46 (0.09) | 4.43 (0.18) | 4.78 (0.10) | 4.75 (0.04) |
| claude-4-sonnet | In-Context Regeneration (Task-Anchored) | 3.30 (0.14) | 4.48 (0.19) | 4.35 (0.24) | 3.71 (0.13) | 4.12 (0.09) | 4.20 (0.18) | 4.39 (0.12) | 4.46 (0.07) |
| claude-4-sonnet | System Prompt (General) | 3.32 (0.17) | 4.64 (0.12) | 4.34 (0.24) | 4.09 (0.17) | 4.27 (0.09) | 4.16 (0.18) | 4.62 (0.13) | 4.56 (0.07) |
| claude-4-sonnet | System Prompt (Task-Anchored) | 3.30 (0.18) | 4.58 (0.10) | 4.38 (0.26) | 3.77 (0.16) | 3.93 (0.09) | 3.83 (0.19) | 4.30 (0.19) | 4.33 (0.08) |
| gemini-2.5-flash | In-Context Regeneration (General) | 2.88 (0.14) | 4.57 (0.18) | 4.31 (0.26) | 4.13 (0.14) | 4.84 (0.06) | 4.50 (0.19) | 4.74 (0.10) | 4.38 (0.09) |
| gemini-2.5-flash | In-Context Regeneration (Task-Anchored) | 2.92 (0.15) | 4.54 (0.19) | 4.19 (0.24) | 3.83 (0.14) | 4.76 (0.06) | 4.30 (0.22) | 4.56 (0.10) | 4.41 (0.08) |
| gemini-2.5-flash | System Prompt (General) | 3.16 (0.16) | 4.67 (0.12) | 4.33 (0.23) | 4.35 (0.12) | 4.49 (0.08) | 3.94 (0.22) | 4.78 (0.07) | 4.40 (0.07) |
| gemini-2.5-flash | System Prompt (Task-Anchored) | 2.91 (0.15) | 4.67 (0.19) | 4.13 (0.24) | 4.27 (0.13) | 4.53 (0.08) | 3.34 (0.20) | 4.55 (0.09) | 4.20 (0.09) |

Table 36: # of Functionally Diverse Responses by Model, Sampling Strategy, and Task Category.

| Model | Sampling Strategy | A | B | C | D | E | F | G | H |
|---|---|---|---|---|---|---|---|---|---|
| | (Using Only GPT-4o as the Functional Diversity Judge) | | | | | | | | |
| Llama-3.1-8B-Instruct | Temperature (t=0.1) | 1.08 (0.05) | 1.25 (0.14) | 1.71 (0.27) | 1.93 (0.18) | 1.56 (0.14) | 1.52 (0.23) | 1.47 (0.15) | 1.09 (0.04) |
| Llama-3.1-8B-Instruct | Temperature (t=0.5) | 1.19 (0.09) | 1.75 (0.25) | 2.79 (0.47) | 2.22 (0.18) | 2.02 (0.17) | 1.48 (0.24) | 1.91 (0.21) | 1.47 (0.10) |
| Llama-3.1-8B-Instruct | Temperature (t=1.0) | 1.45 (0.13) | 2.31 (0.30) | 3.71 (0.37) | 2.42 (0.20) | 2.68 (0.21) | 2.00 (0.29) | 2.51 (0.24) | 1.78 (0.13) |
| Online DPO (Wildchat, $\beta = 0.01$, Step 1000) | Temperature (t=0.1) | 1.72 (0.16) | 1.38 (0.20) | 2.00 (0.33) | 1.42 (0.11) | 1.66 (0.12) | 1.35 (0.18) | 1.47 (0.11) | 1.49 (0.09) |
| Online DPO (Wildchat, $\beta = 0.01$, Step 1000) | Temperature (t=0.5) | 3.09 (0.21) | 2.31 (0.30) | 4.00 (0.42) | 2.11 (0.18) | 3.26 (0.18) | 2.22 (0.31) | 2.53 (0.20) | 1.93 (0.14) |
| Online DPO (Wildchat, $\beta = 0.01$, Step 1000) | Temperature (t=1.0) | 3.43 (0.21) | 2.38 (0.30) | 4.71 (0.16) | 2.40 (0.19) | 3.60 (0.20) | 2.70 (0.34) | 3.31 (0.22) | 2.22 (0.16) |
| Online DPO (Wildchat, $\beta = 0.1$, Step 1000) | Temperature (t=0.1) | 1.19 (0.08) | 1.12 (0.09) | 1.64 (0.25) | 2.13 (0.16) | 1.46 (0.09) | 1.43 (0.23) | 2.11 (0.22) | 1.31 (0.07) |
| Online DPO (Wildchat, $\beta = 0.1$, Step 1000) | Temperature (t=0.5) | 1.43 (0.12) | 1.69 (0.27) | 2.43 (0.42) | 2.27 (0.18) | 2.38 (0.15) | 1.87 (0.26) | 2.71 (0.23) | 1.43 (0.08) |
| Online DPO (Wildchat, $\beta = 0.1$, Step 1000) | Temperature (t=1.0) | 1.64 (0.15) | 2.12 (0.26) | 2.86 (0.38) | 2.29 (0.18) | 2.52 (0.17) | 2.13 (0.31) | 2.89 (0.23) | 1.58 (0.10) |
| Online DPO (Ultrafeedback, $\beta = 0.01$, Step 1000) | Temperature (t=0.1) | 2.43 (0.23) | 1.31 (0.20) | 1.86 (0.38) | 1.05 (0.03) | 1.04 (0.03) | 1.48 (0.24) | 1.73 (0.19) | 1.25 (0.09) |
| Online DPO (Ultrafeedback, $\beta = 0.01$, Step 1000) | Temperature (t=0.5) | 2.13 (0.20) | 1.38 (0.22) | 2.14 (0.43) | 1.07 (0.06) | 1.00 (0.00) | 1.43 (0.23) | 1.58 (0.18) | 1.24 (0.09) |
| Online DPO (Ultrafeedback, $\beta = 0.01$, Step 1000) | Temperature (t=1.0) | 2.17 (0.20) | 1.62 (0.31) | 1.86 (0.39) | 1.07 (0.06) | 1.02 (0.02) | 1.52 (0.23) | 1.69 (0.20) | 1.23 (0.08) |
| GRPO (Wildchat, $\beta = 0.001$, Step 1000) | Temperature (t=0.1) | 2.79 (0.21) | 1.62 (0.24) | 2.71 (0.40) | 2.84 (0.19) | 1.12 (0.05) | 1.09 (0.06) | 1.22 (0.08) | 1.12 (0.04) |
| GRPO (Wildchat, $\beta = 0.001$, Step 1000) | Temperature (t=0.5) | 3.55 (0.19) | 2.31 (0.28) | 3.29 (0.34) | 3.13 (0.20) | 1.18 (0.07) | 1.17 (0.08) | 1.47 (0.15) | 1.27 (0.07) |
| GRPO (Wildchat, $\beta = 0.001$, Step 1000) | Temperature (t=1.0) | 4.00 (0.16) | 2.69 (0.35) | 3.93 (0.27) | 3.09 (0.20) | 1.16 (0.06) | 1.04 (0.04) | 1.67 (0.17) | 1.36 (0.10) |
| GRPO (Wildchat, $\beta = 0.01$, Step 1000) | Temperature (t=0.1) | 1.96 (0.15) | 1.88 (0.24) | 3.71 (0.29) | 2.25 (0.16) | 1.26 (0.08) | 1.09 (0.06) | 1.51 (0.15) | 1.26 (0.09) |
| GRPO (Wildchat, $\beta = 0.01$, Step 1000) | Temperature (t=0.5) | 2.19 (0.19) | 2.06 (0.30) | 3.86 (0.40) | 2.29 (0.18) | 1.62 (0.12) | 1.17 (0.10) | 1.71 (0.18) | 1.41 (0.10) |
| GRPO (Wildchat, $\beta = 0.01$, Step 1000) | Temperature (t=1.0) | 2.32 (0.19) | 2.56 (0.34) | 4.29 (0.24) | 2.38 (0.18) | 1.74 (0.15) | 1.35 (0.16) | 2.00 (0.20) | 1.42 (0.10) |
| GRPO (Ultrafeedback, $\beta = 0.001$, Step 1000) | Temperature (t=0.1) | 3.15 (0.19) | 1.69 (0.24) | 3.14 (0.42) | 2.25 (0.17) | 1.20 (0.06) | 1.52 (0.21) | 1.96 (0.20) | 1.24 (0.06) |
| GRPO (Ultrafeedback, $\beta = 0.001$, Step 1000) | Temperature (t=0.5) | 3.66 (0.20) | 2.56 (0.30) | 3.64 (0.32) | 2.73 (0.21) | 1.40 (0.11) | 1.87 (0.30) | 2.36 (0.22) | 1.45 (0.09) |
| GRPO (Ultrafeedback, $\beta = 0.001$, Step 1000) | Temperature (t=1.0) | 3.94 (0.18) | 2.75 (0.30) | 4.29 (0.27) | 2.67 (0.20) | 1.50 (0.13) | 1.83 (0.29) | 2.58 (0.23) | 1.51 (0.10) |

Table 37: # of Functionally Diverse Responses by Model, Sampling Strategy, and Task Category.

| Model | Sampling Strategy | A | B | C | D | E | F | G | H |
|---|---|---|---|---|---|---|---|---|---|
| | | (Using Only GPT-4o as the Functional Diversity Judge) | | | | | | | |
| Llama-3.1-8B-Instruct | Temperature (t=0.1) | 1.08 (0.05) | 1.25 (0.14) | 1.71 (0.27) | 1.93 (0.18) | 1.56 (0.14) | 1.52 (0.23) | 1.47 (0.15) | 1.09 (0.04) |
| Llama-3.1-8B-Instruct | Temperature (t=0.5) | 1.19 (0.09) | 1.75 (0.25) | 2.79 (0.47) | 2.22 (0.18) | 2.02 (0.17) | 1.48 (0.24) | 1.91 (0.21) | 1.47 (0.10) |
| Llama-3.1-8B-Instruct | Temperature (t=1.0) | 1.45 (0.13) | 2.31 (0.30) | 3.71 (0.37) | 2.42 (0.20) | 2.68 (0.21) | 2.00 (0.29) | 2.51 (0.24) | 1.78 (0.13) |
| GRPO (Wildchat, $\beta = 0.001$, Step 1000) | Temperature (t=0.1) | 2.79 (0.21) | 1.62 (0.24) | 2.71 (0.40) | 2.84 (0.19) | 1.12 (0.05) | 1.09 (0.06) | 1.22 (0.08) | 1.12 (0.04) |
| GRPO (Wildchat, $\beta = 0.001$, Step 1000) | Temperature (t=0.5) | 3.55 (0.19) | 2.31 (0.28) | 3.29 (0.34) | 3.13 (0.20) | 1.18 (0.07) | 1.17 (0.08) | 1.47 (0.15) | 1.27 (0.07) |
| GRPO (Wildchat, $\beta = 0.001$, Step 1000) | Temperature (t=1.0) | 4.00 (0.16) | 2.69 (0.35) | 3.93 (0.27) | 3.09 (0.20) | 1.16 (0.06) | 1.04 (0.04) | 1.67 (0.17) | 1.36 (0.10) |
| GRPO (Ultrafeedback, $\beta = 0.001$, Step 1000) | Temperature (t=0.1) | 3.15 (0.19) | 1.69 (0.24) | 3.14 (0.42) | 2.25 (0.17) | 1.20 (0.06) | 1.52 (0.20) | 1.96 (0.20) | 1.24 (0.06) |
| GRPO (Ultrafeedback, $\beta = 0.001$, Step 1000) | Temperature (t=0.5) | 3.66 (0.20) | 2.56 (0.30) | 3.64 (0.32) | 2.73 (0.21) | 1.40 (0.11) | 1.87 (0.30) | 2.36 (0.22) | 1.45 (0.09) |
| GRPO (Ultrafeedback, $\beta = 0.001$, Step 1000) | Temperature (t=1.0) | 3.94 (0.18) | 2.75 (0.30) | 4.29 (0.27) | 2.67 (0.20) | 1.50 (0.13) | 1.83 (0.29) | 2.58 (0.23) | 1.51 (0.10) |
| GRPO w/DARLING (Wildchat, $\beta = 0.001$, Step 1000) | Temperature (t=0.1) | 3.47 (0.19) | 1.69 (0.20) | 3.14 (0.44) | 2.60 (0.20) | 1.70 (0.15) | 1.83 (0.28) | 2.11 (0.23) | 1.35 (0.08) |
| GRPO w/DARLING (Wildchat, $\beta = 0.001$, Step 1000) | Temperature (t=0.5) | 3.94 (0.16) | 2.25 (0.25) | 3.29 (0.40) | 2.75 (0.19) | 2.06 (0.14) | 2.22 (0.32) | 2.82 (0.23) | 1.64 (0.12) |
| GRPO w/DARLING (Wildchat, $\beta = 0.001$, Step 1000) | Temperature (t=1.0) | 4.26 (0.14) | 3.19 (0.29) | 3.64 (0.41) | 2.65 (0.19) | 2.52 (0.19) | 2.13 (0.32) | 3.53 (0.22) | 1.89 (0.14) |
| GRPO w/DARLING (Ultrafeedback, $\beta = 0.001$, Step 1000) | Temperature (t=0.1) | 2.43 (0.18) | 1.19 (0.14) | 2.29 (0.42) | 2.29 (0.18) | 1.50 (0.12) | 1.26 (0.13) | 1.62 (0.16) | 1.10 (0.04) |
| GRPO w/DARLING (Ultrafeedback, $\beta = 0.001$, Step 1000) | Temperature (t=0.5) | 3.42 (0.20) | 1.56 (0.27) | 2.86 (0.47) | 2.38 (0.18) | 1.58 (0.13) | 1.83 (0.31) | 2.13 (0.23) | 1.28 (0.07) |
| GRPO w/DARLING (Ultrafeedback, $\beta = 0.001$, Step 1000) | Temperature (t=1.0) | 3.53 (0.20) | 1.75 (0.27) | 3.14 (0.40) | 2.25 (0.19) | 1.52 (0.12) | 1.83 (0.29) | 2.16 (0.22) | 1.33 (0.08) |
| GRPO w/DARLING & Task Diversity Judges (Wildchat, $\beta = 0.001$, Step 1000) | Temperature (t=0.1) | 2.58 (0.19) | 1.19 (0.10) | 2.21 (0.45) | 3.07 (0.19) | 3.10 (0.19) | 2.70 (0.33) | 3.44 (0.23) | 2.11 (0.14) |
| GRPO w/DARLING & Task Diversity Judges (Wildchat, $\beta = 0.001$, Step 1000) | Temperature (t=0.5) | 3.42 (0.19) | 1.88 (0.26) | 2.43 (0.45) | 3.29 (0.20) | 3.98 (0.17) | 3.43 (0.35) | 3.93 (0.22) | 2.92 (0.16) |
| GRPO w/DARLING & Task Diversity Judges (Wildchat, $\beta = 0.001$, Step 1000) | Temperature (t=1.0) | 3.89 (0.18) | 2.25 (0.32) | 3.21 (0.38) | 3.47 (0.17) | 4.46 (0.13) | 4.00 (0.29) | 4.36 (0.19) | 3.32 (0.16) |
| GRPO w/DARLING & Task Diversity Judges (Ultrachat, $\beta = 0.001$, Step 1000) | Temperature (t=0.1) | 1.87 (0.16) | 1.25 (0.11) | 1.21 (0.15) | 1.95 (0.17) | 1.78 (0.17) | 2.00 (0.31) | 2.22 (0.22) | 1.33 (0.08) |
| GRPO w/DARLING & Task Diversity Judges (Ultrachat, $\beta = 0.001$, Step 1000) | Temperature (t=0.5) | 3.23 (0.21) | 1.62 (0.27) | 2.07 (0.37) | 2.44 (0.18) | 2.66 (0.18) | 2.87 (0.36) | 2.89 (0.24) | 1.81 (0.12) |
| GRPO w/DARLING & Task Diversity Judges (Ultrachat, $\beta = 0.001$, Step 1000) | Temperature (t=1.0) | 3.68 (0.21) | 2.31 (0.28) | 3.14 (0.40) | 2.65 (0.19) | 3.14 (0.21) | 3.17 (0.35) | 3.84 (0.24) | 2.43 (0.16) |

