# OpenReview forum: "LLM Output Homogenization is Task Dependent"
_ICLR.cc/2026/Conference — Submitted to ICLR 2026_

### Official Review · Reviewer_Qpei · 2025-10-17

**Soundness:** 2
**Presentation:** 3
**Contribution:** 2
**Rating:** 4
**Confidence:** 4

**Summary:**

The paper describes work on proposing a task-specific perspective for evaluating and mitigating output homogenization in LLMs. The authors define output homogenization where LLMs produce similar outputs which can be problematic in scenarios that require high diversity (e.g., creative writing) but in ones that require verifiability (e.g, math problems), hence the need for a task-dependent approach. The authors introduce the notion of “functional diversity” where a user might find two or more responses to be useful based on the task they use an LLM for. The authors then propose a simple task-anchored sampling technique aimed to maximize functional diversity for task categories where homogenization is needed as well for tasks where it is not. The authors lay down an experiment setup that mainly uses commercial models (Claude, GPT-4o, Gemini) for both generation and response evaluation and sample prompts from five existing datasets. In terms of measuring functional diversity, results show that task-anchored sampling performs better than general sampling techniques and oftentimes reduces homogenization for crucial categories such as Encyclopedia Inquiry, Creative Writing, and Advice or Opinions. In terms of diversity-quality tradeoff, using task-anchored functional diversity removes said tradeoff which is evident if general diversity metrics are used such as vocabulary diversity.

**Strengths:**

The concept of output homogenization and its inherent nature of being both essential and to be mitigated based on task is an important study. One strength I found with the paper is how well the motivation for the work is laid out that anyone can readily appreciate and understand the problem that needs to be solved. I also appreciate the simplicity with the task taxonomy and task-anchored sampling technique that the authors propose in order to increase functional diversity from LLMs. I agree that the proposed method’s simplificty will be key in the ease of adoption of future work with output homogenization.

**Weaknesses:**

My main issue with the current study is the LLM-as-a-judge-centric evaluation for the concept of diversity in the experiments. The authors define functional diversity as “a user would perceive two responses as meaningfully different for a given task”, hence, I was fully expecting human evaluation as the main driver of the evaluation procedures across the task taxonomy rather than LLM-as-a-judge.

Another issue I have is why did the authors not explore any open-weight models for both the functional diversity and diversity-quality tradeoff experiments and only used commercial ones? This reduces the reproducibility of the work that could continue to on focusing "why" (factors affecting) output homogenization happens as future work. The authors also mentioned that for all LLM-as-judge setups, the same models (GPT4o, Clause, GeminiFlash) are essentially evaluating their own responses even if they are averaged together. This setup sounds like bias may be introduced in the evaluation procedure of the study. What is the justification of the authors for this?

The authors mentioned that the work does not focus on answering the "why" output homogenization happens but the current results of the paper feels very limited and minimal at its current state. I would have appreciated it if the authors could have linked variations of functional diversity to factors such as model scale/size, language coverage, effect of preference optimization vs no preference optimization, etc. Especially that some of these factors like RLHF-ed models have been investigated by previous works to negatively affect output diversity. It would have been supplemental if the authors also investigated the same direction with the functional diversity whether this changes for non RHLF-ed models.

The paper should provide a clear disclaimer that this framework has only been tested with English-centric evaluation procedures. The results related to model output homogenization specifically functional diversity and diversity tradeoffs do not generalize in multilingual models unless otherwise tested sufficiently with the same level of rigor done for English.

**Questions:**

What are the justification of the authors for exclusively using commercial models and not open-source/open-weight models?

Were the prompts in Table 3 for each dataset and task category randomly sampled or handpicked?

---

> ### Author Response · Authors · 2025-11-20
> **Response to Reviewer Qpei**
>
> We thank the reviewer for deeply engaging with the work. We really appreciate your time, and are encouraged that you have underlined the main contributions and strengths of the work. Namely, the review highlights “how well the motivation for the work is laid out” and notes “I also appreciate the simplicity with the task taxonomy and task-anchored sampling technique”.
>
> The following comments address each of the reviewers' questions and concerns. Given our responses, we kindly ask that the reviewer consider increasing their score.

---

> ### Author Response · Authors · 2025-11-20
> **Response to "LLM-as-a-judge centric evaluation"**
>
> We agree that a limitation of our experiments is that we rely on LLM-judges to evaluate functional diversity. We have updated the Discussion section (page 9, lines 475-480) to clearly acknowledge this limitation and to cite relevant work that documents reliability concerns with LLM-judges. However, we want to highlight that 1) we did validate the functional diversity judges with human annotation (page 6, footnote 2), and that 2) we always report the average of 3 LLM-judges to increase reliability.
>
>
> Furthermore, we have clarified our definition of functional diversity as to not anchor on user perception given that this is not a user study. In our revised paper (lines 177-180), we now say that previous works define functional diversity based on user perception, but that we anchor on “meaningful differences” based on our task categories. We agree that there is an opportunity for future work to do user studies building on the present paper, and have added this point to our Discussion (lines 475-480)).

---

> ### Author Response · Authors · 2025-11-20
> **Response to "for all LLM-judge setups, the same models are essentially evaluating their own responses"**
>
> For the LLM-judge setup, we want to clarify that the generation and judgement processes are independent, and that human validation is used to cross-check the judges. We primarily report the average of 3 different LLM-judges to increase reliability.
>
> In our revised paper, we have further added results based on each LLM-judge separately (Appendix Tables 28-33). The average differences across individual judges are small (3% for functional diversity and 5% for checklist-based quality). Thus, our overall conclusions do not change when calculating metrics without “self-judging” where only different models evaluate responses.

---

> ### Author Response · Authors · 2025-11-20
> **Response to "why did the authors not explore any open-weight models"**
>
> We appreciate the suggestion to add experiments with open-weight models! We have added results for Llama-3.1-8B-Instruct and Mistral-7B-Instruct-v0.3. Please see our comment titled “Evaluating Open-Weight Models” in our Summary of Revisions. In general, we find that our task-anchored sampling is able to promote functional diversity in these open-weight LLMs as well.

---

> ### Author Response · Authors · 2025-11-20
> **Response to "the work does not focus on answering the "why" output homogenization happens"**
>
> We agree this would be interesting to test! Since we added results for Llama-3.1-8B-Instruct and Mistral-7B-Instruct-v0.3, we plan to use these models to add preliminary alignment experiments in the Appendix (since they are smaller models and feasible to fine-tune). So far, we have observed that functional diversity does not collapse for Llama-3.1-8B during DPO and GRPO (although the logit entropy decreases significantly). We will clarify in the main paper that this could be an interesting direction for future work!

---

> ### Author Response · Authors · 2025-11-20
> **Response to "English-centric evaluation procedures"**
>
> Thank you for raising the point about our English-centric evaluation procedures. We have added a clear disclaimer in the Discussion (page 9, line 470).

---

> ### Author Response · Authors · 2025-11-20
> **Response to Reviewer Questions**
>
> 1. We have added results for open-weight models! Please see the comment titled “Evaluating Open-Weight Models” in our summary of revisions.
>
>
> 2. The prompt examples in Table 3 were handpicked to be illustrative of tasks for that category. However, all the prompts/datasets we used in our evaluation are publicly available, and we have added a column in Table 3 to reference “Example Datasets” for each category.

---

> ### Comment · Reviewer_Qpei · 2025-11-24
> **Response acknowledged**
>
> This is to confirm that I have read the authors' response to my review as well as my co-reviewers' feedback and corresponding responses by the authors. Thanks to the authors for clarifying my concerns as well as doing some additional experiments for the open-weight models.
>
> I'm still on the fence regarding the veracity of the LLM-as-a-judge setup, with post-human validation done by the authors themselves, which may introduce bias. Also, is there any discussion on the links between variations of functional diversity to factors such as model scale/size, language coverage, effect of preference optimization vs no preference optimization? This would strengthen the paper.

---

> > ### Author Response · Authors · 2025-11-27
> > **Response to Reviewer Acknowledgement of Rebuttal**
> >
> > Thank you for reading our rebuttal and continued feedback on our work!
> >
> > On the "veracity" of the LLM-judge setup, we emphasize that our evaluation is fully transparent and reproducible. We include all judge prompts (Appendix A.5–A.6) and provide examples illustrating how we define functional diversity for each task category (Appendix Tables 10–11). We also include a clear disclaimer in the Discussion (lines 475–480) about the limitations of our LLM-judge approach and note that a future user study on functional diversity could build on our results.
> >
> > On “factors such as model scale/size, language coverage, effect of preference optimization”, we have further revised the paper to discuss this! Specifically, we highlight:
> > 1. Lines 387-388 mentions: “Among temperature-sampled outputs, smaller open-weight models tend to have less homogenization than larger commercial models, possibly due to their less extensive alignment.”
> > 2. Lines 432-435 discuss: “With task-based metrics, the diversity-quality tradeoff is more noticeable for smaller open-weight models (Appendix Figures 11-12), but still small (about 0.5 on our 5-point scale). One reason may be that smaller open-weight models have much lower task classification accuracy (only about 50\% compared to 85\% for commercial models)."
> > 3. Lines 435-438 notes that “system prompt sampling may have lower response quality as the number of generated responses approaches the maximum output length for a single generation”.
> > 4. We have added preliminary experiments exploring whether functional diversity collapses during DPO and GRPO. Lines 452-455 discuss that: “Future work may further explore how [functional diversity] applies to alignment. While previous studies show that token entropy collapses during alignment (Lanchantin et al. 2025b), our preliminary results show that functional diversity does not necessarily collapse (Appendix Figures 15-16).
> > 5. Lines 470-475 include a clear disclaimer about language: “An important limitation is that our taxonomy is English-centric; we only define functional diversity concepts in English, and we only evaluate English prompts. Future work may adapt or expand our taxonomy and evaluation approach.”
> > 6. Lines 489-496 provide an extended discussion on how “our task-anchored approach may be applied to other diversity-promoting methods that modify the alignment process”.

---

### Official Review · Reviewer_b2mQ · 2025-10-26

**Soundness:** 3
**Presentation:** 3
**Contribution:** 2
**Rating:** 4
**Confidence:** 4

**Summary:**

This paper studies output homogenization in large language models, arguing that it should be evaluated and mitigated in a task-dependent way rather than uniformly across tasks. The authors develop an eight-category task taxonomy distinguishing settings where consistent outputs are desirable versus those where diversity is preferred. They introduce task-anchored functional diversity and a task-anchored sampling approach that adapts decoding to the task’s diversity requirements. Experiments on multiple LLMs suggest that this approach improves meaningful diversity without degrading overall quality.

**Strengths:**

- The paper raises an important and underexplored perspective by framing output homogenization as a task-dependent issue rather than a universal flaw of LLMs.

- The proposed task taxonomy and task-anchored sampling offer a clear and interpretable framework that connects conceptual reasoning about diversity with practical decoding strategies.

**Weaknesses:**

- The paper’s main conclusion that task-anchored sampling resolves the diversity-quality trade-off relies entirely on LLM-judge evaluations. Both “functional diversity” and “checklist-based quality” are assessed by aligned models under task-specific prompts, and only the former is partially validated against human judgments. Without independent human validation for the quality metric, the claim of “no trade-off” may reflect a biased artifact of LLM-based evaluation rather than a genuine improvement in human-perceived output quality. This circular setup weakens the paper’s main result.

- The eight-category task taxonomy that supports the framework is neither empirically grounded nor fully supported by data. Some categories, such as “Problem Solving or Design Subjective”, lack any corresponding dataset, while others have ambiguous or overlapping boundaries, such as “Encyclopedia Inquiry” and “Advice or Opinions”. In addition, the taxonomy itself is generated and validated by LLMs rather than humans, which risks reinforcing model-specific biases instead of revealing genuine task distinctions. This undermines the robustness and generality of the proposed task-dependent framing.

- The proposed task-anchored sampling depends on accurate task classification, yet this step is never evaluated. The experiments assume access to ground-truth task labels derived from model majority votes, meaning the system is tested under idealized conditions. In real-world scenarios, even moderate misclassification would cause the model to apply the wrong sampling behavior, reducing quality in exactly the tasks where consistency or factual accuracy is most important. The lack of analysis of this dependency leaves the practical reliability of the approach uncertain.

**Questions:**

N/A

---

> ### Author Response · Authors · 2025-11-19
> **Response to Reviewer b2mQ**
>
> We thank the reviewer for your time and thoughtful feedback! We appreciate that you agree with the main contributions and strengths of the work. Specifically, we are encouraged that you noted “the paper raises an important and underexplored perspective by framing output homogenization as a task-dependent” and that “the proposed task taxonomy and task-anchored sampling offer a clear and interpretable framework that connects conceptual reasoning about diversity with practical decoding strategies.”
>
> In the following comments, we address each of the reviewers' questions and concerns. Given our responses, we kindly ask that the reviewer consider increasing their score.

---

> ### Author Response · Authors · 2025-11-19
> **Response to "...relies entirely on LLM judge evaluations"**
>
> We agree that the LLM-judge approach has certain limitations. We have updated the Discussion section (page 9, lines 475-480) to clearly acknowledge this limitation and to cite relevant work that documents reliability concerns with LLM-judges. Regarding “human validation for the quality metric”, we want to highlight that we adopt the checklist-based quality approach from previous works in last year’s ICLR conference (WildBench and RocketEval). These works extensively validated checklist-based grading with human annotation and preferences.
>
>
> We further independently verify each of the generated checklists (one per prompt and judge), and have clarified this in the revised paper (page 7, line 350).  We include examples of generated checklists in Appendix Table 12, and have added examples of “Very Good” responses in Appendix Table 13. In several cases (as Table 13 shows), we observed that reward models (used for quality assessment) assigned lower scores to responses that, from a human perspective, were of comparable or even superior quality. This highlights the limitations of automated reward models and underscores the value of human review. We also want to note that our evaluation is not circular: the generation and judgement processes are independent, and human validation is used to cross-check the judges.

---

> ### Author Response · Authors · 2025-11-19
> **Response to "the framework is neither empirically grounded nor fully supported by data"**
>
> We appreciate the opportunity to clarify the motivation and support for the task taxonomy. The review inaccurately states that “the taxonomy itself is generated and validated by LLMs rather than humans”. The taxonomy is developed by the authors to highlight different functional diversity concepts, which are conceptually motivated by the existing literature on output homogenization which often studies homogenization in specific task domains. We have added a new Appendix section A.1 (page 16-17) to help clarify this. In particular, Appendix Table 3 maps our taxonomy to related works.
>
>
> While our taxonomy is meant to clarify different functional diversity concepts, we also show that it captures many real-world LLM use-cases. We have added Appendix Tables 4 and 5 to map common use-cases for ChatGPT and Claude (identified in recent work) with categories in our taxonomy. We show how all text-based use-cases correspond to at least one category in our taxonomy.
>
>
> Regarding the point about “overlapping boundaries”, we agree that some tasks could apply to multiple categories or fall outside our taxonomy. In our human annotation of ground-truth task categories, only 11 (out of 300) prompts received conflicting labels, which the annotators resolved after discussion. While we limit our evaluation to one category per task, these tasks could be evaluated for multiple functional diversity concepts (e.g. responses should reflect different perspectives and creative elements). Our taxonomy is one example of a task-based categorization, but our evaluation and mitigation approaches are generalizable to alternative taxonomies and categorizations.

---

> ### Author Response · Authors · 2025-11-19
> **Response to "The proposed task-anchored sampling depends on accurate task classification"**
>
> Thanks so much for raising this point! We have adjusted our experiments to evaluate functional diversity based on human-annotated task categories. Our task-anchored sampling does use the model’s task categorization for each prompt, in order to test the practical setting as you note. We have also added the accuracy of task classification in Appendix A.3, the agreement rate with ground-truth categories is 82% for GPT-4o, 86% for Claude-4-Sonnet, and 84% for Gemini-2.5-Flash.
>
> Overall, we find that these adjustments do not change our conclusions since the task classification accuracy is high, and because models often tend to misclassify similar tasks (e.g. encyclopedia vs opinions), where the diversity concepts are similar. Thus, even with these minor misclassifications, we observe marginal differences in quality based on the checklist-based grading metric.

---

### Official Review · Reviewer_wNpm · 2025-11-01

**Soundness:** 4
**Presentation:** 4
**Contribution:** 4
**Rating:** 8
**Confidence:** 4

**Summary:**

The paper makes the following key contributions. They observe a gap in the literature that prior work largely discusses the output diversity of LLMs in a task-agnostic manner. This might lead one to assume that homogenization is 'always an issue'. To address this gap, they propose to measure 'functional diversity' or the effective diversity, or the number of valid solutions generated by a system that is anchored in the expected answer format for the prompt/task. They also provide a comprehensive taxonomy of tasks, dividing them based on the expected answer format, which helps operationalize this definition into a formal evaluation. Through experiments with 3 frontier LLMs (GPT-4o, Gemini-2.5-Flash, Claude-4-Sonnet), they show that targeted task-anchored prompting strategies (basically a prompt that provides some functional definition of the expected output) can increase functional diversity for tasks where it is desired. The paper is well motivated, clearly written, and provides sufficient evidence for its claims.

**Strengths:**

1. The positioning and framing of this paper are really good. It notices an over-simplified finding in the literature (about homogenizing of output of LLMs as a single amorphous truth in all tasks), provides a clear and tractable method (by categorizing tasks) to create a more usable definition of functional diversity, and provides benchmarking of simple methods to mitigate homogenization as needed.

2. The definition of functional diversity is clear, and also references past work that implements versions similar to it.

3. The proposed taxonomy is extensive (and the authors also note that it is non-exhaustive in a refreshing change from recent ML discourse!), and the experiments provide clear evidence for the value of task-specific prompting techniques in domains where diversity is desired. Fig. 2 displays this result really well.

4. Section 4.3 contextualizes prior findings of a trade-off between diversity and output quality as one of having mismatched metrics and that when measured correctly, these are not always at odds with one another.

**Weaknesses:**

1. The definition in Def 3.1 should be extended to a set of responses $y \in Y$ since that is really the thing you're interested in, and what you measure in the experiments (L.313-317). Given a fixed sampling budget (an answer set size), what is the effective number of unique responses in it?

2. This is more for reproducibility, but I would advocate for experiments with an open-weight LLM since the models on which results are reported are all black-box and can be updated under the hood.

3. For Section 4.3, before making a strong claim of 'not seeing an effect', I think you should also evaluate on tasks with a very large set of responses (and not just 5). It could just be that the effect is not observed at the current sampling budget.

**Questions:**

1. Re footnote 2 on page 6 - Aren't MacGyver [1] or CoPoet [2] as used in [3] examples of a task in Category E, i.e., "Tasks to solve a problem with many verifiable solutions"? Basically, these involve coming up with solutions to real-world physical reasoning problems ('Iron a shirt with a coat hanger, steamer, and kettle') or creative instructions that have semantic constraints and stylistic variants.

2. In [4], they find that many diversity measures are correlated with simple information compression algorithms like GZIP. I'm curious how well functional diversity correlates with information compression.

3. In Fig. 3, L.422, should the caption say (a) and not (a)-(c)?

4. It would be interesting if you could compare base and aligned models on functional diversity, given past findings [5, 6] that feedback tuning reduces diversity.

[1] Tian, Yufei, et al. "MacGyver: Are Large Language Models Creative Problem Solvers?." Proceedings of the 2024 Conference of the North American Chapter of the Association for Computational Linguistics: Human Language Technologies (Volume 1: Long Papers). 2024.

[2] Padmakumar, Vishakh, et al. "Beyond Memorization: Mapping the Originality-Quality Frontier of Language Models." arXiv preprint arXiv:2504.09389 (2025).

[3] Chakrabarty, Tuhin, et al. "Help me write a poem: Instruction Tuning as a Vehicle for Collaborative Poetry Writing." Proceedings of the 2022 Conference on Empirical Methods in Natural Language Processing. 2022.

[4] Shaib, Chantal, et al. "Standardizing the Measurement of Text Diversity: A Tool and a Comparative Analysis of Scores." arXiv preprint arXiv:2403.00553 (2024).

[5] West, Peter, and Christopher Potts. "Base models beat aligned models at randomness and creativity." arXiv preprint arXiv:2505.00047 (2025).

[6] Padmakumar, Vishakh, and He He. "Does Writing with Language Models Reduce Content Diversity?." The Twelfth International Conference on Learning Representations.

---

> ### Author Response · Authors · 2025-11-19
> **Response to Reviewer wNpm**
>
> We thank the reviewer for their time, deep engagement with the work, and thoughtful feedback! We are encouraged that the reviewer notes that the “positioning and framing of this paper are really good”, the work “provides a clear and tractable method” and “provides benchmarking of simple methods”. Further, the reviewer underlines that “the definition of functional diversity is clear” and “the proposed taxonomy is extensive”. And the reviewer highlights the quality-diversity tradeoff clarification as a strength.
>
> In the following comments, we address each of the reviewers' concerns and questions. Given our responses, we kindly ask that the reviewer consider maintaining their score.

---

> ### Author Response · Authors · 2025-11-19
> **Responses to Reviewer-Identified Weaknesses**
>
> 1. The reviewer notes “The definition in Def 3.1 should be extended to a set of responses”. Good point! In our revised paper, we have added a definition (page 4, Def 3.2) for the number of functionally unique responses in a set.
>
> 2. Thanks for the suggestion to add experiments with open-weight LLMs! We have added results for Llama-3.1-8B-Instruct and Mistral-7B-Instruct-v0.3. Please see our comment titled “Evaluating Open-Weight Models” in our Summary of Revisions. In general, we find that our task-anchored sampling is able to promote functional diversity in these open-weight LLMs as well.
>
> 3. The reviewer notes that “I think you should also evaluate on tasks with a very large set of responses (and not just 5). It could just be that the effect is not observed at the current sampling budget.” This is an interesting point, we are also curious if generating more than 5 responses would reduce quality. Given that pairwise comparisons grow quadratically, we chose to only generate 5 responses per prompt (5 choose 2 = 10 comparisons for functional diversity). But we can run a few experiments with larger N to investigate if our claim holds … we will include those results in the Appendix and revise Section 4.3 accordingly to soften our quality-diversity trade-off claim.

---

> ### Author Response · Authors · 2025-11-19
> **Responses to Reviewer Questions**
>
> 1. We appreciate that the reviewer points out datasets that would allow us to evaluate category E. Thank you! We have added experiments based on the MacGyver dataset. Figure 2 shows how GPT-4o can produce around 3-4 functionally unique responses with our task-based sampling, while temperature sampling only produces 1 unique response. The results are similar for other models (Appendix page 30, Figures 5-8). In particular, Claude-4-Sonnet is able to produce almost 5 functionally unique responses for category E using our sampling methods (Figure 5).
>
> 2. We’ve added results for a compression diversity metric (based on GZIP) in Appendix Tables 21-22. On average, compression diversity has a pearson correlation of 0.22 with functional diversity. In comparison, token diversity has a pearson correlation of 0.47 with functional diversity. This suggests that compression diversity may not be representative of meaningful functional differences between responses. One reason could be that many of our evaluation prompts involve shorter responses, and GZIP may have some overhead where it cannot reduce size effectively for short responses.
>
> 3. We fixed this minor typo: “In Fig. 3, L.422, should the caption say (a) and not (a)-(c)?”
>
> 4. The reviewer notes “It would be interesting if you could compare base and aligned models on functional diversity, given past findings [5, 6] that feedback tuning reduces diversity.” We agree! Since we added results for Llama-3.1-8B-Instruct and Mistral-7B-Instruct-v0.3, we plan to use these models to add preliminary alignment experiments in the Appendix (since they are smaller models and feasible to fine-tune). So far, we have observed that functional diversity does not collapse for Llama-3.1-8B during DPO and GRPO (although the logit entropy decreases significantly). We will clarify in the main paper that this could be an interesting direction for future work.

---

> ### Comment · Reviewer_wNpm · 2025-11-25
> **Rebuttal Acknowledgment**
>
> Thanks again to the authors for adding more results based on review feedback! In particular, adding open-weight models, evaluating the effect of feedback tuning, and calculating the correlation to diversity measures improve the quality of the draft. I agree that cost is prohibitive for tasks with a very large sample size, and softening the claim is the better way to go about it. I'm retaining my high score of 8 for the paper - a well-written work that synthesizes observations from prior work into a clear set of findings. I'd request the AC to nudge other reviewers to respond to the rebuttals as well.

---

### Official Review · Reviewer_cajZ · 2025-11-02

**Soundness:** 3
**Presentation:** 3
**Contribution:** 3
**Rating:** 6
**Confidence:** 5

**Summary:**

This paper hypothesizes that quantifications and inductions towards output homogenization should be task dependent. They develop a task taxonomy enlisting which kind of variations is meaningful for that task and use that to construct prompts that describe task-dependent axes of variations to guide the generation more adaptively (to the task’s homogenization needs). Their task-anchored method increases diversity and does not reduce quality.

**Strengths:**

- The key hypothesis is unique and has high potential for explaining the shortcomings of several works that categorize the diversity of the LM generations
- Highlights an interesting shortcoming about prior work using scalar rewards (task-agnostic) to assign quality scores.

**Weaknesses:**

- The task taxonomy and prompt design appear somewhat arbitrary and intuition-based, not derived from user studies or large-scale data.
A small ablation that applies mismatched task prompts (e.g., creative prompt for factual task) would strengthen confidence that each mapping is truly optimal.Since the sampling methods themselves are not entirely new, the value of the paper rests heavily on how sound and justifiable these task definitions are.

- All evaluations rely on LLM judgments. While the use of checklist-style grading for task-relevant quality is an improvement over scalar rewards, there are no human-based validations. Because the LLM-graded metrics show larger gains for the system-prompt variant and weaker discrimination elsewhere, a human diversity-quality study would help verify whether these numerical gains reflect actual perceptual differences.

**Questions:**

- What fraction of your evaluation prompts fell into this “uncategorized” case? Could you report the performance of the methods on these?
- How accurate is the task-to-category classification? Did you run any verification (human or otherwise) to ensure that prompts were correctly labeled before applying task-specific system prompts?

---

> ### Author Response · Authors · 2025-11-19
> **Response to Reviewer cajZ**
>
> We thank the reviewer for your time and thoughtful feedback. We appreciate that you point out the main contributions and strengths of the work. In particular, the reviewer states “The key hypothesis is unique and has high potential for explaining the shortcomings of several works that categorize the diversity of the LM generations” noting that the paper does a good job of highlighting “an interesting shortcoming about prior work using scalar rewards (task-agnostic) to assign quality scores”.
>
> In the following comments, we address each of the reviewers' questions and concerns. Given our responses, we kindly ask that the reviewer consider maintaining or increasing their score.

---

> ### Author Response · Authors · 2025-11-19
> **Response to "The task taxonomy and prompt design appear somewhat arbitrary..."**
>
> We appreciate the opportunity to clarify the motivation and support for the task taxonomy! We have added Appendix A.1 in our revised paper to help clarify this (pages 16-17). The goal of our taxonomy is to highlight different functional diversity concepts, not necessarily to be exhaustive of all LLM use-cases. We designed our taxonomy to cover a variety of tasks studied in the existing homogenization literature, as Table 3 illustrates. However, we also show that our taxonomy does cover many common LLM use-cases identified for ChatGPT and Claude. In Appendix Tables 4-5, we map common LLM use-cases identified in recent work to the relevant categories in our taxonomy. We also appreciate the suggestion to conduct an ablation with mismatched task prompts, we will run this experiment to check.

---

> ### Author Response · Authors · 2025-11-19
> **Response to "All evaluations rely on LLM judgments"**
>
> We agree that the main limitation of our experiments is that we rely on LLM-judges to evaluate functional diversity and quality. We have updated the Discussion section (page 9, lines 475-480) to clearly acknowledge this limitation and to cite relevant work that documents reliability concerns with LLM-judges. In particular, we want to highlight that: 1) we did validate the functional diversity judges with human annotation, and that 2) we always report the average of 3 LLM-judges to increase reliability. As Footnote 2 on page 6 states, “we validate the LLM-judges on a stratified random sample of 225 response pairs across models, tasks, and sampling strategies. Two authors independently labeled these responses for functional diversity, and agreed 79% with the LLM-judges (80% agreement between annotators).”
>
> We further include all our judge prompts for transparency, and have also added examples of functional diversity for each task category (Appendix Tables 10 and 11). For the checklist-based quality metric, we follow the approach of previous works in last year’s ICLR conference (WildBench and RocketEval) that extensively validated checklist-based grading with human annotation.

---

> ### Author Response · Authors · 2025-11-19
> **Response to "What fraction of your evaluation prompts fell into this “uncategorized” case?"**
>
> Thanks for this question, we have updated Appendix A.3 to clarify! Based on human annotation, 6 prompts did not fit into our taxonomy (all from WildBench) due to missing information (e.g. prompts that referenced an unattached document) or language (we only evaluate English prompts). This low number of “uncategorized” prompts is expected, since we designed our taxonomy to cover common LLM use-cases as mentioned above. We exclude these 6 prompts from our results as our method relies on the default model behavior for prompts outside our taxonomy. If our approach is employed in a real pipeline and a new prompt does not fit within the existing taxonomy, practitioners can either expand the taxonomy or allow the model to default to standard behavior.

---

> ### Author Response · Authors · 2025-11-19
> **Response to "How accurate is the task-to-category classification?"**
>
> We have updated Appendix A.3 to report this, thank you for noticing the importance of the classification! In our revised paper, we always evaluate the functional diversity of responses to a given prompt based on the ground truth task category as determined by human annotation. But in order to test the practical setting, our task-based sampling method uses the model’s task classification to promote diversity. Appendix Figure 4 visualizes the accuracy of models’ task-to-category classification based on human-labeled task categories. The accuracy is 82% for GPT-4o, 86% for Claude-4-Sonnet, and 84% for Gemini-2.5-Flash.

---

### Author Response · Authors · 2025-11-20
**Global Response & Summary of Minor Revisions**

We are very appreciative of all  reviewers for their time, engagement with our work, and thoughtful feedback! We are encouraged that each reviewer highlighted strengths of our work, including our task-dependent argument, taxonomy, and task-anchored sampling approach. While we have responded to each reviewer individually, this comment provides a global response addressing common concerns and outlining the minor revisions we made.


# LLM-Judge Metrics & Human Validation
We agree with reviewers that one limitation of our experiments is using LLM-judges to evaluate functional diversity and task-based quality. We have updated the Discussion section (page 9, lines 475-480) to clearly acknowledge this limitation and to cite relevant work that documents reliability concerns with LLM-judges.

We further want to highlight:
1. We validate the functional diversity judges with human annotation. Two authors independently labeled 225 response pairs for functional diversity and agreed with LLM-judges 79% of the time, with 80% inter-annotator agreement (page 6, footnote 2).
2. For the checklist-based quality metric, we follow the approach of previous works in last year’s ICLR conference (WildBench & RocketEval) which extensively validated checklist-based grading with human annotation. We also manually confirmed that all our judge-generated checklists are appropriate for each prompt.
3. We primarily report the average of 3 different LLM-judges to increase reliability. We have further added results based on each LLM-judge separately (Appendix Tables 28-33). The average differences across individual judges are small (3% for functional diversity & 5% for checklist-based quality) and do not change our overall conclusions.
4. We include all our judge prompts for transparency (Appendix A.5-A.6), and have added examples of response pairs that are and are not functionally diverse for each task category (Appendix Tables 10-11). We also add examples of “Very Good” responses based on checklist-based grading for each task category (Appendix Table 13).


# Taxonomy Motivations
Some reviewers asked how our taxonomy relates to the homogenization literature and real-world LLM use-cases. We have added Appendix A.1 to help clarify this!

Our taxonomy is grounded in the existing literature. We observed that many studies evaluate homogenization in specific task domains, suggesting that problematic notions of homogenization are task dependent. We designed our taxonomy to cover a variety of these task categories. Appendix Table 3 shows how our taxonomy maps to related work.

While our taxonomy is meant to clarify different functional diversity concepts, we also show that it captures many real-world LLM use-cases. Appendix Tables 4 & 5 map common use-cases for ChatGPT and Claude with categories in our taxonomy, and all text-based use-cases correspond to at least 1 category in our taxonomy.

Ultimately, our taxonomy is non-exhaustive and represents one categorization of functional diversity concepts. However, our task-based evaluation and sampling approaches are generalizable to alternative task taxonomies.


# Evaluation with Human-Labeled Task Classification
We appreciate reviewers for noting that we should evaluate with ground-truth task categories! We have adjusted our experiments to evaluate functional diversity based on human-labeled task categories. In order to test the practical setting, our task-anchored sampling uses the model's task categorization for each prompt.  We further add results showing the accuracy of models’ task classification (Appendix Figure 4); GPT-4o, Gemini-2.5-Flash, and Claude-4-Sonnet are all highly accurate (~85%). Thus, our results show that 1) models can identify the appropriate task category, and 2) when given the functional diversity concept for that category, they can generate many functionally unique responses.


# Evaluating Open-Weight Models
We appreciate the suggestions to evaluate open-weight models! We have added experiments for Llama-3.1-8B-Instruct and Mistral-7B-Instruct-v0.3.

Figures 7-8 show results for functional diversity for these models, and we generally observe that our task-anchored sampling improves diversity to similar levels as commercial models. Among temperature-sampled outputs, we also note that open-weight models have higher functional diversity than commercial models, possibly due to the less extensive alignment in open-weight models.

Figures 11-12 illustrate the diversity-quality tradeoff for these models. In general, we observe a similar pattern of a large tradeoff when using general metrics, and a smaller tradeoff when using task-based metrics. However, the tradeoff is more noticeable for open-weight models (roughly 0.5 on our 5-point scale) than commercial models. One reason is that open-weight models have much lower task classification accuracy (only ~50% compared to ~85% for commercial models). Other factors may be model size and less extensive alignment.

---

### Meta-Review · Area_Chair_ovFm · 2026-01-01

**Summary:**

The paper observes that prior work broadly discusses LLM output diversity in a task-agnostic manner. This might lead one to assume that homogenization is 'always an issue'. To address this gap, they propose to measure 'functional diversity' or the effective diversity, or the number of valid solutions generated by a system that is anchored in the expected answer format for the prompt/task. They also provide a comprehensive taxonomy of tasks, dividing them by expected answer format, which helps operationalize this definition into a formal evaluation. Experiments with 3 frontier LLMs (GPT-4o, Gemini-2.5-Flash, Claude-4-Sonnet) and two open-weights LLMs (added during the rebuttal), the authors show that targeted task-anchored prompting strategies (basically a prompt that provides a functional definition of the expected output) can increase functional diversity when desired. The paper is well motivated, clearly written, and provides an extensive taxonomy of functional diversity, which could be helpful in practical settings. I summarize below the reviewers' concerns and the authors response:

1.Muplitple reviewers were unhappy with the use of LLM-judges for both functional diversity and quality metrics without sufficient human validation. The authors argued that  human validation by themselves (2 authors)  shows 79% agreement between human annotators and LLM-judges (80% inter-annotator agreement) on 225 response pairs. The authors also argued that they follow established methods from prior ICLR work (WildBench, RocketEval) that extensively validated checklist-based grading.

2. Two reviewers argued that the Task Taxonomy seems intuition-based rather than derived from user studies or data. Some categories lack datasets, have overlapping boundaries, or are validated by LLMs rather than humans. The authors added Appendix A.1 showing their
Taxonomy is grounded in existing homogenization literature (Table 3 maps to related work) and
maps to real-world use cases.

3. One reviewer argued that task-anchored sampling depends on accurate task classification, but this step was never evaluated. Real-world misclassification could apply the wrong sampling behavior. The authors revised their experiments for assessing functional diversity based on human-labeled task categories and added task classification accuracy in the Appendix, showing that even with misclassifications, models tend to confuse similar tasks with similar diversity concepts, so the quality impact is marginal

4. Two reviewers complained that only commercial black-box models were tested, reducing reproducibility and preventing investigation of factors affecting homogenization (model size, alignment effects, etc.). The authors added experiments with
Llama-3.1-8B-Instruct and Mistral-7B-Instruct-v0.3.

**Reviewer Concerns:**

The authors did their best to address the reviewers' concerns. However, two important issues have not been adequately addressed, in my opinion.

- It is simply not good enough to delegate independent human evaluation to two authors. You should probably conduct an independent user study with a large enough sample to draw meaningful conclusions. It is in your best interest and will strengthen your claims. Also, what guidelines were the authors given? Were they given any guidelines?

- It is also not good to use as judges the LLMs whose behavior you are evaluating. You should probably leave one of the commercial LLMs out as an independent judge. Show its judgments correlate with human ratings, and then use this LLM as a judge but not as an object of study. Otherwise, there are too many confounds in your study.

- If you are revising your paper, I would strengthen the open-weights part of it, include the Gemma and Qwen families and run a somewhat more substantial study to assess whether model size, instruction tuning, modality (Gemma and Qwen are multimodal) correlate with functional diversity.

- I appreciate the authors have performed correlations with existing diversity metrics and shown these are not very high. Are these correlations significant? You could also try n-gram repetition as a measure of diversity.

**Reviewer Scores:**

I am not sure any reviewer would have changed their score. Reviewer  Qpei gave a 4, read the response, and cited they were still on the fence. Reviewer wNpm was positive (gave an 8), I doubt they would increase their score. Finally reviewer cajZ have similar issues with reviewer Qpei, but gave a 6 which I doubt would increase.

---

### Decision · Program_Chairs · 2026-01-26

Reject